# WIP1 mutations suppress DNA damage triggered bypass of the mitotic timer

Tomoaki Sobajima [ID], Luke J Fulcher, Caleb Batley [ID], Susanna J Alsop [ID], Jonah Veakins & Francis A Barr [ID] [✉]

## Abstract

**Prolonged mitosis results in the destruction of MDM2, initiating a p53-dependent G1 cell-cycle arrest in the absence of DNA damage. Here, we investigate how DNA damage earlier in the cell cycle affects this mitotic-timer response. We find that G2-DNA damage triggers highly penetrant bypass of mitosis and of the mitotic timer, generating tetraploid cells arrested in G1. Collapse of G2 to G1 after DNA damage is initiated by p21-mediated CDK2 inhibition and rendered irreversible by the destruction of G2/M-cyclins A and B. This behaviour is altered in cells with cancer-associated mutations in the p53-phosphatase WIP1 (PPM1D), which increase the threshold for DNA-damage signalling, enabling DNA-damaged G2 cells to enter mitosis with elevated levels of MDM2, thereby suppressing mitotic-timer-dependent G1 cell-cycle arrest. Importantly, neither WIP1 mutations nor knockout prevent p53-dependent G1-arrest in response to prolonged mitosis in the absence of DNA damage. Prolonged mitosis and G2-DNA damage thus promote p53-dependent G1 cell-cycle exit through discrete routes with differential requirements for WIP1 and genotoxic stress.**

**Keywords** Cell Cycle; Mitosis; Cell Cycle Checkpoints; DNA Damage
**Subject Categories** Cell Cycle; DNA Replication, Recombination & Repair

## Introduction

Conserved checkpoints delay cell cycle progression or promote cell cycle exit in response to DNA damage, thus preventing genome instability and aneuploidies commonly associated with cancers (Matthews et al, 2022). During G1 and G2 cell cycle phases, the DNA damage response results in stabilisation of a conserved transcription factor p53 and subsequent p53-dependent induction of the cyclin-dependent kinase (CDK) inhibitor protein p21. This response limits CDK activity in G1 and prevents passage through the restriction point into S phase if damaged DNA is present (Blackford and Jackson, 2017). If DNA damage or DNA replication stress occurs in late-S or G2 phase, DNA damage signalling, in addition to regulation of p53, simultaneously reinforces the activity

of the CDK1 inhibitory kinases Wee1 and Myt1 and inhibits the CDK1-activating phosphatase Cdc25, thereby preventing mitotic entry in the presence of DNA damage (Donzelli and Draetta, 2003; O'Connell et al, 2000). In addition, prolonged DNA damage in G2 or replication stress caused by cyclin E overexpression can drive p53- and p21-dependent cell cycle exit from G2 to G1 without cell division due to the inhibition and untimely destruction of the G2 and mitotic cyclins A and B, respectively (Gallo et al, 2022; Johmura et al, 2014; Krenning et al, 2014; Mullers et al, 2014; Zeng et al, 2023). Inactivation of these G1 and G2 DNA damage checkpoints is frequently observed in aneuploid and chromosomally or otherwise genomically unstable cancers. Most notably, loss-of-function mutations in components of these G1 and G2 checkpoint pathways are often viewed as a prerequisite for tumorigenesis and the proliferation of aneuploid cells (Abbas and Dutta, 2009; Otto and Sicinski, 2017; Vousden and Prives, 2009). Thus, p53-dependent checkpoints delay cell cycle progression or promote cell cycle exit to prevent the replication and inheritance of potentially damaged DNA, a protective cytostatic response lost in many cancers.

During mitosis, the spindle assembly checkpoint prevents anaphase onset until all chromosomes are bioriented, thereby ensuring equal distribution of the genome between the daughter cells and reducing the chance of genome instability and aneuploidy (Li and Zhu, 2022; Santaguida and Amon, 2015; Yang et al, 2008). Although the spindle assembly checkpoint is independent of p53, p53 nevertheless has a key role monitoring the period of checkpoint activation. When spindle checkpoint-dependent arrest exceeds a threshold time, indicative of problems in chromosome alignment and segregation associated with aneuploidy, these potentially damaged or aneuploid daughter cells undergo p53-dependent G1 cell cycle arrest in the ensuing G1 (Thompson and Compton, 2010; Uetake and Sluder, 2010). In contrast to the canonical p53-dependent pathway, this mitotic timer response occurs in the absence of detectable DNA damage, suggesting it has alternative causes that delay progression through mitosis, including centrosome aberrations and aneuploidy (Fong et al, 2016; Meitinger et al, 2016; Meitinger et al, 2024; Meitinger et al, 2020; Thompson and Compton, 2010; Wong et al, 2015; Yang et al, 2008). MDM2, the conserved p53 E3 ubiquitin ligase (Haupt et al, 1997; Honda et al, 1997; Kubbutat et al, 1997), has been shown to be a key timer component in this pathway (Fulcher et al, 2025). During mitosis, MDM2 is slowly destroyed by a self-catalysed ubiquitination

Department of Biochemistry, University of Oxford, South Parks Road, OX1 3QU Oxford, UK. ✉E-mail: francis.barr@bioch.ox.ac.uk

mechanism and is not replenished due to the global attenuation of protein synthesis and loss of MDM2 mRNA (Fulcher et al, 2025). Thus, if mitosis completes within the normal time frame of under 60 min, there is still sufficient MDM2 present in early G1 daughter cells to limit p53 activity and allow passage into the next cell cycle. However, if mitosis is delayed beyond a threshold time of 60 min in hTERT-RPE1 cells, MDM2 becomes limiting in daughter cells, leading to p53 stabilisation, p21 induction and G1 cell cycle arrest (Fulcher et al, 2025). Thus, even brief mitotic delays can trigger a p53-dependent G1 arrest in the daughter cells, demonstrating that spindle assembly checkpoint signalling in mitosis can influence cell cycle fate in the next G1 phase. Intriguingly, other components of this pathway include a PLK1-regulated complex of the potential p53 deubiquitinating enzyme USP28 and the p53-interacting protein 53BP1 (Burigotto et al, 2023; Fong et al, 2016; Lambrus et al, 2016; Meitinger et al, 2016; Meitinger et al, 2024; Zhang et al, 2006). MDM2 and USP28-53BP1 therefore work together to set the threshold for p53-dependent cell cycle arrest in G1 cells following delays in mitosis.

A key difference between the mitotic timer pathway and G1 or G2 checkpoints is their DNA damage dependence (Fig. 1A). Thus, one might expect DNA damage signalling kinases and their counteracting phosphatase regulators to play a less crucial role in the DNA damage-independent mitotic timer pathway. However, some evidence suggests that p53 wild-type tumour cell lines carrying heterozygous gain-of-function mutations in the p53-phosphatase WIP1 (PPM1D) are defective for G1 arrest through the mitotic timer pathway (Meitinger et al, 2024). One caveat is that tumour cell lines deficient for the mitotic timer response carry many other mutations, and thus this correlation has not been confirmed as a causal change leading to attenuation of the timer response. Indeed, a role for WIP1 in the mitotic timer pathway would bring in to question the notion that this is a DNA damage-independent pathway and might suggest very low levels of DNA damage are in fact present in the daughter cells undergoing cell cycle arrest in G1. To explore these questions further, we set out to test how DNA damage events in the preceding G2 phase would impact the mitotic timer pathway in the subsequent mitosis and whether the DNA damage signalling phosphatase WIP1 has a direct role in the mitotic timer response.

## Results

### DNA damage in G2 triggers p53- and p21-dependent bypass of the mitotic timer

To understand how DNA damage might influence the mitotic timer pathway and behaviour of new G1 cells arising from cell division, it was important to first establish conditions that created transient low levels or higher levels of DNA damage in G2 prior to mitotic entry, yet did not result in elevated cell death. To achieve this, p53$^{WT}$ and p53$^{KO}$ hTERT-RPE1 cells (Fig. EV1A) were treated with a 1 h pulse of the DNA double-strand break-inducing ionizing radiation mimetic drug Neocarzinostatin (NCS) (Ishida et al, 1965). This resulted in a dose-dependent reduction in proliferation in p53$^{WT}$ cells with only a small effect on p53$^{KO}$ cells up to 100 ng/ml NCS (Fig. 1B). At 200 ng/ml NCS there was an abrupt fall in cell proliferation in both p53$^{WT}$ and p53$^{KO}$ hTERT-RPE1 cells

suggesting this is a toxic level of DNA damage. These effects were matched by an increase in the number of 53BP1 and γ-H2A.X foci within the nucleus (Figs. 1C and EV1B), indicating successful DNA damage induction (Rogakou et al, 1998; Schultz et al, 2000). In agreement with microscopy data, Western blotting showed dose-dependent increases in γ-H2A.X and phosphorylation of p53 on Ser15, an established p53 phosphorylation site in the DNA damage response (Shieh et al, 1997), which were temporally coincident with an increase in total p53 protein levels (Fig. EV1C,D). To test if the NCS-induced DNA damage was reversible, cells were treated with increasing doses of NCS for 1 h and then washed out into nocodazole-containing medium to capture any cells that had been able to enter mitosis, indicative of successful DNA damage repair. This revealed that increasing doses of NCS resulted in a reduction in the number of cells entering mitosis after the pulse of DNA damage (Fig. 1C). Failure to enter mitosis was inversely correlated with the number of 53BP1 foci. Below an NCS concentration of 25 ng/ml cells were observed to enter mitosis, suggesting that the DNA damage was repairable and the G2 checkpoint was silenced (Fig. 1C). However, above 50 ng/ml NCS, no mitotic cells were observed, suggesting this level of damage was not repairable within the experimental time frame and G2 checkpoint signalling was maintained (Fig. 1C).

To explore the cell cycle fate of hTERT-RPE1 cells after DNA damage, a FUCCI sensor was used to generate colour-coded plots of cell cycle trajectories as a function of time to indicate the cell cycle fate of individual cells (Koh et al, 2017; Sakaue-Sawano et al, 2008). Two DNA damage conditions were defined for these experiments, low (12.5 ng/ml NCS) and high damage (100 ng/ml NCS), respectively. This revealed different cell cycle fates for G2 cells exposed to these low or high levels of DNA damage (Figs. 1D,E and EV1E). Control p53$^{WT}$ cells with no DNA damage spent an average of 6.0 ± 3.1 h in G2 before entering mitosis and dividing, with 10.0 ± 1.8 h spent in G1 before carrying on into the next cell cycle (Figs. 1D–F and EV1E, p53$^{WT}$ no DNA damage). Low DNA damage increased G2 length to 15.3 ± 8.0 h, with 83% of cells entering mitosis and dividing (Figs. 1D–F and EV1E, p53$^{WT}$ low DNA damage). Mean G1 duration was also increased from 10.0 ± 1.8 h to 26.1 ± 11.2 h in response to low levels of DNA damage (Fig. 1G, p53$^{WT}$ no and low DNA damage). These findings show that cells can delay in response to DNA damage in G2, then continue in the cell cycle and arrest or delay in G1 of the immediately following cell cycle, a notion supported by previous observations (Barr et al, 2017). This suggests either the continued presence of, or a memory of DNA damage in cells passing from G2 through mitosis to G1 that could affect the threshold for the mitotic timer response.

Cells exposed to higher levels of DNA damage in G2 showed a different behaviour, with 98 ± 2% of p53$^{WT}$ cells undergoing cell-cycle collapse from G2 to a G1 like state in the absence of mitosis (Figs. 1D,E and EV1E, p53$^{WT}$ high DNA damage). Under these conditions, G2 was lengthened to 20.5 ± 8.0 h compared to 6.0 ± 3.1 h in the absence of DNA damage (Fig. 1F, p53$^{WT}$ high DNA damage). In agreement with the mode-of-action of NCS, inhibition of the upstream DNA damage kinases ATM/ATR in p53$^{WT}$ cells prevented this cell-cycle collapse in response to DNA damage, and over 80% of cells entered mitosis (Fig. 1E, p53$^{WT}$ ATMRi high DNA damage). ATM/ATR inhibition reduced G2 length to 11.9 ± 6.9 h, an expected outcome following reduced DNA

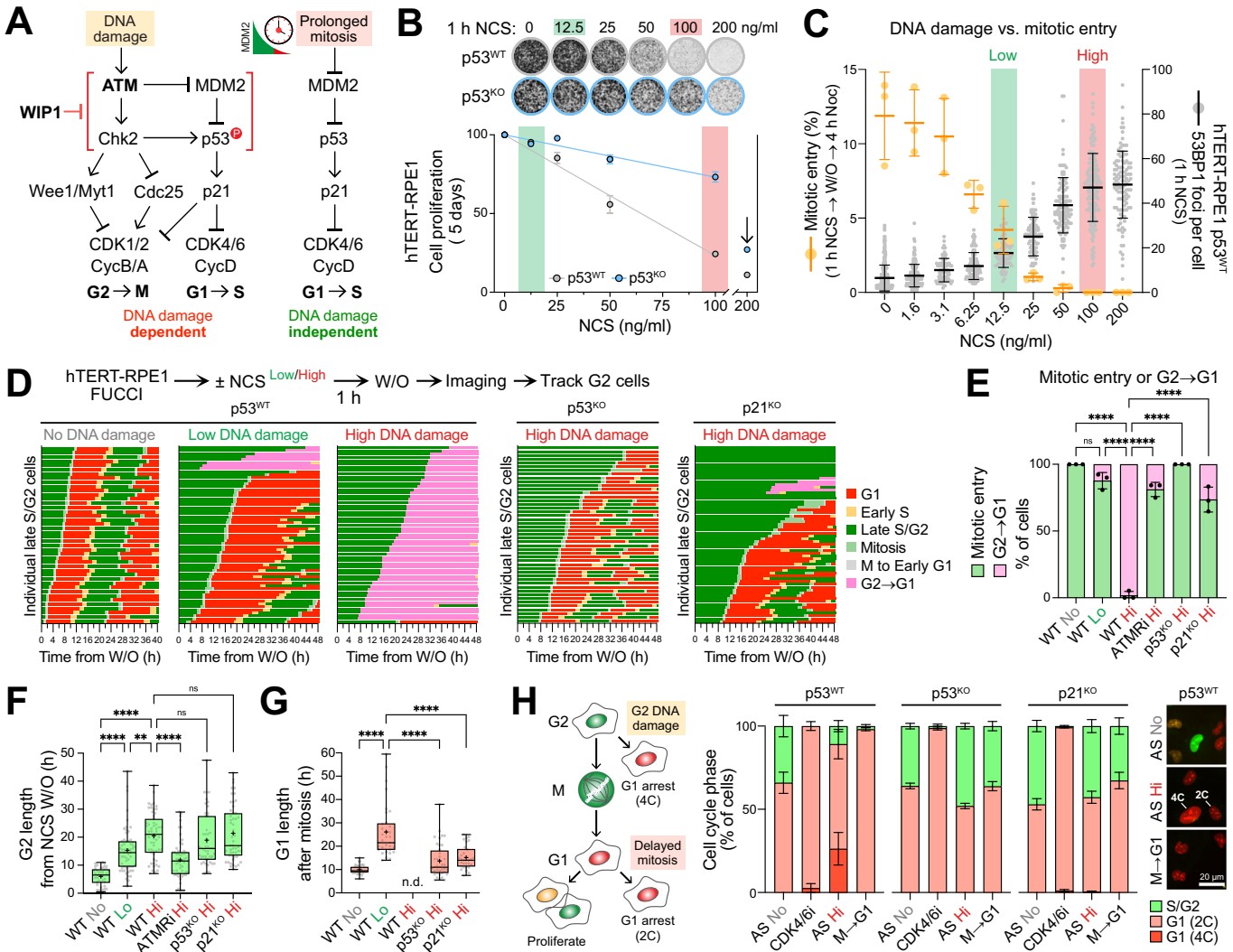

**Figure 1. DNA damage in G2 triggers either a transient cell cycle arrest or a bypass of mitosis and arrest in G1.**

(A) Schematic highlighting the DNA damage-dependent and independent p53 pathways functioning during the cell cycle. (B) p53[WT] and p53[KO] hTERT-RPE1 FUCCI cells were treated with the indicated doses of NCS for 1 h. NCS was removed by washing and cells allowed to proliferate for 5 days before fixing and staining with crystal violet. Relative cell proliferation is quantified underneath. Low dose (12.5 ng/ml) and high dose (100 ng/ml) NCS doses used in subsequent experiments are highlighted in green and red, respectively (mean ± SEM; $n = 3$ independent experiments). (C) p53[WT] hTERT-RPE1 FUCCI cells were treated with the indicated doses of NCS for 1 h. NCS was removed by washing and cells were either fixed and stained immediately for the number of 53BP1 foci or after 4 h treatment with 25 ng/ml nocodazole to determine the percentage of mitotic entry (mean ± SD; $n = 3$ independent experiments). (D) p53[WT] hTERT-RPE1 FUCCI cells were treated with either low dose or high dose NCS for 1 h. NCS was removed by washing and Late S/G2 cells were imaged continuously for 2 days. Cell cycle fate is plotted for individual cells. Untreated cells with no DNA damage were imaged and plotted as a control (left). The analysis was repeated in p53[KO] and p21[KO] hTERT-RPE1 FUCCI cells (right). Pooled analyses are shown from 3 independent experiments. (E) The percentage of mitotic entry and G2 to G1 cell-cycle collapse is plotted from (D). ATM and ATR inhibitor treated cells (ATMRi) are included as a control (mean ± SD; $n = 3$ independent experiments). Statistical significance was analysed using an ordinary one-way ANOVA with Tukey's multiple comparisons test. (F) G2 length in cells treated as in (D) is plotted for individual cells in a box and whiskers plot (median, 25th and 75th percentiles and whiskers extending to minimum and maximum values; mean (+) for the different conditions. ATM and ATR inhibitor treated cells (ATMRi) are included as a control. Pooled analyses are shown from three independent experiments. Statistical significance was analysed using an ordinary one-way ANOVA with Tukey's multiple comparisons test. (G) G1 length following mitosis in cells that entered mitosis following treatments as in (D) is plotted for individual cells in a box and whiskers plot (median, 25th and 75th percentiles and whiskers extending to minimum and maximum values, mean (+) for the different conditions; pooled analyses shown from three independent experiments). n.d.: no data points could be obtained due to no cells entering mitosis. Statistical significance was analysed using an ordinary one-way ANOVA with Tukey's multiple comparisons test. (H) Schematic detailing possible cell cycle fates of cells after DNA damage (left). DNA damage was induced in asynchronous (AS) p53[WT], p53[KO] and p21[KO] hTERT-RPE1 FUCCI cells by a high dose of 100 ng/ml NCS (AS Hi) for 24 h. DMSO (AS No) was used as a negative control. Cells held in mitosis (M) for 18 h and released into G1 for 24 h were used to test the outcome of mitotic timer pathway induced G1 arrest (M → G1). Cells trapped in G1 with the CDK4/6 inhibitor Palbociclib (CDK4/6i) for 24 h were used as a control for normal G1 arrest. The percentage of S/G2 and 2C/4C G1 cells is plotted (mean ± SEM, $n = 3$ independent experiments) (middle). Representative images of some key conditions are shown. Scale bar: 20 µm (right). Significance for all experiments: **$P < 0.01$; ****$P < 0.0001$; ns not significant. All $P$ values are listed in Dataset EV1.

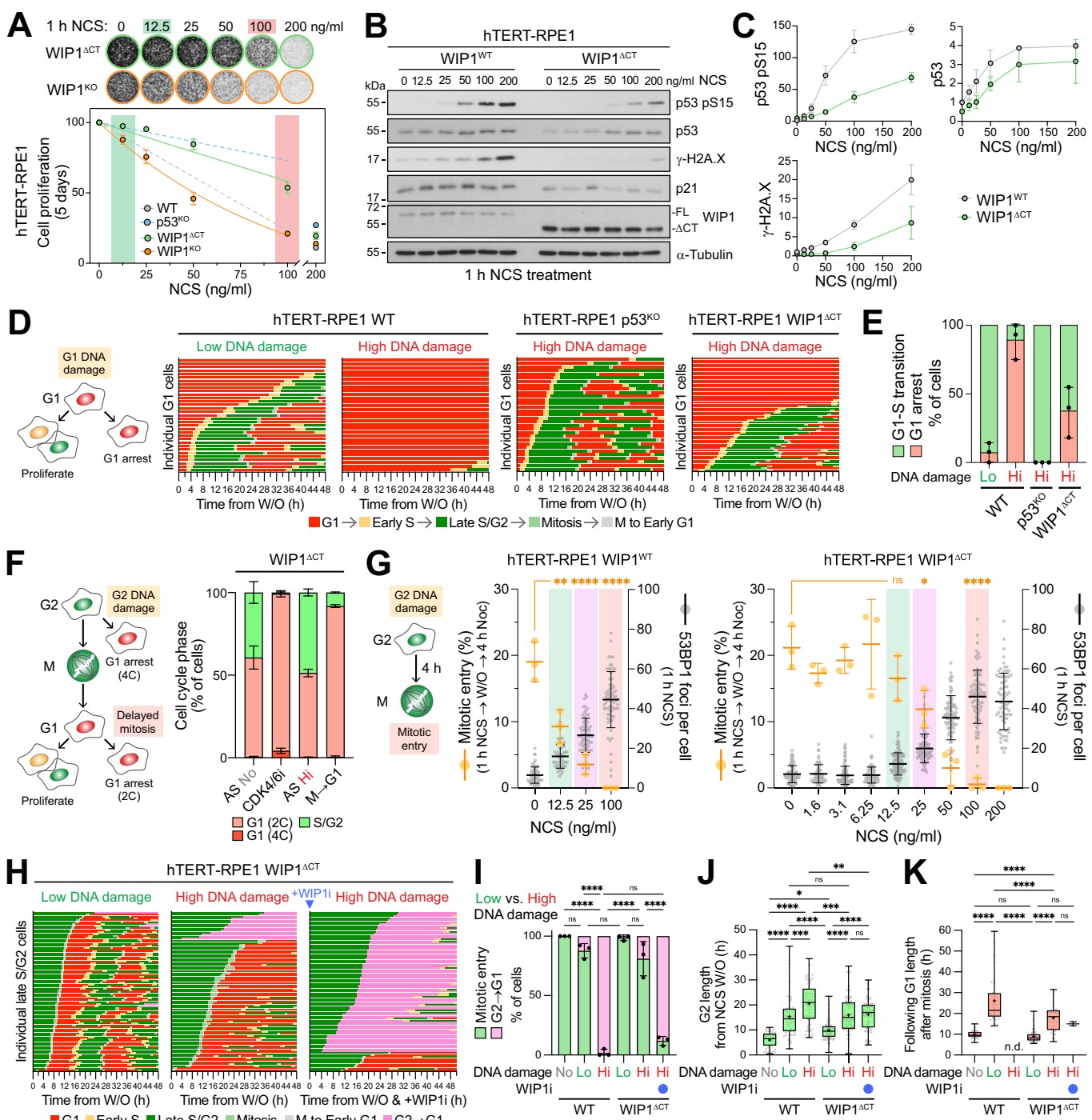

damage signalling to the G2 checkpoint (Fig. 1F, p53[WT] no, high, and ATMRi high DNA damage). Cells moving from G2 directly to G1 remained arrested in G1 for the duration of the experiment up to 48 h (Figs. 1D and EV1E). Since no cells passed through mitosis after high DNA damage, it was not possible to measure the length of the following G1. G2 to G1 cell cycle collapse was not readily observed in p53 or p21 knockout cell lines (Fig. 1D,E, p53[KO] or p21[KO] high DNA damage) indicating this is a p53- and p21-dependent response. Importantly, G2 length was still extended in

p53[KO] and p21[KO] cells to 19.0 ± 9.3 h and 21.4 ± 9.7 h, respectively, showing that they still have a functional G2 checkpoint (Fig. 1F, p53[KO] and p21[KO] high DNA damage). Together, these findings are consistent with the idea that p21 is the major p53 target required for both the G2 to G1 cell cycle collapse and subsequent G1 arrest, whereas neither p53 nor p21 are directly involved in controlling the length of G2. We conclude that in the absence of p53 or p21, the canonical G2 DNA damage checkpoint holds cells in G2 for an extended period up to ~20 h after which time cells then break

◀ **Figure 2. G2 DNA damage-dependent bypass of mitosis and G1 arrest are attenuated in WIP1$^{\Delta CT}$ cells.**

(A) WIP1$^{\Delta CT}$ and WIP1$^{KO}$ hTERT-RPE1 FUCCI cells were treated with the indicated doses of NCS for 1 h. Relative cell proliferation after 5 days is quantified underneath. Low dose (12.5 ng/ml) and high dose (100 ng/ml) NCS doses are highlighted in green and red, respectively (mean ± SEM; $n = 3$ independent experiments). (B) DNA damage markers were assessed by Western blotting of WIP1$^{WT}$ and WIP1$^{\Delta CT}$ hTERT-RPE1 FUCCI cells treated with the indicated concentrations of NCS for 1 h (mean ± SEM; $n = 3$ independent experiments). (C) Quantification of p53, p53 pSer15, and γ-H2A.X accumulation from (B) in WIP1$^{WT}$ and WIP1$^{\Delta CT}$ hTERT-RPE1 FUCCI cells (mean ± SD; $n = 3$ independent experiments). (D) Schematic detailing possible cell cycle fates of cells following DNA damage in G1 (left). Wild-type, p53$^{KO}$, and WIP1$^{\Delta CT}$ hTERT-RPE1 FUCCI cells were treated with either low or high dose NCS for 1 h, and the individual G1 cells were continuously imaged for 48 h. Cell cycle fate is plotted for individual cells. Pooled analyses are shown from three independent experiments (right). (E) The percentage of cells undergoing the G1/S transition or G1 arrest following the treatments described in (D) is plotted (mean ± SD; $n = 3$ independent experiments). (F) Schematic detailing possible cell cycle fates of cells after DNA damage in G2 (left). DNA damage was induced in WIP1$^{\Delta CT}$ hTERT-RPE1 FUCCI cells by a high dose of 100 ng/ml NCS (AS Hi) for 24 h. DMSO (AS No) was used as a negative control. Cells held in mitosis for 18 h and released into G1 for 24 h were used to test the outcome of mitotic timer pathway induced G1 arrest (M → G1). Cells trapped in G1 with the CDK4/6 inhibitor Palbociclib (CDK4/6i) for 24 h were used as a control for normal G1 arrest. The percentage of S/G2 and 2C/4C G1 cells is plotted (mean ± SEM, $n = 3$ independent experiments) (right). (G) Wild-type (WIP1$^{WT}$) (left) and WIP1$^{\Delta CT}$ (right) hTERT-RPE1 FUCCI cells were treated with the indicated doses of NCS for 1 h. NCS was removed by washing and cells were either fixed and stained immediately for the number of 53BP1 foci or after 4 h treatment with 25 ng/ml nocodazole to determine the percentage of mitotic entry (mean ± SD; $n = 3$ independent experiments). Statistical significance was analysed using an ordinary one-way ANOVA with Dunnett's multiple comparisons test. (H) WIP1$^{\Delta CT}$ hTERT-RPE1 FUCCI cells were treated with low or high dose NCS for 1 h. NCS was removed by washing and Late S/G2 cells were imaged continuously for 2 days. As a control, WIP1 inhibitor (WIP1i) was added at the same time as the high DNA damage treatment. Cell cycle fate is plotted for individual cells. Pooled analyses are shown from 3 independent experiments. (I) The percentage of mitotic entry and G2 to G1 cell-cycle collapse is plotted (mean ± SD; $n = 3$ independent experiments) from (H). Statistical significance was analysed using an ordinary one-way ANOVA with Tukey's multiple comparisons test. (J) G2 length in cells treated as in (H) is plotted for individual cells in a box and whiskers plot (median, 25th and 75th percentiles and whiskers extending to minimum and maximum values; mean (+) for the different conditions). Pooled analyses are shown from three independent experiments. Statistical significance was analysed using an ordinary one-way ANOVA with Tukey's multiple comparisons test. (K) G1 length following mitosis in cells that entered mitosis following treatments as in (H) is plotted for individual cells in a box and whiskers plot (median, 25th and 75th percentiles and whiskers extending to minimum and maximum values, mean (+) for the different conditions; pooled analyses shown from three independent experiments. n.d.: no data points could be obtained due to no cells entering mitosis. Statistical significance was analysed using an ordinary one-way ANOVA with Tukey's multiple comparisons test. Data for wild-type cells from Fig. 1 are included to aid comparisons in (I–K). Significance for all experiments: *$P < 0.05$; **$P < 0.01$; ***$P < 0.001$; ****$P < 0.0001$; ns not significant. All $P$ values are listed in Dataset EV1.

through into mitosis. A key hallmark of the G1 arrest is the ploidy of the arrested cells, which given their origin in G2 are predicted to have undergone a whole genome doubling and thus be tetraploid rather than diploid (Fig. 1H). In agreement with this idea, a p53- and p21-dependent increase in tetraploid cells was observed after a pulse of DNA damage in asynchronous hTERT-RPE1 cell cultures (Fig. 1H, p53$^{WT}$, p53$^{KO}$ and p21$^{KO}$ AS Hi). In contrast, delays in mitosis or treatment with a CDK4/6 inhibitor resulted in DNA damage-independent G1 arrest with only diploid cells (Fig. 1H, p53$^{WT}$, p53$^{KO}$ and p21$^{KO}$ CDK4/6i, and M → G1), as expected from previous work (Fry et al, 2004; Fulcher et al, 2025; Pennycook and Barr, 2021). Thus, engagement of the mitotic timer pathway largely preserves normal ploidy and does not trigger whole genome doubling, whereas G2 DNA damage can trigger penetrant bypass of mitosis and whole genome duplication.

## G2 DNA damage-dependent bypass of mitosis and G1 arrest are attenuated in WIP1$^{\Delta CT}$ cells

A crucial further difference between the mitotic timer pathway and G1 or G2 DNA damage checkpoints is the obvious requirement for DNA damage in the latter pathways (Fig. 1A). Thus, DNA damage signalling kinases and their counteracting phosphatase regulators should not be required for the DNA damage-independent mitotic timer pathway. Speaking against this simple interpretation, tumour cell lines such as HCT116 and U2OS which have gain-of-function mutations in the p53 phosphatase WIP1 are defective for the mitotic timer pathway (Meitinger et al, 2024). However, these tumour cell lines carry numerous mutations affecting cell proliferation and cell cycle control, and WIP1 mutation has therefore not been confirmed as the causal change leading to attenuation of the mitotic timer response. To address the function of WIP1 in the DNA damage signalling and mitotic timer pathways, human hTERT-RPE1 cells were gene edited to create

homozygous gain-of-function alleles in WIP1 (WIP1$^{\Delta CT}$) or to knock out WIP1 (WIP1$^{KO}$) (Figs. EV1A and EV2A,B). The effect of DNA damage on cell proliferation was reduced in these hTERT-RPE1 WIP1$^{\Delta CT}$ cells to a similar extent as p53$^{KO}$ cells, whereas WIP1$^{KO}$ cells were slightly more sensitive to DNA damage than the wild-type control (Figs. 1B and 2A). In agreement with previous studies on cancer cell lines with WIP1 mutations (Kahn et al, 2018; Kleiblova et al, 2013), WIP1$^{\Delta CT}$ has a longer half-life and is more abundant than the normal WIP1 protein (Fig. EV2C). Furthermore, WIP1$^{\Delta CT}$ cells showed attenuated accumulation of p53 pS15, γ-H2A.X pS139 and CHK2 pT68 phosphorylation after DNA damage, which declined further when ATM/ATR was inhibited (Figs. 2B,C and EV2D). These findings support the view that hTERT-RPE1 WIP1$^{\Delta CT}$ cells recapitulate the attenuated DNA damage signalling seen in cancer cells and thus provide a suitable model for the subsequent experiments.

To determine the consequences of WIP1 mutation for cell cycle progression after DNA damage in G1 or G2, and after mitotic delays, single cell imaging of FUCCI cell lines was performed. After DNA damage in G1, WIP1$^{\Delta CT}$ hTERT-RPE1 cells showed attenuated G1 cell cycle arrest (Fig. 2D,E, WT and WIP1$^{\Delta CT}$ high DNA damage), and behaved more like wild-type cells with low levels of DNA damage (Fig. 2D,E, WT low DNA damage and WIP1$^{\Delta CT}$ high DNA damage). Under the same conditions p53$^{KO}$ completely abrogated the G1 cell cycle arrest (Fig. 2D,E, WT and p53$^{KO}$ high DNA damage). Unlike wild-type cells, cultures of WIP1$^{\Delta CT}$ hTERT-RPE1 cells exposed to DNA damage in G2 did not accumulate tetraploid G1 cells (Fig. 2F, WIP1$^{\Delta CT}$ AS Hi). However, like wild-type cells, delays in mitosis or treatment with a CDK4/6 inhibitor resulted in penetrant G1 arrest with only diploid WIP1$^{\Delta CT}$ cells (Fig. 2F, WIP1$^{\Delta CT}$ CDK4/6i and M → G1), confirming that WIP1$^{\Delta CT}$ cells are still competent for G1 arrest. To understand the reduction in tetraploid cells after DNA damage, the G2 DNA damage threshold for mitotic entry was compared in

WIP1 wild-type and WIP1$^{\Delta CT}$ cells. Like wild-type cells, a reduction in the number of WIP1$^{\Delta CT}$ cells entering mitosis was seen with increasing levels of DNA damage (Fig. 2G, WIP1$^{WT}$ and WIP1$^{\Delta CT}$). However, the threshold for this response was moved to higher levels of DNA damage in WIP1$^{\Delta CT}$ cells compared to the wild-type cells. In the WIP1$^{\Delta CT}$ cells, 12.5 ng/ml NCS had little effect on mitotic entry, and significant reductions in mitotic entry were only seen above 25 ng/ml NCS (Fig. 2G). The FUCCI sensor was then used to track the cell cycle fate of hTERT-RPE1 WIP1$^{\Delta CT}$ cells after DNA damage. When exposed to low DNA damage, WIP1$^{\Delta CT}$ cells exhibited G2 length of 9.9 ± 3.7 h within the range of control cells in the absence of DNA damage and continued into the next cell cycle without extending G1 length or arresting in G1 (Figs. 2H–K and EV2E, WT and WIP1$^{\Delta CT}$ low DNA damage). In response to high levels of DNA damage, WIP1$^{\Delta CT}$ cells did not undergo cell cycle collapse from G2 directly to G1 and instead entered mitosis and divided (Figs. 2H,I and EV2E, WIP1$^{\Delta CT}$ high DNA damage). The ability of WIP1$^{\Delta CT}$ cells to enter mitosis in the presence of high DNA damage could be readily reversed by inhibiting WIP1 with a chemical inhibitor (Figs. 2H,I and EV2E, WIP1$^{\Delta CT}$ high DNA damage + WIP1i) (Gilmartin et al, 2014). This behaviour was thus similar to wild-type cells exposed to low DNA damage (Fig. 1D–H, WT low and 2H-2K, WIP1$^{\Delta CT}$ high DNA damage). One difference was that WIP1$^{\Delta CT}$ cells took ~50% longer to complete mitosis after high levels of DNA damage, with a mean time in mitosis of 91 ± 35 min. Although these cells showed extended G1 length of 17.9 ± 6.5 h the majority were able to enter S phase, suggesting that in addition to attenuation of the DNA damage response they bypass the mitotic timer response if DNA damage is present (Fig. 2H,K, WIP1$^{\Delta CT}$ high DNA damage). Thus, while wild-type cells do not enter mitosis following higher levels of G2 DNA damage and instead collapse to G1, WIP1$^{\Delta CT}$ mutations bypass this line of defence, enabling cells to complete mitosis and carry on into the next cell cycle.

## WIP1$^{\Delta CT}$ protects G2 cells against untimely mitotic cyclin destruction after DNA damage

We then sought to understand the mechanisms by which cells leave G2 to bypass mitosis and the mitotic timer after DNA damage, and the effect of WIP1 mutation on this response. Premature destruction of mitotic cyclins in G2 cells after DNA damage has been reported previously (Johmura et al, 2014; Krenning et al, 2014; Mullers et al, 2014). However, those reports did not explore the relationship with the mitotic timer or the role of WIP1 in this pathway. To address those questions, the levels of key cell cycle regulators and p53 pathway components in response to low and high levels of DNA damage in G2 were compared in wild-type hTERT-RPE1 and WIP1$^{\Delta CT}$ cells. Cells for biochemical analysis were first synchronised in G2 using a selective CDK1-inhibitory drug RO-3306 (Vassilev et al, 2006) and then treated with different concentrations of NCS for 8 h to generate low or high levels of DNA damage. In the absence of DNA damage or in cells with low DNA damage there was little change in the G2 cyclins A2 and B1, or the Wee1/Myt1-dependent pT14/pY15 inhibitory modifications of CDK1, showing that most cells remained trapped in G2 (Fig. 3A,B, WT no DNA damage (–) and LD). In wild-type cells, high DNA damage in G2 resulted in the destruction of cyclin A2 and B1, and loss of CDK1 pT14/Y15 signal (Fig. 3A,B, WT HD).

This was temporally coincident with stabilisation of p53, and accumulation of p53 target genes p21 and MDM2 (Fig. 3A, WT HD). Furthermore, Rb phosphorylation, a marker for progression through the restriction point (Narasimha et al, 2014; Yao et al, 2008), was lost in the high DNA damage condition yet remained stable after low DNA damage (Fig. 3A,C, WT HD and LD), in agreement with single cell imaging data showing cells leave G2 and directly move to a G1-like state only in high damage conditions (Figs. 1 and 2). Total levels of Rb were also slightly reduced following high DNA damage, consistent with a recent report demonstrating Rb destruction in early G1 (Zhang et al, 2024). In WIP1$^{\Delta CT}$ cells, DNA damage-dependent induction of p21 (Fig. 3D, WIP1$^{\Delta CT}$ HD and WT HD) and the reduction in cyclins A2 and B1 (Fig. 3A,B WIP1$^{\Delta CT}$ HD and WT HD) were all strongly attenuated compared to the wild-type cells, explaining why WIP1$^{\Delta CT}$ cells stay in a G2 state competent for mitosis. Knockout of p53 or the CDK inhibitor p21 abolished the downstream response leading to cyclin A2 and B1 destruction and loss Rb phosphorylation in response to high damage (Fig. 3A–C, p53$^{KO}$ and p21$^{KO}$ HD). In contrast, cyclin D, the G1 target for p21 (Harper et al, 1993), which is not a substrate for the anaphase promoting complex (APC/C) (Chai-kovsky et al, 2021; Maiani et al, 2021) was not lost under any of these conditions (Fig. 3A,C). Thus, G2 cyclin destruction in response to G2 DNA damage requires induction of the G1 and G2 CDK-inhibitor p21 triggered by DNA damage above a threshold level, which is set by WIP1 and attenuated by cancer-associated gain-of-function mutations that increase the level and therefore activity of WIP1 phosphatase.

## APC/C triggers cyclin destruction in G2 after DNA damage

It was important to understand the timing and mechanism of G2 cyclin destruction and whether this occurred before or after cells initiate entry into mitosis. If DNA damaged G2 cells transiently entered mitosis en route to G1, MDM2 levels would drop, stabilising p53, and possibly explain p21-induction through a combination of the DNA damage and mitotic timer pathways (Fulcher et al, 2025). Therefore, both the timing and kinetics of cyclin destruction were precisely followed by single cell imaging of hTERT-RPE1 CCNA2-GFP cells with wild-type WIP1, p53 and p21. In unperturbed normal cells, cyclin A2 accumulated during S and G2 phase, before it was abruptly destroyed during the G2-M transition prior to cell division (Fig. 4A,B, No damage), as previously reported (Di Fiore and Pines, 2010; Wolthuis et al, 2008). By contrast, in cells exposed to high DNA damage, cyclin A2 was destroyed before mitotic entry without nuclear envelope breakdown, and no cell division events were observed (Fig. 4A,B, High damage). It was notable that the rate of destruction was far slower in DNA damaged G2 cells than in normal cells entering mitosis, taking over 2 h rather than 10–15 min, respectively. This suggested that the APC/C, albeit active, had not achieved maximal activity under these conditions. Taken together with the biochemical data in Fig. 3, these single cell imaging experiments support the idea that cells exposed to DNA damage, exit G2 and return to a G1-like state due to untimely G2/M cyclin destruction before entering mitosis.

G2/M cyclin destruction normally occurs after entry into mitosis due to APC/C activation through two co-activator subunits Cdc20 and Cdh1 in mitosis and G1, respectively, each with different

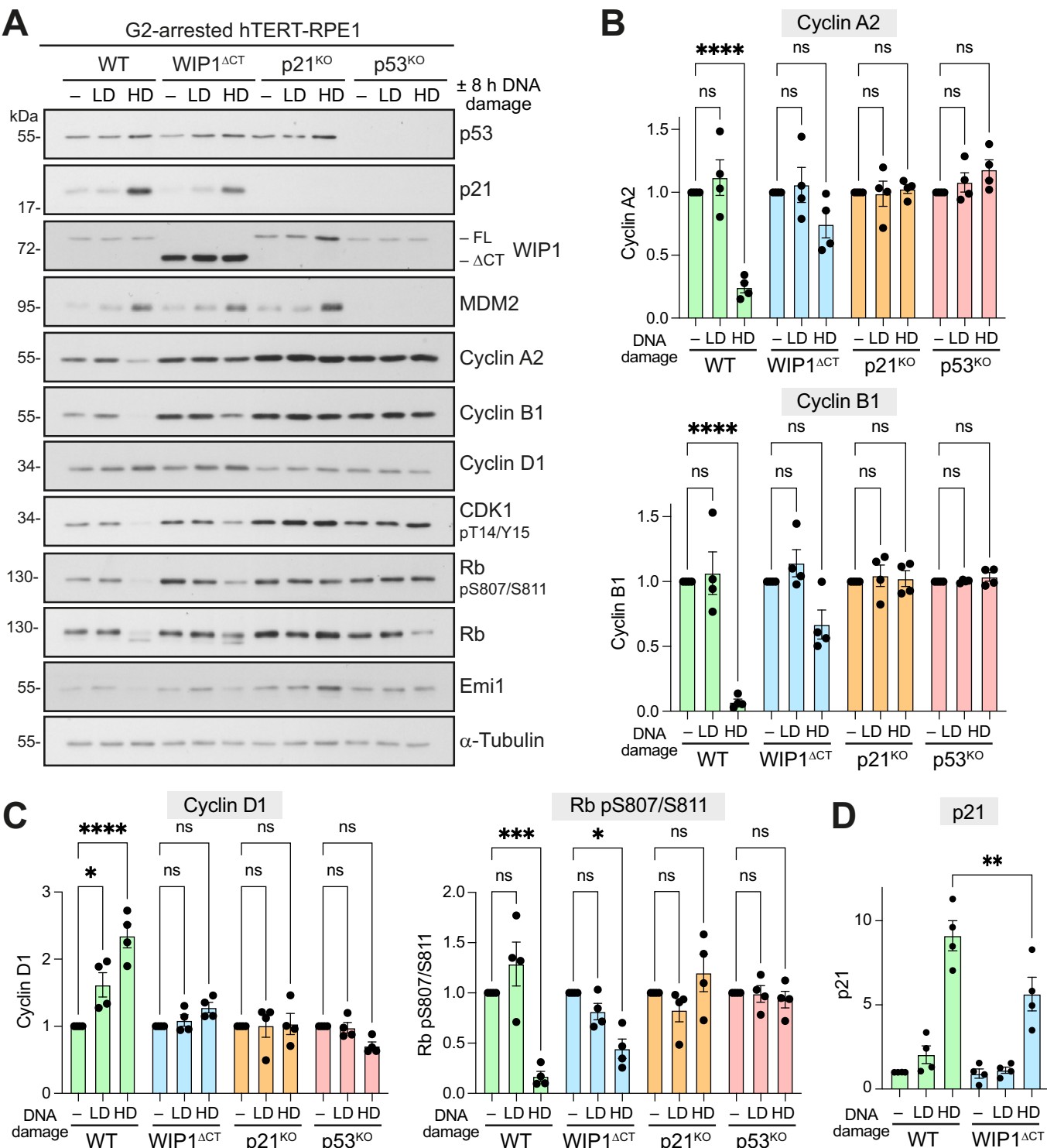

**Figure 3. DNA damage in G2 triggers p21 induction and premature G2/M cyclin destruction.**

(A) Wild-type (WT), WIP1^ΔCT, p21^KO, and p53^KO hTERT-RPE1 FUCCI cells synchronised in G2 with CDK1 inhibitor for 18 h were treated with either low dose (LD) or high dose (HD) NCS for 8 h to trigger DNA damage, then Western blotted for cell cycle and DNA damage markers (n = 4 independent experiments). (B–D) The levels of cyclin A2 and cyclin B2 (B), cyclin D1 and phosphorylated Rb pS807/S811 (C), or p21 (D) are plotted as mean ± SEM (n = 4 independent experiments) for cell lines treated as in (A). Statistical significance was analysed using an ordinary one-way ANOVA with Tukey's multiple comparisons test. Significance for all experiments: *P < 0.05; **P < 0.01; ***P < 0.001; ****P < 0.0001; ns not significant. All P values are listed in Dataset EV1.

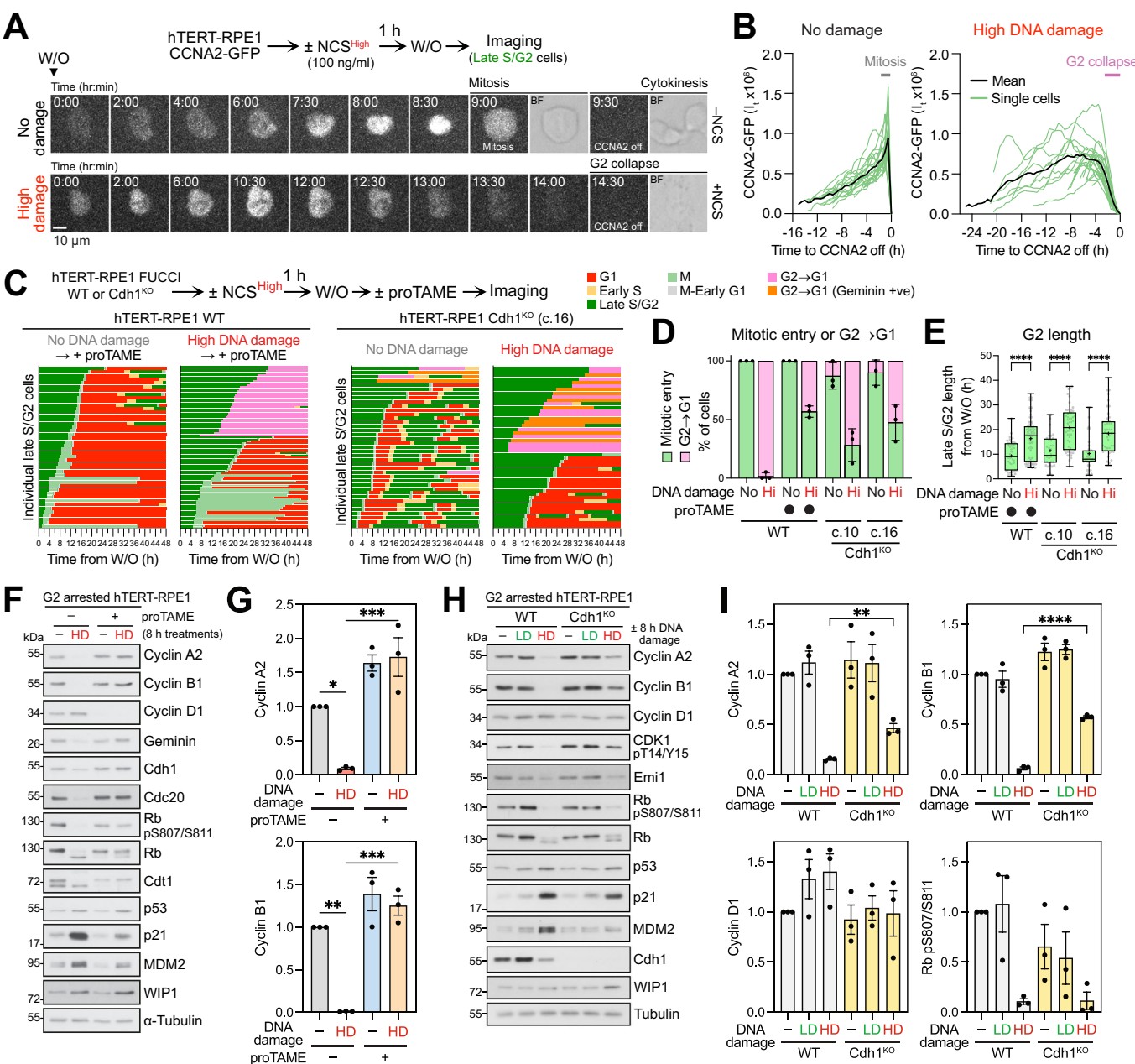

**Figure 4. APC/C-dependent G2/M cyclin destruction triggers G2-G1 collapse prior to mitotic entry.**

(A) Cyclin A2-GFP hTERT-RPE1 cells were imaged after a 1 h pulse of high dose NCS (High damage). Untreated (No damage) cells were imaged as a control ($n = 3$ independent experiments). Scale bar: 10 μm. (B) Cyclin A2-GFP fluorescence in individual cells from (A) is plotted during mitotic entry to mitotic exit (left, No damage) or during G2 to G1 cell-cycle collapse (right, High DNA damage). (C) Wild-type (WT) and Cdh1[KO] hTERT-RPE1 FUCCI cells were treated with or without high dose NCS for 1 h. WT cells were then treated with the APC/C inhibitor proTAME, while Cdh1[KO] cells were not. Cell cycle fate is plotted for individual cells (pooled analysis from three independent experiments). (D) Cell cycle fate of G2 cells treated as described in (C) along with additional control conditions including an alternative Cdh1[KO] clone were analysed. Mitotic entry and G2 to G1 collapse are plotted (mean ± SD; $n = 3$ independent experiments). (E) G2 length in cells treated as in (C) is plotted in a box and whiskers plot (median, 25th and 75th percentiles and whiskers extending to minimum and maximum values; mean (+) for the different conditions). Pooled analyses from three independent experiments. Statistical significance was analysed using an ordinary one-way ANOVA with Tukey's multiple comparisons test. (F) WT hTERT-RPE1 FUCCI cells synchronised in G2 with CDK1 inhibitor for 18 h were treated with high dose (HD) NCS for 8 h to trigger DNA damage, in the presence or absence of proTAME, then Western blotted for cell cycle markers ($n = 3$ independent experiments). (G) Quantification of cyclin A2 (top) and cyclin B1 (bottom) levels from (F) (mean ± SEM; $n = 3$ independent experiments). Statistical significance was analysed using an ordinary one-way ANOVA with Tukey's multiple comparisons test. (H) WT and Cdh1[KO] hTERT-RPE1 FUCCI cells synchronised in G2 by treatment with CDK1 inhibitor for 18 h were then treated with a pulse of either low dose (LD) or high dose (HD) NCS and samples collected 8 h later. Samples were Western blotted for cell cycle markers ($n = 3$ independent experiments). (I) Cyclin A2 (top left), cyclin B1 (top right), cyclin D1 (bottom left) or Rb phosphorylation pSer807/Ser811 (bottom right) levels after 8 h are plotted as mean ± SEM ($n = 3$ independent experiments) for cell lines treated as in (H). Statistical significance was analysed using an unpaired two-tailed $t$ test. Significance for all experiments: *$P < 0.05$; **$P < 0.01$; ***$P < 0.001$; ****$P < 0.0001$. All $P$ values are listed in Dataset EV1.

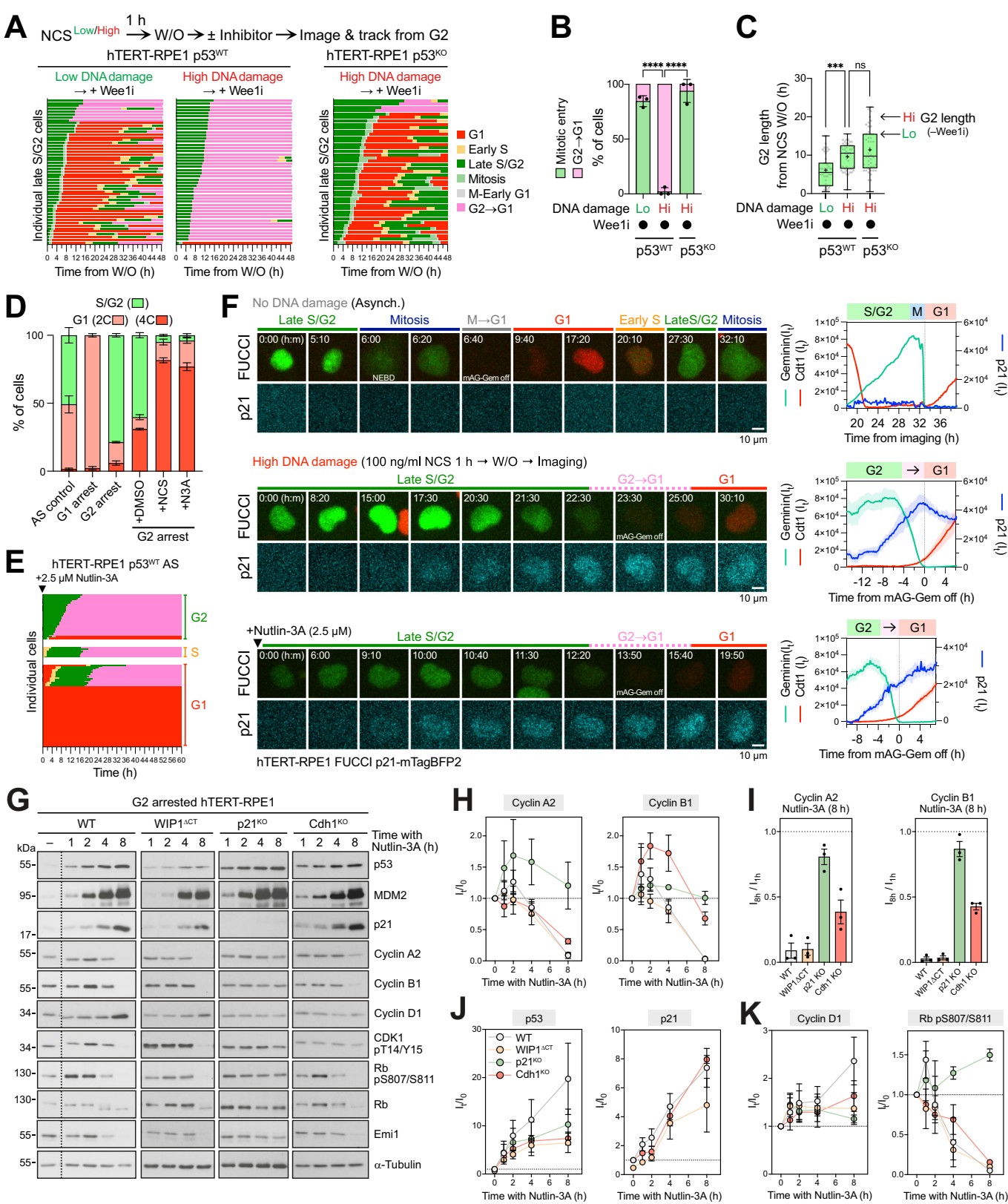

**Figure 5.　p53 stabilisation in the absence of DNA damage results in p21 induction, mitotic cyclin destruction and G1 arrest.**

(A) $p53^{WT}$ and $p53^{KO}$ hTERT-RPE1 FUCCI cells were treated with low or high dose NCS for 1 h. NCS was removed by washing and cells given Wee1 inhibitor (Wee1i). Late S/G2 cells were imaged continuously for 2 days. Cell cycle fate is plotted for individual cells. Pooled analyses are shown from three independent experiments. (B) The percentage of mitotic entry and G2 to G1 cell-cycle collapse is plotted from (A) (mean ± SD; $n = 3$ independent experiments). Statistical significance was analysed using an ordinary one-way ANOVA with Tukey's multiple comparisons test. (C) G2 length in cells treated as in (A) is plotted for individual cells in a box and whiskers plot (median, 25th and 75th percentiles and whiskers extending to minimum and maximum values; mean (+) for the different conditions). Arrows indicate the mean G2 length for $p53^{WT}$ cells treated with low or high dose NCS without Wee1 inhibitor. Pooled analyses are shown from three independent experiments. Statistical significance was analysed using an ordinary one-way ANOVA with Tukey's multiple comparisons test. (D) Asynchronous (AS) $p53^{WT}$ hTERT-RPE1 FUCCI cells were treated with DMSO (AS control), CDK4/6 inhibitor (G1 arrest) or CDK1 inhibitor (G2 arrest) for 24 h before fixation. $p53^{WT}$ cells arrested in G2 were treated with DMSO, high dose NCS or Nutlin-3A for a further 24 h (G2 arrest + DMSO/NCS/N3A) prior to fixation. The percentage of S/G2 and 2C/4C G1 cells is plotted (mean ± SEM, $n = 3$ independent experiments). (E) Asynchronous (AS) $p53^{WT}$ hTERT-RPE1 FUCCI cells were treated with Nutlin-3A and imaged continuously for 2.5 days. Cell cycle fate is plotted for individual cells, tracking from the phase they were in at the time of Nutlin-3A treatment. Pooled analyses are shown from three independent experiments. (F) p21-mTagBFP2 hTERT-RPE1 FUCCI cells were treated with high dose NCS for 1 h, then washed and imaged (High DNA damage), or treated with Nutlin-3A and imaged (+Nutlin-3A). Representative images are shown, and mean p21-BFP intensity along with FUCCI marker intensities are plotted (mean ± SEM; $n = 5$ cells). Asynchronous cells are included as a control (No DNA damage). Scale bars: 10 µm. (G) Wild-type (WT), $WIP1^{ACT}$, $p21^{KO}$ and $Cdh1^{KO}$ hTERT-RPE1 cells were arrested in G2 by treatment with CDK1i for 18 h, and the G2-arrested cells were treated with Nutlin-3A for the time points indicated. Samples were subjected to immunoblotting for cell cycle markers, using the indicated antibodies ($n = 3$ independent experiments). (H) The behaviour of cyclin A2 and cyclin B1 from cells treated as in (G) is plotted (mean ± SEM; $n = 3$ independent experiments). (I) The levels of cyclin A2 and cyclin B1 remaining after 8 h treatment in the different cell lines treated as in (G) are plotted (mean ± SEM; $n = 3$ independent experiments). (J, K) The behaviour of p53 and p21 (J), and cyclin D1 and Rb phosphorylation (Ser807/Ser811) (K) are plotted for each cell line treated as in (G) (mean ± SEM; $n = 3$ independent experiments). For all experiments, significance: ***$P < 0.001$; ****$P < 0.0001$; ns not significant. All $P$ values are listed in Dataset EV1.

regulatory properties (Sivakumar and Gorbsky, 2015). Cdc20 is essential for mitotic exit and is the target of the spindle assembly checkpoint, whereas Cdh1 although not essential is required for normal regulation of G1 and the transition into S phase (Musacchio, 2015; Pennycook and Barr, 2020). To test if collapse of G2 to G1 requires APC/C activity, wild-type hTERT-RPE1 FUCCI cells were treated with the APC/C inhibitor proTAME (Zeng et al, 2010). Consistent with the idea G2 cyclin destruction is an APC/C-dependent process, the addition of proTAME following high DNA damage reduced the percentage of cells moving directly from G2 into G1 (Fig. 4C,D; Appendix Figs S1A,B, proTAME ± DNA damage), compared to cells with high DNA damage in the absence of proTAME (Fig. 1D, $p53^{WT}$ High DNA damage). Because proTAME inhibits APC/C-Cdc20 and hence mitotic exit, mitosis was also prolonged in these cells (Fig. 4C; Appendix Fig. S1A,B, proTAME ± DNA damage). Under these conditions, only a subset of cells with normal length mitosis and no DNA damage entered S-phase (Fig. 4C, proTAME No DNA damage, yellow bars). Since most proTAME treated mitotic cells eventually entered G1 we concluded that proTAME only partially inhibits the APC/C under these conditions. Western blotting confirmed that cyclin A and cyclin B were stabilised in proTAME treated cells exposed to a pulse of high DNA damage compared to wild-type cells (Fig. 4F,G, WT ± proTAME HD). The samples for Western blotting were taken at 8 h, explaining the difference to single cell imaging which is on a longer time scale. Next, to test if premature activation of APC/C-Cdh1 in G2 explained cyclin destruction, $Cdh1^{KO}$ hTERT-RPE1 $p53^{WT}$ cells were used (Fig. EV1A). $Cdh1^{KO}$ reduced the percentage of $p53^{WT}$ cells moving directly from G2 into G1 following a pulse of DNA damage compared to untreated $Cdh1^{KO}$ cells with no damage (Fig. 4C,D; Appendix Fig. S1A,B, $Cdh1^{KO}$ ± DNA damage) or $Cdh1^{WT}$ $p53^{WT}$ cells with high damage (Fig. 1D, $p53^{WT}$ High DNA damage). Under all conditions in both Cdh1 wild-type and $Cdh1^{KO}$ cell lines, G2 length was increased in the presence of DNA damage (Fig. 4E), showing that DNA damage signalling was still able to delay CDK1-cyclin B activation and prevent entry into mitosis even in the absence of Cdh1. Western blotting confirmed that cyclin A and cyclin B were stabilised in G2

$Cdh1^{KO}$ cells exposed to a pulse of high DNA damage compared to wild-type cells (Fig. 4H,I, WT and $Cdh1^{KO}$ HD). However, this was a partial effect, supporting the idea that APC/C-dependent G2/M cyclin destruction requires both Cdc20 and Cdh1. Interestingly, after high DNA damage the APC/C inhibitor Emi1 is slowly lost in G2 cells concomitantly with the G2 cyclins (Fig. 4H). This is in line with the view that rather than the APC/C being prematurely activated in G2, it is the loss of APC/C inhibitory factors that triggers G2 cyclin destruction after DNA damage. Because $Cdh1^{KO}$ cells still stabilised p53 and robustly induced p21 resulting in a reduction of Rb phosphorylation (Fig. 4H,I), the simplest conclusion is that the APC/C acts downstream of p21 to divert G2 cells to G1, rather than at the level of p53 or p21 stability.

## Cell cycle collapse to a tetraploid G1 state requires p21 induction but not extended G2

DNA damage extends G2 by delaying CDK1-cyclin B activation through CDC25-mediated dephosphorylation and by inducing the broad spectrum CDK-inhibitor p21. It was therefore important to understand which of these pathways was required to divert G2 cells to G1. To investigate the importance of extended G2 length, Wee1 inhibitors were used to block the T14/Y15 inhibitory phosphorylation of the CDK1-cyclin B complex. Inhibition of Wee1 after DNA damage did not reduce the number of $p53^{WT}$ G2 cells bypassing mitosis and arresting in G1 (Fig. 5A,B, $p53^{WT}$ High damage + Wee1i, and Fig. 1D,E, $p53^{WT}$ High damage). Crucially, after DNA damage in $p53^{WT}$ cells, Wee1 inhibition shortened G2 length to 6.1 ± 3.9 h and 9.6 ± 3.4 h, in low and high DNA damage, respectively (Fig. 5C, $p53^{WT}$ + Wee1i). This compares to 15 h and 20 h for low and high DNA damage conditions, respectively, in the absence of Wee1 inhibitor (Fig. 1F), confirming effective Wee1 inhibition. Importantly, although mitotic bypass from G2 and subsequent G1 arrest were p53-dependent, G2 extension after DNA damage and Wee1-dependent shortening of G2 were unaltered by $p53^{KO}$ (Fig. 5A–C, $p53^{KO}$). Thus, the CDK1-cyclin B tyrosine phosphorylation pathway although crucial for extension of G2 length after DNA damage, is not essential for $p53^{WT}$ G2 cells to bypass mitosis and undergo G1 arrest. These findings exclude increased G2 length as the primary cause of the cell cycle collapse, and

suggest an alternative explanation nonetheless involving p21, which is necessary for the G1 arrest (Fig. 1D–H, p21$^{KO}$).

To test if p21-induction in G2 was sufficient to cause G2 cells to revert to G1 independent of induced DNA damage, the MDM2 inhibitor Nutlin-3A was used to stabilise p53 (Vassilev et al, 2004). In synchronised G2 cells, p21-induction by both DNA damage and Nutlin-3A caused highly efficient >80% collapse to a G1 tetraploid state within one cell cycle (Fig. 5D, G2 arrest + NCS and N3A). Under normal asynchronous culture (AS) few cells are tetraploid, and this is maintained when the cells were arrested in G1 or G2 without DNA damage (Fig. 5D, AS, G1 and G2 arrest). Time lapse imaging of hTERT-RPE1 FUCCI cells revealed different cell cycle fates depending on the cell cycle phase at the time of Nutlin-3A addition. Whereas early G1 cells treated with Nutlin-3A arrested directly in G1, G2 cells bypassed mitosis and arrested in G1 (Fig. 5E; Appendix Fig. S2A), cells in late G1 that are already committed to the next cell cycle, entered S phase, and proceeded to G2 before bypassing mitosis and arresting in G1 (Fig. 5E; Appendix Fig. S2A). Similarly, S phase cells entered G2 before bypassing mitosis and arresting in G1 (Fig. 5E; Appendix Fig. S2A). Importantly, mean G2 length of 8.5 ± 3.8 h was not altered by Nutlin-3A since it does not cause DNA damage. To understand how these outcomes relate to the timing of p21 induction, single cell imaging of hTERT-RPE1 FUCCI p21-BFP cells was used (Appendix Fig. S2B,C). This approach showed that after p53 stabilisation by Nutlin-3A addition or DNA damage, p21 rose in G2 prior to the loss of the FUCCI APC/C reporter geminin and accumulation of the G1 reporter Cdt1 (Fig. 5F, Nutlin-3A and High DNA damage). Compared to the rapid destruction of geminin within 30 min during normal mitotic progression (Fig. 5F, No DNA damage), geminin destruction was slower after DNA damage or Nutlin-3A treatment, taking over 4 h to complete (Fig. 5F, Nutlin-3A and High DNA damage). Taken together, the biochemical observations (Fig. 3A,B) and single cell imaging of slow geminin and cyclin A destruction (Figs. 4A,B and 5F) show that although the APC/C is active, it does not reach the high activity state associated with the transition from metaphase to anaphase when these same proteins are much more rapidly destroyed. In agreement with the idea that WIP1 mutations will have little consequence in the absence of DNA damage, western blot analysis of G2 arrested cells treated with Nutlin-3A revealed equivalent loss of cyclin A and B in both wild-type and WIP1$^{ΔCT}$ cells (Fig. 5G–I, WT and WIP1$^{ΔCT}$). This was accompanied by p53-stabilisation and p21-induction and reduction in Emi1 and Rb phosphorylation, whereas cyclin D remained stable consistent with the idea cells are in G1 (Fig. 5G,J,K, WT and WIP1$^{ΔCT}$). By contrast, knockout of p21 or Cdh1 abolished or attenuated this response, respectively, stabilising cyclin A and B (Fig. 5G–K, p21$^{KO}$ and Cdh1$^{KO}$). One key difference is that Rb phosphorylation was reduced in Cdh1$^{KO}$ but not p21$^{KO}$ (Fig. 5G,K, p21$^{KO}$ and Cdh1$^{KO}$), in line with the idea that a p21-inhibited CDK is responsible for this modification. These data show that G2 to G1 collapse is driven by p21-dependent inhibition of CDK activity and APC/C-Cdc20 and -Cdh1 dependent destruction of G2 cyclins, triggered by p53 stabilisation and p21 induction after DNA damage.

## Inhibition of CDK2 in G2 cells triggers cyclin destruction, bypassing mitosis

Taken together, the results presented thus far suggest that following G2 DNA damage above a threshold level, p53-dependent p21-induction inhibits CDK activity and directly collapses G2 back to a G1-like state avoiding mitosis, rather than the alternative possibility that cells enter but fail to complete mitosis. The different roles of CDKs in sustaining the G2 state and promoting mitosis are crucial for differentiating these possibilities. Entry into mitosis is triggered by activation of CDK1-cyclin B, whereas maintenance of G2 is mediated by CDK2-cyclin A (Hochegger et al, 2008) supporting the idea that p21 is likely to target CDK2-cyclin A during G2 collapse. Western blot analysis of p21-bound CDK-cyclin complexes isolated from cells showed that only cyclins D and E are bound to p21 in G1, whereas in G2 both cyclin A and B are present (Fig. EV3A, G1 and G2). Pulldowns from asynchronous cell cultures revealed that all tested CDKs and cyclins were present (Fig. EV3A, AS). CDK1-cyclin B appeared to be a poorer binder of p21 than CDK2-cyclin A since it was only enriched if p21 was induced with Nutlin-3A (Fig. EV3A, G2 ± Nutlin-3A). Because CDK2 appears to bind p21 preferentially and the biochemical assays are carried out in G2 cells synchronised by CDK1 inhibition, this strongly favours the possibility that G2 collapse is due to inhibition of CDK2 by p21. In agreement with that idea, addition of two structurally unrelated CDK2-selective inhibitors to G2 arrested cells triggered destruction of cyclins A and B, similar to the effect of Nutlin-3A addition or a pan-CDK inhibitor (Figs. 6A,B and EV3B,C). Efficient cyclin A and B destruction was observed in both wild-type and WIP1$^{ΔCT}$ cells (Fig. 6C,D, WT and WIP1$^{ΔCT}$), in agreement with the idea that WIP1 mutation is of little consequence in the absence of DNA damage. Knockout of p53 or p21 abolished the effect of Nutlin-3A which requires p21, but not of the CDK2 inhibitory drug, showing that CDK2 inhibition is both necessary and sufficient to cause cyclin destruction in G2 cells (Fig. 6C,D, p53$^{KO}$ and p21$^{KO}$). By contrast, inhibition of CDK4/6 in G2 cells results in G1 arrest after successful cell division but does not trigger bypass of mitosis (Fig. EV3D). However, CDK2 inhibition alone does not explain why cells remain arrested in G1 after collapse from G2. Single cell imaging revealed that although CDK2 inhibition in wild-type cells triggered penetrant collapse to G1 from G2, these cells then entered S phase and did not remain arrested in G1 (Fig. 6E–G, WT). Consistent with this interpretation, after CDK2 inhibition we observed re-accumulation of the DNA licensing factor Cdt1 (Figs. 6A,C and EV3B), a marker for late G1 (Nishitani et al, 2001). Knockout of p53 or p21 did not alter this behaviour in response to CDK2 inhibition (Fig. 6E–G, p53$^{KO}$ and p21$^{KO}$), although there were differences in G1 length between p53$^{KO}$ and p21$^{KO}$ cell lines that cannot be readily explained at this time. Because a key property of p21 is its ability to bind and inhibit multiple CDK complexes, including CDK4/6 with G1 cyclins (Harper et al, 1993), the most parsimonious explanation is that p21 triggers bypass of mitosis by inhibition of CDK2, leading to APC/C-dependent destruction of cyclin A and B, and that p21 then maintains G1 arrest due to inhibition of CDK4/6.

## WIP1 inhibition or knockout do not rescue the mitotic timer pathway in tumour cell lines

WIP1$^{ΔCT}$ gain-of-function mutations attenuate DNA damage signalling and p53-stabilisation, reducing p21-dependent inhibition of CDKs and enabling cell cycle progression in the presence of DNA damage (Figs. 2–6). However, they have no obvious effect on

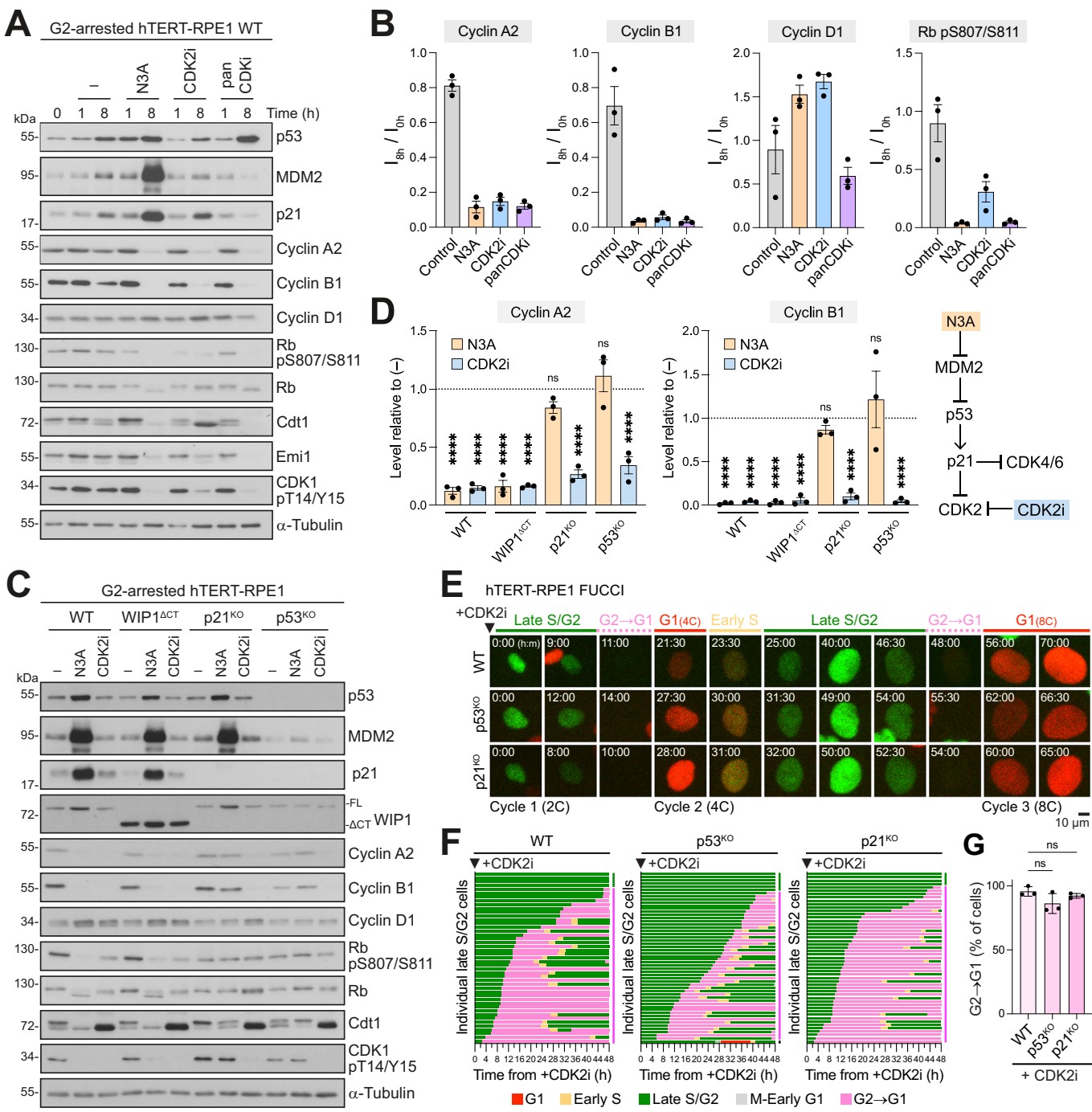

**Figure 6. CDK2 inhibition in G2 cells bypasses mitosis and triggers G1 arrest.**

(A) Wild-type hTERT-RPE1 FUCCI cells were synchronised in G2 with a CDK1 inhibitor for 18 h. G2-arrested cells were treated with Nutlin 3 A (N3A), CDK2 inhibitor (CDK2i), a pan-CDK inhibitor (pan CDKi) or DMSO (–) for 1 or 8 h and then Western blotted ($n = 3$ independent experiments). (B) Cyclins A2, B1, D1 and phospho-Rb (Ser807/Ser811) levels after 8 h treatment as in (A) are plotted (mean ± SEM; $n = 3$ independent experiments). (C) Wild-type, WIP1$^{ΔCT}$, p21$^{KO}$, and p53$^{KO}$ hTERT-RPE1 FUCCI cells were synchronised in G2 with CDK1 inhibitor for 18 h, treated with N3A, CDK2i or DMSO (–) for 8 h, and Western blotted for cell cycle markers. (D) Cyclin A2 and B1 levels after 8 h treatment as in (C) are plotted (mean ± SEM; $n = 3$ independent experiments) (left). A schematic detailing the targets of the drugs used in the experiment (right). Statistical significance was analysed using an ordinary one-way ANOVA with Tukey's multiple comparisons test where each condition is compared to DMSO control. (E) Wild-type, p53$^{KO}$, and p21$^{KO}$ hTERT-RPE1 FUCCI cells were treated with CDK2i, and Late S/G2 cells were imaged. Representative images are shown ($n = 3$ independent experiments). Scale bar: 10 μm. (F) Cell cycle fates are plotted for the cells in (E). Pooled analyses are shown from three independent experiments. (G) The percentage of G2 to G1 collapse is plotted for the cells described in (E) (mean ± SD; $n = 3$ independent experiments). Statistical significance was analysed using an ordinary one-way ANOVA with Tukey's multiple comparisons test. For all experiments, significance: ****$P < 0.0001$; ns not significant. All $P$ values are listed in Dataset EV1.

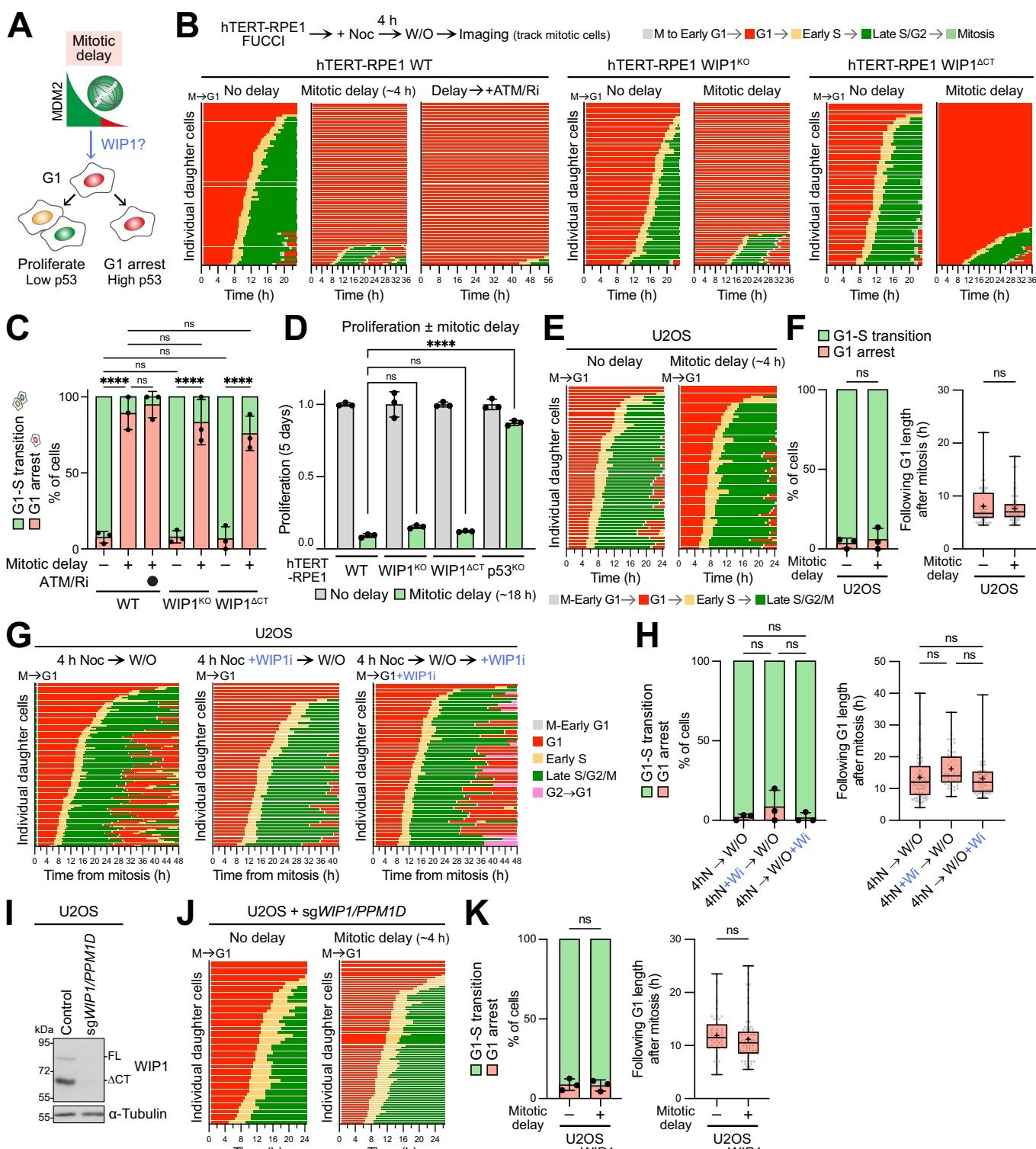

cell cycle progression in the absence of DNA damage, in agreement with other published work (Burocziova et al, 2019; Kleiblova et al, 2013; Stoyanov et al, 2024). This creates the possibility that unlike wild-type cells, WIP1$^{\Delta CT}$ cells can activate the mitotic timer after G2 DNA damage, which would normally induce p21 and collapse G2 to G1 bypassing mitosis. Wild-type, WIP1$^{KO}$ and WIP1$^{\Delta CT}$ cells

were therefore compared in terms of their response to delays in mitosis in the absence of DNA damage (Fig. 7A). In response to a 4 h delay in mitosis both hTERT-RPE1 wild-type, WIP1$^{KO}$ and WIP1$^{\Delta CT}$ cells showed G1 arrest (Fig. 7B,C, Mitotic delay). In agreement with the idea the mitotic timer is normally a DNA damage-independent pathway, ATM/ATR inhibition did not

**Figure 7. WIP1 does not play a role in the mitotic timer response in the absence of DNA damage.**

(A) Schematic detailing the mitotic timer pathway. How WIP1 affects this intrinsic timing mechanism is not known. (B) Wild-type (WIP1^WT) hTERT-RPE1 FUCCI cells (left) were treated with nocodazole to induce mitotic delay (4 h) or left untreated (No delay). Nocodazole was removed by washing, and the cells then imaged continuously in the presence of absence of ATM/ATR inhibitors (ATM/Ri) (left). WIP1^KO (middle) and WIP1^ΔCT (right) hTERT-RPE1 FUCCI cells were treated in parallel. Cell cycle fate is plotted for individual cells. Pooled analyses are shown from three independent experiments. (C) The percentage of cells undergoing the G1/S transition or G1 arrest after mitotic delay or no delay described in (B) plus an ATM/ATR inhibited condition (ATM/Ri) is plotted (mean ± SD; $n = 3$ independent experiments). Statistical significance was analysed using an ordinary one-way ANOVA with Tukey's multiple comparisons test. (D) Wild-type (WIP1^WT), WIP1^KO, WIP1^ΔCT and p53^KO hTERT-RPE1 cells were arrested in mitosis for 18 h using nocodazole. The mitotic cells were isolated by shake-off, washed out from nocodazole, and re-plated for 5 days to assess proliferation. Asynchronous cells were seeded in parallel as controls. Relative proliferation for each cell line is plotted (mean ± SD; $n = 3$ independent experiments). Statistical significance was analysed using an ordinary one-way ANOVA with Tukey's multiple comparisons test. (E) U2OS FUCCI cells were treated with nocodazole to induce mitotic delay (4 h) or left untreated (No delay). Nocodazole was removed by washing, and the cells then imaged continuously for 24 h. Cell cycle fate is plotted for individual cells. Pooled analyses are shown from three independent experiments. (F) The percentage of cells undergoing the G1/S transition or G1 arrest after mitotic delay or no delay described in (E) is plotted (left) (mean ± SD; $n = 3$ independent experiments). Statistical significance was analysed using an unpaired two-tailed $t$ test. G1 length following mitosis in cells that entered mitosis following treatments as in (E) is plotted for individual cells in a box and whiskers plot (median, 25th and 75th percentiles and whiskers extending to minimum and maximum values, mean (+) for the different conditions; pooled analyses shown from three independent experiments) (right). Statistical significance was analysed using an unpaired two-tailed $t$ test. (G) U2OS FUCCI cells were treated with nocodazole to induce mitotic delay (4 h) in the presence or absence of WIP1 inhibitor (WIP1i). Nocodazole/WIP1i was removed by washing, and the cells then imaged continuously for 48 h with or without WIP1i addition. (H) The percentage of cells undergoing the G1/S transition or G1 arrest for mitotic delay or no delay described in (G) is plotted (left) (mean ± SD; $n = 3$ independent experiments). Statistical significance was analysed using an ordinary one-way ANOVA with Tukey's multiple comparisons test. G1 length following mitosis in cells that entered mitosis following treatments as in (G) is plotted for individual cells in a box and whiskers plot (median, 25th and 75th percentiles and whiskers extending to minimum and maximum values, mean (+) for the different conditions; pooled analyses shown from three independent experiments) (right). Statistical significance was analysed using an ordinary one-way ANOVA with Tukey's multiple comparisons test. (I) Western blot showing efficient WIP1 knockout in U2OS FUCCI cells targeted with WIP1 single guide RNA (sgWIP1/PPM1D) ($n = 1$ experiment). (J) U2OS FUCCI cells described in (I) were treated with nocodazole to induce mitotic delay (4 h) or left untreated (No delay). Nocodazole was removed by washing, and the cells then imaged continuously for 24 h. Cell cycle fate is plotted for individual cells. Pooled analyses are shown from three independent experiments. (K) The percentage of cells undergoing the G1/S transition or G1 arrest after mitotic delay or no delay described in (J) is plotted (left) (mean ± SD; $n = 3$ independent experiments). Statistical significance was analysed using an unpaired two-tailed $t$ test. G1 length following mitosis in cells that entered mitosis following treatments as in (J) is plotted for individual cells in a box and whiskers plot (median, 25th and 75th percentiles and whiskers extending to minimum and maximum values, mean (+) for the different conditions; pooled analyses shown from three independent experiments) (right). Statistical significance was analysed using an unpaired two-tailed $t$ test. Significance for all experiments: ****$P < 0.0001$; ns not significant. All $P$ values are listed in Dataset EV1.

overcome G1 arrest following a 4 h delay in mitosis in wild-type cells (Fig. 7B,C, Mitotic delay + ATM/Ri). This was accompanied by an equally penetrant loss of proliferation in WIP1 wild-type WIP1^KO and WIP1^ΔCT cells after mitotic delays (Fig. 7D). Previous work has suggested that WIP1 mutations may explain the loss of the mitotic timer response in some tumour cell lines including U2OS and HCT116 which express both WIP1^WT and the WIP1^ΔCT mutant form and have higher overall levels of WIP1 (Fig. EV4A–C). In addition, these cells carry numerous other known oncogenic mutations summarised in Fig. EV4D. Compared to diploid hTERT-RPE1 cells, U2OS cells, despite being hypertriploid and spending slightly longer time in mitosis, do not arrest in G1 either under unperturbed growth conditions when mitosis exceeds 60 min (Fig. EV4E,F), or when mitosis is delayed for 4 h using a microtubule poison (Fig. 7E,F). In contrast to a previous report (Meitinger et al, 2024), addition of a WIP1 inhibitor did not increase the number of G1 cells arrested after 4 h delays in mitosis (Fig. 7G,H). To eliminate the possibility that the inhibitor was ineffective under these conditions, WIP1 was knocked out to eliminate the wild-type and mutant alleles (Fig. 7I). These U2OS WIP1^KO cells behaved like the parental cell lines and again showed limited G1 arrest after 4 h delays in mitosis (Fig. 7J,K). To confirm these findings, these experiments were repeated in HCT116 wild-type (WIP1^WT/ΔCT) and HCT116 WIP1^KO cells with equivalent results (Fig. EV4G–K). Highly aneuploid HeLa cells have greatly extended mitosis and wild-type WIP1, but still fail to show a mitotic timer response (Fig. EV4E–H), most likely due to the suppression of p53 activity by the human papillomavirus E6/E7 proteins (Scheffner et al, 1993). These findings provide compelling evidence that WIP1 gain-of-function mutations are unlikely to be the causal change explaining the loss of the mitotic timer pathway

in U2OS or HCT116 cancer cell lines. How then can the observation that WIP1 mutation appears to suppress G1 cell cycle arrest after mitotic delays be explained?

## The effect of WIP1^ΔCT on the mitotic timer response is conditional on DNA damage

WIP1 plays a crucial role in setting the threshold for DNA damage signalling in both G1 and G2 cells (Fig. 8A). As already established, unlike wild-type cells which delay in G2 or undergo G2 to G1 collapse, WIP1^ΔCT cells fail to maintain G2 arrest and enter mitosis. Building on those observations, it was therefore possible to test if DNA damage alters the threshold for the mitotic timer response. To do this, WIP1^WT, WIP1^KO and WIP1^ΔCT hTERT-RPE1 cells in asynchronous culture were tracked as they passed through the cell cycle and the stochastic variations in the length of mitosis and outcomes in terms of G1 cell cycle fate were recorded. This approach was previously used to establish that there is a sharp threshold time in mitosis, after which normal cells arrest in G1 of the next cell cycle in the absence of detectable DNA damage (Fulcher et al, 2025). First the length of mitosis was analysed in the presence or absence of low levels of DNA damage in the preceding G2. In all cases, mitosis was ~50 min with no significant differences between WIP1^WT, WIP1^KO and WIP1^ΔCT hTERT-RPE1 cell lines (Fig. 8B). G1 cell cycle arrest was observed for cells spending longer than 60 min in mitosis irrespective of the state of WIP1 in the presence and absence of low levels of DNA damage (Fig. 8C). Thus, the sharp threshold for the mitotic timer response was maintained despite the differences in WIP1 status. It is important to note that WIP1^ΔCT cells like p53^KO cells are also defective for cell cycle arrest following DNA damage in G1 (Fig. 2D,E),

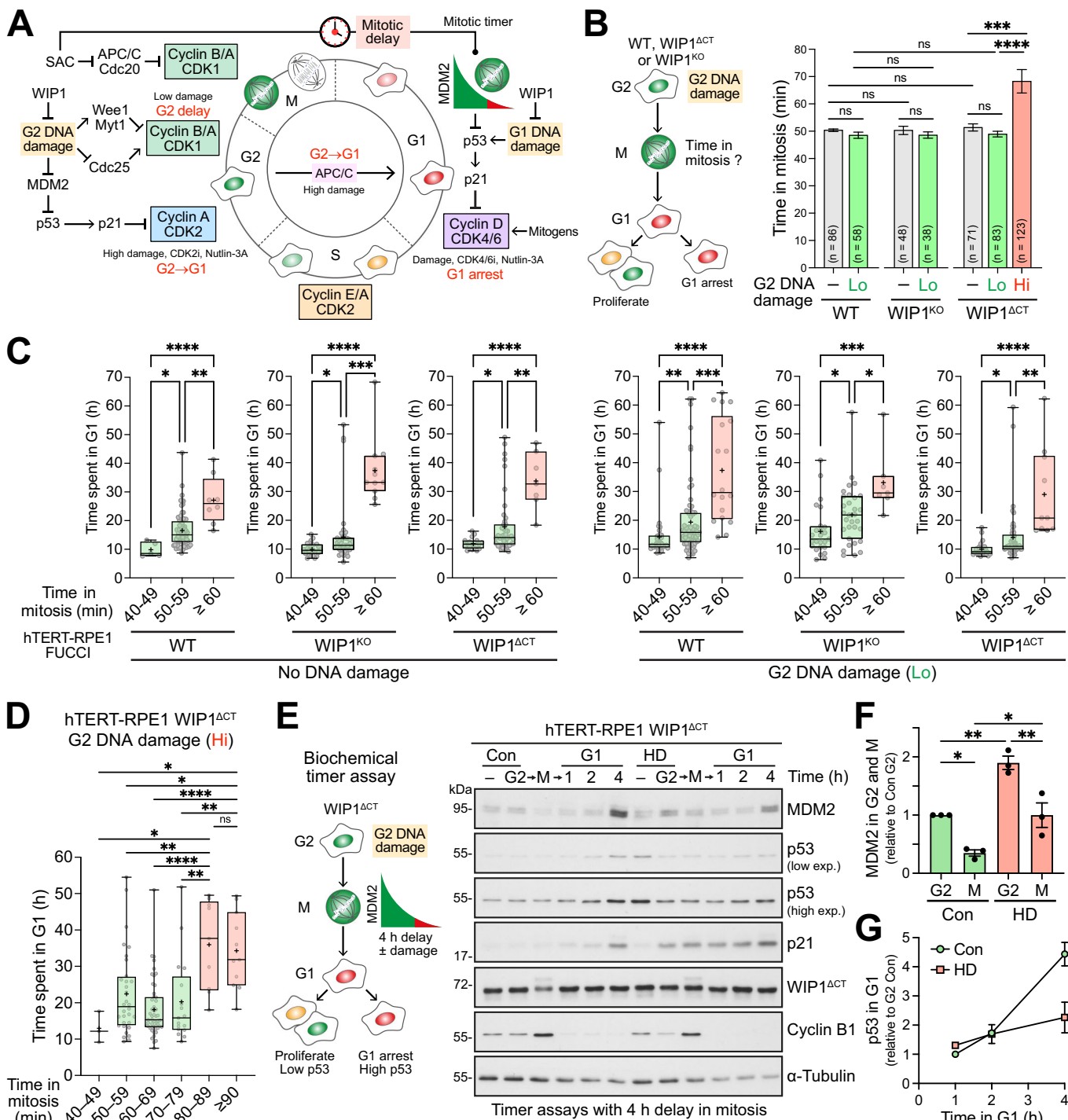

demonstrating that WIP1$^{\Delta CT}$ can act in both G1 and G2. Taken together these results show WIP1$^{\Delta CT}$ cells show an attenuated response to DNA damage in G1 and G2, and yet still undergo G1 cell cycle arrest in response to delays in mitosis either in the absence of DNA damage or following low levels of DNA damage (Fig. 8C).

A key difference between wild-type and WIP1$^{\Delta CT}$ cells is the response to higher levels of G2 DNA damage at which wild-type

cells cannot enter mitosis and undergo cell-cycle collapse from G2 to a tetraploid G1 like state. As already established, WIP1$^{\Delta CT}$ cells do enter mitosis after induction of higher levels of DNA damage, and although showing an extended G1, do not show a penetrant G1 arrest (Fig. 2G–K). Interestingly, the length of mitosis in these cells was prolonged at ~70 min, above the threshold time which triggers G1 cell cycle arrest in wild-type cells (Fig. 8B,C) (Fulcher et al, 2025). This suggested that the threshold time in mitosis that

◄ **Figure 8. The effect of WIP1^ΔCT on the mitotic timer response is conditional on DNA damage.**

(A) Overview of pathways linking DNA damage-dependent and independent checkpoints to p53 function and cell cycle control. (B) Schematic showing the potential G1 cell cycle outcomes for wild-type and WIP1^ΔCT cells undergoing mitotic delays after DNA damage in G2 (left). Time in mitosis in G2 for wild-type, WIP1^KO and WIP1^ΔCT hTERT-RPE1 FUCCI cells treated with or without a prior 1 h low dose pulse of NCS to create DNA damage is plotted (mean ± SD; pooled analyses shown from 3 independent experiments) (right). Statistical significance was analysed using a one-way ANOVA with Tukey's multiple comparisons test. (C) G1 length for wild-type (WIP1^WT), WIP1^KO and WIP1^ΔCT hTERT-RPE1 FUCCI cells treated as in (B) are plotted in a box and whiskers plot, categorised based on their time in mitosis (median, 25th and 75th percentiles and whiskers extending to minimum and maximum values; mean (+) for the different conditions; pooled analyses shown from 3 independent experiments). Statistical significance was analysed using a Kruskal–Wallis test with Dunn's multiple comparisons test. (D) WIP1^ΔCT hTERT-RPE1 FUCCI cells were treated with a pulse of HD NCS for 1 h, and cells were tracked from G2, through mitosis into G1 for up to 60 h. G1 length is plotted in a box and whiskers plot, categorised based on their time in mitosis (median, 25th and 75th percentiles and whiskers extending to minimum and maximum values; mean (+) for the different conditions; pooled analyses shown from 3 independent experiments). Statistical significance was analysed using an ordinary one-way ANOVA with Tukey's multiple comparisons test. (E) Schematic depicting the predicted effects of increased MDM2 levels following G2 DNA damage on the mitotic timer pathway in WIP1^ΔCT cells (left). WIP1^ΔCT hTERT-RPE1 cells were treated with a pulse of 100 ng/ml NCS (high dose, HD) or DMSO (con) for 1 h and allowed to recover for 16 h (G2). Nocodazole (25 ng/ml) was then added for 1 h and mitotic cells were isolated by shake-off. Isolated mitotic cells were kept arrested in mitosis for a further 3 h, before nocodazole was removed by washing and a mitotic sample (M) was taken. The remaining cells were re-plated and allowed to enter G1. G1 samples were taken at 1, 2 and 4 h post mitosis. All samples were Western blotted (right). (F) MDM2 and (G) p53 levels are shown for the conditions indicated (mean ± SEM; n = 3 independent experiments). Statistical significance was analysed using an ordinary one-way ANOVA with Tukey's multiple comparisons test. Significance for all experiments: *P < 0.05; **P < 0.01; ***P < 0.001; ****P < 0.0001; ns not significant. All P values are listed in Dataset EV1.

triggers the mitotic timer was increased in these cells. This idea was confirmed using single cell imaging which revealed the crucial time in mitosis threshold was indeed increased to >80 min in WIP1^ΔCT cells exposed to high levels of DNA damage in G2 (Fig. 8D). One simple explanation for this outcome relates to the concentration of MDM2 at the point of mitotic entry in WIP1^ΔCT cells exposed to high levels of DNA damage in G2. Indeed, as MDM2 is a transcriptional target of p53, MDM2 concentration, like p21, rises during the G2 DNA damage response (Fig. 3A). While DNA damaged wild-type cells do not enter mitosis and instead collapse to G1 without cell division, WIP1^ΔCT cells enter mitosis with a higher MDM2 concentration and it is likely that this elevated MDM2 concentration suppresses the timer response, as it would take longer for MDM2 to fall below the critical threshold limiting p53 activation in the G1 daughter cells. To test this idea, mitotic timer assays were performed using WIP1^ΔCT cells held for 4 h in mitosis, with or without prior exposure to a pulse of high DNA damage in the preceding G2 phase (Fig. 8E). Western blotting showed that in both the absence and presence of DNA damage, WIP1^ΔCT cells entered mitosis and destroyed cyclin B upon entry into G1 (Fig. 8E). In the absence of DNA damage MDM2 was consumed during the prolonged mitosis (Fig. 8E,F), leading to p53 stabilisation in the first 4 h of G1 (Fig. 8E,G), and an increase in p21 in later G1 (Fig. 8E). MDM2, like p21, is a p53 target gene and thus reaccumulates later in G1 starting at 4 h (Fig. 8E). In contrast, following the pulse of high DNA damage prior to mitosis, MDM2 level rose initially in G2, before being maintained at a higher level following the 4 h mitosis compared to in the absence of DNA damage (Fig. 8E,F). In turn, p53 although initially increased in G2 compared to the absence of DNA damage, did not increase further in G1 after a 4 h delay in mitosis (Fig. 8E,G). Likewise, the target genes MDM2 and p21 did not increase to the higher level seen for a 4 h delay in mitosis in the absence of DNA damage (Fig. 8E). In conclusion, the effects of WIP1 mutation on the mitotic timer pathway are conditional on the presence of DNA damage above a threshold level. In the absence of DNA damage or following exposure to lower levels of DNA damage in the preceding G2, the mitotic timer pathway triggers the expected cell cycle arrest in G1 of the following cell cycle.

## Discussion

How the multiple cell cycle checkpoint pathways are coordinated to channel damaged or stressed cells into cell cycle arrest or senescence is a crucial question in cell and cancer biology. Cells entering G2 with DNA damage or under-replicated DNA halt to allow repair of the damage and then select one of two possible fates. If damage is repaired cells will continue to mitosis, whereas irreparable or prolonged damage results induces in p53-dependent p21-induction and collapse of G2 to a tetraploid G1 senescence-like state (Johmura et al, 2014; Krenning et al, 2014). Once cells have entered mitosis, delay-induced loss of MDM2, the p53 ubiquitin ligase, can promote a DNA damage-independent but p53- and p21-dependent G1-arrest with the crucial difference that cells are typically diploid rather than tetraploid. Here, we have explored whether DNA damage in G2 could alter p53 and MDM2 concentrations and hence the time threshold for the mitotic timer or create a memory of damage passed through mitosis that could influence the outcome of a mitotic delay. While lower levels of DNA damage result in delays in G2, they did not alter either the length of the ensuing mitosis or the abrupt time threshold for the mitotic timer response. Higher levels of DNA damage in G2 resulted in a p53- and p21-dependent bypass of mitosis and the mitotic timer mechanism, directly channelling damaged cells from G2 into a tetraploid G1 state. Cell cycle collapse from G2 to G1 is driven by p21-dependent inhibition of CDK2-cyclin complexes, leading to precocious activation of the APC/C and destruction of G2/M cyclins A and B (Fig. 8A). In G1, p21 enforces the arrest and prevents S phase entry by inhibiting CDK4/6-cyclin D complexes. Other recent work suggests that CDK4/6 also play a role in sustaining the G2 state (Cornwell et al, 2023), so although we have not tested it here, p21 may also act on CDK4/6-cyclin D complexes in G2 cells. Because CDK2 inhibition alone can trigger G2/M cyclin destruction and collapse of G2 cells to a G1-like state with high levels of cyclin D, we propose that under DNA damage conditions CDK2 is the major relevant target for p21.

Remarkably, we also find that hTERT-RPE1 cells engineered to mimic the effects of cancer-associated mutations in the p53-phosphatase WIP1 attenuate DNA damage signalling and circumvent this response, enabling damaged cells to enter mitosis, divide, and enter the next cell cycle. Crucially, WIP1 mutations do not affect the mitotic timer response in the absence of DNA damage or under low DNA damage conditions,

providing the first selective separation of function mutation for the p53-dependent mitotic timer and DNA damage responses. We conclude that although the mitotic timer and DNA damage pathways share many common components, normal cells can independently initiate and regulate p53-dependent cell cycle exit to generate either diploid or tetraploid arrest states. These conclusions are in close agreement with previous reports showing that the G1 and G2 DNA damage checkpoints and downstream p53-dependent responses are attenuated in cells carrying WIP1 gain-of-function mutations (Burocziova et al, 2019; Kleiblova et al, 2013; Stoyanov et al, 2024). Our work also supports the view that the primary function of WIP1 is to suppresses ATM/ATR-dependent cellular responses to exogenous genotoxic stress, whereas it does not regulate p53 activation by the mitotic timer response, since this is independent of DNA damage signalling. Interestingly, WIP1 gain-of-function mutations are reported to be enriched in patients with haematological cancers following emergence of resistance to chemotherapy, and it therefore provides a promising target for new approaches to help eliminate these cells (Burocziova et al, 2023; Kahn et al, 2018; Miller et al, 2023). As we show, WIP1 inhibitors re-sensitise cells to DNA damage and thereby enable mitotic bypass after genotoxic stress in G2 (Fig. 2H–K). Therefore, although WIP1 plays no direct role in the mitotic timer pathway, it is a crucial factor in setting the DNA damage threshold determining the fate of cells in G2, and whether they can enter mitosis or collapse into a G1-like arrested state.

Intriguingly, both the G2 DNA damage and mitotic timer pathways are modulated by PLK1 activity and have been linked to WIP1 (Burigotto et al, 2023; Jaiswal et al, 2017; Meitinger et al, 2024; van Vugt et al, 2004). After G2 DNA damage, cell cycle progression is transiently halted by ATM and ATR-dependent inhibition of key mitotic kinases such as PLK1 (van Vugt et al, 2004). In turn, PLK1 and WIP1 counteract ATR-dependent signalling and promote resumption of the cell cycle (Jaiswal et al, 2017). One simple possibility is that WIP1 modulates PLK1-regulation of USP28/53BP1 and thereby promotes p53-dependent cell cycle arrest (Burigotto et al, 2023; Meitinger et al, 2024). Whether such regulation occurs or is truly DNA damage-independent remains unclear. Although previous work has suggested a correlation between loss of the timer pathway and WIP1 mutation in aneuploid p53 wild-type tumour cell lines (Meitinger et al, 2024), these cells often have high continued levels of either replication stress or DNA damage (Chunduri and Storchova, 2019; Zhu et al, 2018). It is therefore difficult to eliminate the possibility that the correlation is due to DNA damage which may be present, albeit at low levels. Here, we find that knockout or inhibition of WIP1 in cells lines in which the mitotic timer pathway is inactivated does not rescue the p53-dependent cell cycle arrest after mitotic delays. However, although we find that WIP1 plays no direct role in the timer response, tumour-associated WIP1 mutations alter the threshold for G2 DNA damage with profound impact on the mitotic timer pathway. At higher levels of DNA damage, the WIP1-truncating mutations allow cells with damaged DNA to enter mitosis and bypass the mitotic timer response (Fig. 2). We propose this is due to altered levels of MDM2 at the onset of mitosis in those cells (Fig. 8). Thus, WIP1 mutation will only suppress the mitotic timer response in the presence of higher levels of DNA damage. Our findings may thus explain the relationship between WIP1 and the mitotic timer, and explain how, by attenuating DNA damage signalling, WIP1 mutations enable aneuploid cells, cells with replication stress, or cells with DNA damage repair deficiencies to bypass multiple checkpoint mechanisms and continue proliferating despite normal p53 function.

# Methods

## Reagents and tools table

| Reagent/resource | Reference or source | Identifier or catalog number |
| --- | --- | --- |
| **Experimental models** | | |
| hTERT-RPE1 wild-type (WT) (*H. sapiens*) | ATCC | Cat: #CRL-4000 |
| hTERT-RPE1 WT FUCCI (*H. sapiens*) | Fulcher et al, 2025 | |
| hTERT-RPE1 p53$^{KO}$ FUCCI (*H. sapiens*) | Fulcher et al, 2025 | |
| hTERT-RPE1 p21$^{KO}$ FUCCI (*H. sapiens*) | This paper | |
| hTERT-RPE1 Cdh1$^{KO}$ FUCCI (*H. sapiens*) | This paper | |
| hTERT-RPE1 WIP1$^{ΔCT}$ FUCCI (*H. sapiens*) | This paper | |
| hTERT-RPE1 WIP1$^{KO}$ FUCCI (*H. sapiens*) | This paper | |
| hTERT-RPE1 WT FUCCI p21-mTagBFP2 (*H. sapiens*) | This paper | |
| hTERT-RPE1 Cyclin A2-GFP (*H. sapiens*) | Alfonso-Pérez et al, 2019 | |
| hTERT-RPE1 p21-GFP and mRuby-PCNA (*H. sapiens*) | Barr et al, 2017 | |
| HCT116 (*H. sapiens*) | ATCC | Cat: #CCL-247 |
| HCT116 FUCCI (*H. sapiens*) | This paper | |
| HCT116 FUCCI WIP1$^{KO}$ (polyclonal) (*H. sapiens*) | This paper | |
| U2OS (*H. sapiens*) | ATCC | Cat: #HTB-96 |
| U2OS FUCCI (*H. sapiens*) | This paper | |
| U2OS FUCCI WIP1$^{KO}$ (polyclonal) (*H. sapiens*) | This paper | |
| HeLa (*H. sapiens*) | ATCC | Cat: #CRL-2.2 |
| HeLa FUCCI (*H. sapiens*) | This paper | |
| HEK293T (*H. sapiens*) | ATCC | Cat: #CRL-3216 |
| XL-1 blue (*E. coli*) | Agilent | Cat: #200249 |
| **Recombinant DNA** | | |
| pBOB-EF1-FastFUCCI-Puro | Addgene | Cat: #86849 |
| pMD2.G | Addgene | Cat: #12259 |
| psPAX2 | Addgene | Cat: #12260 |
| LentiCRISPR v2-Blast | Addgene | Cat: #83480 |
| pX459 | Addgene | Cat: #62988 |
| **Antibodies** | | |
| p53 Mouse mAb (WB 1:1000) | Cell Signaling Technology | Cat: #48818 |
| p21 Rabbit mAb (WB 1:1000) | Cell Signaling Technology | Cat: #2947 |
| MDM2 Mouse mAb (WB 1:100) | Millipore | Cat: #OP46 |
| MDM2 Rabbit mAb (WB 1:1000) | Cell Signaling Technology | Cat: #86934 |
| WIP1 Mouse mAb (WB 1:200) | Santa Cruz | Cat: #sc-376257 |
| Phospho-p53 (Ser15) Rabbit pAb (WB 1:1000) | Cell Signaling Technology | Cat: #9284 |

| Reagent/resource | Reference or source | Identifier or catalog number |
|---|---|---|
| γ-H2A.X Mouse mAb (WB 1:1000; IF 1:1000) | Biolegend | Cat: #613402 |
| 53BP1 Rabbit pAb (IF 1:1000) | Novus Bio | Cat: #NB100-304 |
| α-Tubulin Mouse mAb (WB 1:10,000) | Sigma-Aldrich | Cat: #T6199 |
| Cyclin A2 Rabbit mAb (WB 1:10,000) | Abcam | Cat: #ab32386 |
| Cyclin B1 Mouse mAb (WB 1:5000) | Millipore | Cat: #05-373 |
| Cyclin D1 Rabbit mAb (WB 1:10,000) | Abcam | Cat: #ab134175 |
| Cyclin E1 Rabbit mAb (WB 1:1000) | Abcam | Cat: #ab33911 |
| Rb Mouse mAb (WB 1:1000) | Cell Signaling Technology | Cat: #9309 |
| Phospho-Rb (Ser807/811) Rabbit pAb (WB 1:1000) | Cell Signaling Technology | Cat: #9308 |
| Phospho-CDK1 (Thr14/Tyr15) Rabbit pAb (WB 1:5000) | Invitrogen | Cat: #44686G |
| CDK1 Rabbit mAb (WB 1:1000) | Abcam | Cat: #ab133327 |
| Emi1 Mouse mAb (WB 1:1000) | Invitrogen | Cat: #37-6600 |
| Cdt1 Rabbit mAb (WB 1:1000) | Cell Signaling Technology | Cat: #8064 |
| Cdh1 Mouse mAb (WB 1:1000) | Santa Cruz | Cat: #sc-56312 |
| Cdc20 Rabbit pAb (WB 1:2000) | Proteintech | Cat: #10252-1-AP |
| Geminin Mouse mAb (WB 1:1000) | Santa Cruz | Cat: #sc-74496 |
| Chk2 Rabbit pAb (WB 1:1000) | Cell Signaling Technology | Cat: #2662 |
| Phospho-Chk2 (Thr68) Rabbit pAb (WB 1:1000) | Cell Signaling Technology | Cat: #2661 |
| CDK2 Rabbit mAb (WB 1:1000) | Cell Signaling Technology | Cat: #18048 |
| p27 Rabbit mAb (WB 1:1000) | Cell Signaling Technology | Cat: #3686 |
| GFP Mouse mAb (WB 1:500) | ChromoTek | Cat: #gfms |
| Donkey anti-Mouse-HRP | Jackson | Cat: #715-035-150 |
| Donkey anti-Rabbit-HRP | Jackson | Cat: #711-035-152 |
| Donkey anti-Rabbit AlexaFluor647 | Thermo Fisher | Cat: #A31573 |
| Donkey anti-Mouse AlexaFluor647 | Thermo Fisher | Cat: #A31571 |
| **Oligonucleotides and other sequence-based reagents** | | |
| p21 KO gRNA Forward (5′-caccggatgtccgtcagaacccatg -3′) | Thermo Fisher | |
| p21 KO gRNA Reverse (5′-aaaccatgggttctgacggacatcc-3′) | Thermo Fisher | |
| WIP1ΔCT gRNA Forward (5′-caccgatagctcgagagaatgtcca-3′) | Thermo Fisher | |
| WIP1ΔCT gRNA Reverse (5′-aaactggacattctctcgagctatc-3′) | Thermo Fisher | |
| WIP1 KO gRNA1 Forward (5′-caccgcacctcgtccgagccggcta-3′) | Thermo Fisher | |
| WIP1 KO gRNA1 Reverse (5′-aaactagccggctcggacgaggtgc-3′) | Thermo Fisher | |

| Reagent/resource | Reference or source | Identifier or catalog number |
|---|---|---|
| WIP1 KO gRNA2 Forward (5′-caccgtgagcgtcttctccgacca -3′) | Thermo Fisher | |
| WIP1 KO gRNA2 Reverse (5′-aaactggtcggagaagacgctcac -3′) | Thermo Fisher | |
| Cdh1 KO gRNA Forward (5′-caccgtggggagctggcaggcgtca-3′) | Thermo Fisher | |
| Cdh1 KO gRNA Reverse (5′-aaactgacgcctgccagctccccac-3′) | Thermo Fisher | |
| p21-BFP KI gRNA Forward (5′-caccgggaagccctaatccgcccac-3′) | Thermo Fisher | |
| p21-BFP KI gRNA Reverse (5′-aaacgtgggcggattagggcttccc-3′) | Thermo Fisher | |
| Cyclin A2-GFP KI gRNA Forward (5′-caccgtacagatttagtgtctctgg-3′) | Thermo Fisher | |
| Cyclin A2-GFP KI gRNA Reverse (5′-aaacccagagagacactaaatctgtac-3′) | Thermo Fisher | |
| **Chemicals, enzymes and other reagents** | | |
| Neocarzinostatin | Sigma-Aldrich | Cat: #N9162 |
| Nutlin-3A | Selleck Chem | Cat: #S8059 |
| RO-3306 | TOCRIS | Cat: #4181 |
| Nocodazole | Sigma-Aldrich | Cat: #487928 |
| Palbociclib (PD 0332991 isethionate; CDK4/6i) | TOCRIS | Cat: #4786 |
| proTAME | TOCRIS | Cat: #7734 |
| Cycloheximide | Sigma-Aldrich | Cat: #C7698 |
| MK-1775 (Wee1i) | MedChemExpress | Cat: #HY-10993 |
| KU-55933 (ATMi) | Selleck Chem | Cat: #S1092 |
| AZD6738 (ATRi) | Selleck Chem | Cat: #S7693 |
| MK8776 (Chk1i) | Selleck Chem | Cat: #S2735 |
| PV1019 (Chk2i) | Sigma-Aldrich | Cat: #220488 |
| PF-06873600 (CDK2i) | Cambridge Bioscience | Cat: #S8816 |
| INX-315 (CDK2i-2) | MedChemExpress | Cat: #HY-162001 |
| Flavopiridol (pan CDKi) | TOCRIS | Cat: #3094 |
| GSK2830371 (WIP1i) | Cambridge Bioscience | Cat: #CAY16973 |
| GFP TRAP agarose | ChromoTek | Cat: #gta |
| Puromycin | InvivoGen | Cat: # ant-pr-1 |
| Blasticidin | InvivoGen | Cat: #ant-bl-05 |
| **Software** | | |
| Fiji (ImageJ) v.2.14.0 | NIH | |
| GraphPad Prism v.10.3.0 | GraphPad Software | |
| Volocity v.6.3.0 | PerkinElmer | |
| Image Lab v.6 | Bio-Rad | |
| MetaMorph v.7.5 | Molecular Devices | |
| **Other** | | |
| N/A | | |

## Stock cell lines and cell culture

All cell lines are validated stocks purchased from the ATCC. The hTERT-RPE1 p53$^{WT}$ and p53$^{KO}$ FUCCI cell lines and their generation have been described previously (Fulcher et al, 2025). All hTERT-RPE1 cell lines (#CRL-4000) were cultured in DMEM-F12 Ham medium (Sigma #D6421), supplemented with GlutaMAX™ (Gibco #35050087) and 10% (v/v) fetal bovine serum (FBS, Sigma-Aldrich #F9665). HeLa (#CRL-2.2), U2OS (#HTB-96), and HEK293T (#CRL-3216) cells were cultured in DMEM containing GlutaMAX™ (Gibco #10569010) and 10% (v/v) FBS. HCT116 cells (#CCL-247) were cultured in McCoy's 5A medium (Gibco #16600082), supplemented with 10 mM sodium pyruvate (Thermo Fisher #11360088) and 10% (v/v) FBS. For routine passaging, cells were washed in PBS and then incubated with TrypLE™ Express Enzyme cell dissociation reagent (Gibco #12605036) for 5 min at 37 °C, before resuspending detached cells in full medium for passage. All cell lines were maintained at 37 °C, humidified 5% CO$_2$ in a cell culture incubator (Thermo, HERAcell, #51013568). Mycoplasma negative status of cell lines was confirmed using the EZ-PCR Mycoplasma Test Kit with internal control (K1-0210, Geneflow). The cell lines used in our studies are not on the list as commonly misidentified lines. All parental cell line stocks were authenticated by STR profiling (Addgene or NorthGene).

## Generation of FUCCI cell lines

FUCCI reporter constructs were introduced into HeLa (#CRL-2.2), HCT116 (#CCL-247), and U2OS (#HTB-96) cells by lentiviral infection. Briefly, pBOB-EF1-FastFUCCI-Puro was a gift from Kevin Brindle & Duncan Jodrell (Addgene plasmid # 86849), pMD2.G (Addgene #12259) and psPAX2 (Addgene #12260) (gifts from Didier Trono) were used for lentiviral-based packaging in HEK293T cells for 2 days with a media change after 24 h. The resulting lentiviral supernatant was used to infect HeLa, HCT116 or U2OS cells for 24 h. After infection, antibiotic-resistant clones were selected with 0.2 µg/ml puromycin and then expanded in non-selective medium to be screened by fluorescence microscopy for successful integration of the FUCCI sensor.

## Generation of CRISPR cell lines

To generate hTERT-RPE1 FUCCI p21-mTagBFP2 cells, a guide RNA targeting the C-terminus of p21 inserted into a pX459 vector was delivered together with a donor homology template encoding for 20aa linker-mTagBFP2-P2A-BSD, flanked by 500 bp homology arms either side, by transient transfection. To generate hTERT-RPE1 Cyclin A2-GFP cells, a guide RNA targeting the C-terminus of Cyclin A2 inserted into a pX459 vector was delivered together with a donor homology template encoding for 20aa linker-GFP-P2A-BSD, flanked by 500 bp homology arms either side, by transient transfection (Alfonso-Perez et al, 2019). For transfection, transfection mixes were prepared in sterile DNase-free Eppendorf tubes (Eppendorf #0030108035) consisting of 100 µl Opti-MEM (Gibco #11058021), 3 µl Mirus TransIT-LT1 transfection reagent (Mirus #MIR2300), and up to 1 µg of plasmid DNA. Transfection mixes were vortexed briefly for 20 s, and incubated at room temperature for 30 min, before adding dropwise to cells. Lentiviral infection of LentiCRISPR v2-Blast vectors containing the appropriate guide RNA was used to generate the WIP1$^{ΔCT}$, WIP1$^{KO}$, p21$^{KO}$ and Cdh1$^{KO}$ hTERT-RPE1 FUCCI cell lines. In all cases, after brief

selection in blasticidin, monoclonal lines were screened by extensive quality controls including immunoblotting and amplification of the targeted genomic region by PCR. Genomic DNA was isolated from clones using QuickExtract™ DNA isolation solution (Lucigen #QE09050). PCRs were performed with KOD hot-start polymerase (Merck #71086), using primers flanking the gRNA recognition site. PCR products were ligated into pSC-A vectors using a Blunt-end PCR cloning kit (Agilent #240207). Ligated plasmids were transformed into XL-1 blue competent cells, and around ten colonies were mini-prepped (Qiagen #27107) and sequenced using Sanger DNA sequencing (Source Genomics, Cambridge, UK) to cover all edited alleles. For WIP1$^{KO}$ in U2OS and HCT116 FUCCI cell lines, following antibiotic selection after lentiviral delivery of the WIP1-targeting gRNA1 and gRNA2, WIP1 knockout was confirmed by western blotting.

## Cell synchronisation

To arrest cells in G2, cells were seeded at 50,000 cells per well of a 6-well plate and cultured for 24 h at 37 °C before addition of 6 µM RO-3306 for 18 h. To arrest cells in mitosis, 70% confluent cells were treated with 25 ng/ml nocodazole for 18 h at 37 °C. For immunoblot analysis, mitotic cells were isolated by shake-off prior to lysis. To arrest cells in G1, cells were seeded at 50,000 cells per well of a six-well plate and cultured for 24 h at 37 °C before addition of 1 µM Palbociclib for 18 h.

## Cell proliferation assays

To test for proliferation, hTERT-RPE1 FUCCI cells were treated as described in figure legends and cultured at 37 °C for 5 days. Following incubation, cells were washed in PBS and stained in 5% (w/v) crystal violet (Sigma-Aldrich #C0775) in (25% (v/v) methanol (VWR #20847.307) for 30 min at room temperature. Cells were washed in water and left to dry before imaging on a Bio-Rad Gel Doc XR+ imaging system (Bio-Rad, 1708195EDU) using Image Lab 6 software (Bio-Rad). The extent of cell proliferation was assessed by densitometry using Fiji/ImageJ software (National Institutes of Health). Note, hTERT-RPE1 cells are highly mobile and do not form distinct colonies.

## Immunoblotting

To prepare cell lysates, cells treated as described in the figure legends were washed in PBS and lysed in directly in 1× sample buffer (0.1875 mM Tris-HCl (Sigma-Aldrich, #T4661) pH 6.8, 1% (w/v) SDS (Sigma-Aldrich, #75746), 10% (v/v) glycerol (Sigma-Aldrich, #G9012), 0.05% (w/v) bromophenol blue (Sigma-Aldrich #B5525), and 10% (v/v) β-mercaptoethanol (Sigma-Aldrich, #M3148) in water). Lysates were boiled at 95 °C with intermittent vortexing to help shear the DNA. Proteins were separated by SDS-PAGE (Bio-Rad, Mini-PROTEAN® Tetra cell, #1658000) and transferred onto nitrocellulose membranes (Bio-Rad #1704158, #1704159) using a Trans-blot Turbo system (Bio-Rad, #1704150). After transfer, blots were blocked in 5% (w/v) milk powder in PBS (PanReac AppliChem, #A0965,9100)-TWEEN® 20 (Sigma-Aldrich #P1379) (PBS-T), before incubation with primary antibodies diluted in 5% (w/v) milk in PBS-T for 1 h at room temperature, or at 4 °C overnight. Blots were then washed in PBS-T before incubating with the secondary HRP-conjugated antibodies for 1 h at room temperature, before washing again in PBS-T. Signals were

revealed using ECL (GE Healthcare #RPN2106) and visualised on X-ray films (GE Healthcare #GE28-9068-37), developed on an OPTIMAX 2010 X-ray Film Processor (PROTEC-Med).

## Immunoprecipitation

Pellets of $1 \times 10^6$ asynchronous or synchronised hTERT-RPE1 p21-GFP cells (Barr et al, 2017) were resuspended in 1 ml of lysis buffer (20 mM Tris-HCl (Sigma-Aldrich, #T4661) pH 7.4, 150 mM NaCl (Sigma-Aldrich #S3014), 1% (v/v) IGEPAL (Sigma-Aldrich, #S3014), 0.1% (w/v) sodium deoxycholate (Sigma-Aldrich, #D6750), 40 mM sodium β-glycerophosphate (Sigma-Aldrich, #G9422), 10 mM NaF (Sigma-Aldrich, #201154), 0.3 mM sodium orthovanadate (Sigma-Aldrich, #S6508), 100 nM okadaic acid (Enzo Life Sciences #ALX-350-011-M001), 200 nM microcystin-LR (Enzo Life Sciences #ALX-350-012-M001), 1 mM DTT (Sigma-Aldrich #11583786001), protease inhibitor cocktail (Roche, cOmplete¯, EDTA-free Protease Inhibitor Cocktail, #11873580001), and 1× phosphatase inhibitor cocktail (Sigma, P0044) in water) on ice for 30 min, and the lysates were then clarified at 14,000 rpm for 30 min at 4 °C. A 5% (v/v) aliquot of cell lysate was mixed with sample buffer as input for immunoblotting. The proteins of interest were isolated from the remaining cell lysate by immunoprecipitation with GFP-TRAP agarose beads (ChromoTek, #gta) at 4 °C for 2 h. The beads were then washed three times with lysis buffer at 4 °C, resuspended in sample buffer and boiled at 95 °C to elute. IP and input samples were analysed by immunoblotting as described above.

## Immunofluorescence microscopy

Cells plated on glass coverslips were washed in PBS, and fixed with 3% (w/v) PFA in PBS pH 7.4 at room temperature for 15 min. Unreacted PFA was quenched by a 10-min incubation in 50 mM $NH_4Cl$ (Sigma-Aldrich #213330) in PBS. Cells were then washed three times in PBS, followed by permeabilization with 0.2% (v/v) Triton X-100 (Sigma-Aldrich #X100) in PBS at room temperature for 5–10 min. Permeabilised cells were washed three times in PBS before incubating with the appropriate primary antibodies. Antibody dilutions were performed in PBS, and coverslips were placed cell-side down on droplets of antibody solutions, inside a humidified chamber for 1 h at room temperature. Following incubation, cells were washed three times in PBS and then incubated with the secondary AlexaFluor-conjugated antibodies for 1 h at room temperature. To stain DNA, DAPI (Sigma-Aldrich, #D9542) was added at the same time as the secondary antibodies. After incubation, cells were washed three times in PBS and once in water, before mounting on glass microscopy slides in Mowiol® 4-88 (Sigma-Aldrich, #81381). Samples were imaged on an upright microscope (BX61, Olympus) with filter sets for DAPI, GFP/AlexaFluor 488, Cy3/AlexaFluor 555 and Cy5/AlexaFluor 647 (Chroma Technology Corp.), a 2048 × 2048-pixel complementary metal oxide semiconductor camera (Prime, Photometrics), and MetaMorph 7.5 imaging software (Molecular Devices). Illumination was provided by an LED light source (pE300, CoolLED Illumination Systems). Image stacks with a spacing of 0.4 μm through the cell volume were maximum intensity projected and cropped in ImageJ (National Institutes of Health). Imaging data were quantified using Fiji (ImageJ) (v.2.14.0).

## Live cell imaging

For single cell tracking, cells were plated in 35-mm dishes with a 14-mm 1.5 thickness cover glass window on the bottom (MatTek Corp, P35G-1.5-

14-C). For imaging, the dishes were placed in a 37 °C and 5% $CO_2$ environment chamber (Tokai Hit) on the microscope stage. Imaging was performed on an Ultraview Vox spinning disk confocal system running Volocity software (PerkinElmer) using the 20×/0.75 NA UPlanSApo objective on an Olympus IX81 inverted microscope equipped with an electron multiplying charge coupled device (EM-CCD) camera (C9100-13, Hamamatsu Photonics). FUCCI probes were imaged with 488 nm and 561 nm lasers using 50–200 ms exposures at 2.5–4% laser power. Cyclin A2-GFP was imaged with a 488 nm laser using 200 ms exposures at 7% laser power. p21-mTagBFP2 was imaged with a 405 nm laser using 300 ms exposures at 10% laser power. Brightfield reference images were also taken to visualize cell shape with 50 ms exposures. Image stacks, 24 planes at 0.9 μm spacing were collected at the time intervals indicated in the figures and then maximum intensity projected and cropped in Fiji/ImageJ (National Institutes of Health) for further analysis. Before the imaging, cells were treated as described in the figure legends.

## Quantification and statistical analysis

Statistical analysis was performed using GraphPad Prism v.10.3.0. Statistical significance was analysed using an unpaired two-tailed *t* test, a one-way ANOVA, a Kruskal–Wallis test, or Dunn's multiple comparison test. A Dunnett's or Tukey's multiple comparisons tests was performed following the ANOVA analyses. Graphs display the mean ± SD or SEM as indicated in the figure legends. Where statistical tests were performed, *P* values are shown on graphs as follows: $P \geq 0.05$ = not significant (ns), $*P < 0.05$, $**P < 0.01$, $***P < 0.001$, $****P < 0.0001$. No statistical methods were used to predetermine the experimental sample sizes. The sample size was chosen to balance replication and efficiency in the experiments. No blinding was performed. For immunofluorescence imaging, cells were imaged based on the DNA stain (DAPI) channel, to limit bias during the imaging procedure. The exact sample size of the associated experiments is described in the Figure legends. The exact number of replicates/experiments are indicated in the corresponding figures or their legends.

## Data availability

The source data produced in this study have been deposited to the BioStudies (https://www.ebi.ac.uk/biostudies/) under the accession number: S-BSST2031.

The source data of this paper are collected in the following database record: biostudies:S-SCDT-10_1038-S44318-025-00495-0.

## Peer review information

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

## Acknowledgements

We thank members of the Barr Lab for helpful discussions during this project. This work was funded by a Cancer Research UK program grant award DRCRPG-May23/100006 (FAB).

## Author contributions

**Tomoaki Sobajima**: Conceptualization; Data curation; Formal analysis; Supervision; Investigation; Visualization; Methodology; Writing—original draft; Writing—review and editing. **Luke J Fulcher**: Conceptualization; Data curation; Formal analysis; Supervision; Investigation; Visualization; Writing—original draft; Writing—review and editing. **Caleb Batley**: Formal analysis; Supervision; Investigation; Writing—original draft; Writing—review and editing. **Susanna J Alsop**: Investigation. **Jonah Veakins**: Investigation. **Francis A Barr**: Conceptualization; Data curation; Supervision; Funding acquisition; Writing—original draft; Project administration; Writing—review and editing.

Source data underlying figure panels in this paper may have individual authorship assigned. Where available, figure panel/source data authorship is listed in the following database record: biostudies:S-SCDT-10_1038-S44318-025-00495-0.

## Disclosure and competing interests statement

The authors declare no competing interests.

# Expanded View Figures

**Figure EV1. Cell cycle response to DNA double strand breaks in hTERT-RPE1 cells.**

(A) Western blot analysis of the indicated p53$^{KO}$, p21$^{KO}$, WIP1$^{KO}$, and Cdh1$^{KO}$ hTERT-RPE1 FUCCI cell lines (blots are representative of 3 independent experiments). (B) Representative images of cells stained for DNA damage markers 53BP1 and γ-H2A.X pS139 analysed in Fig. 1C ($n = 3$ independent experiments). Scale bars: 50 μm and 10 μm (inset). (C) p53$^{WT}$ hTERT-RPE1 cells treated with the indicated doses of NCS for 1 h were Western blotted with the antibodies indicated ($n = 3$ independent experiments). (D) Levels of γ-H2A-X pS139, p53 and p53 pSer15 from panel (C) are plotted (mean ± SEM; $n = 3$ independent experiments). (E) Representative images for the cells described in Fig. 1D ($n = 3$ independent experiments). Scale bar: 10 μm.

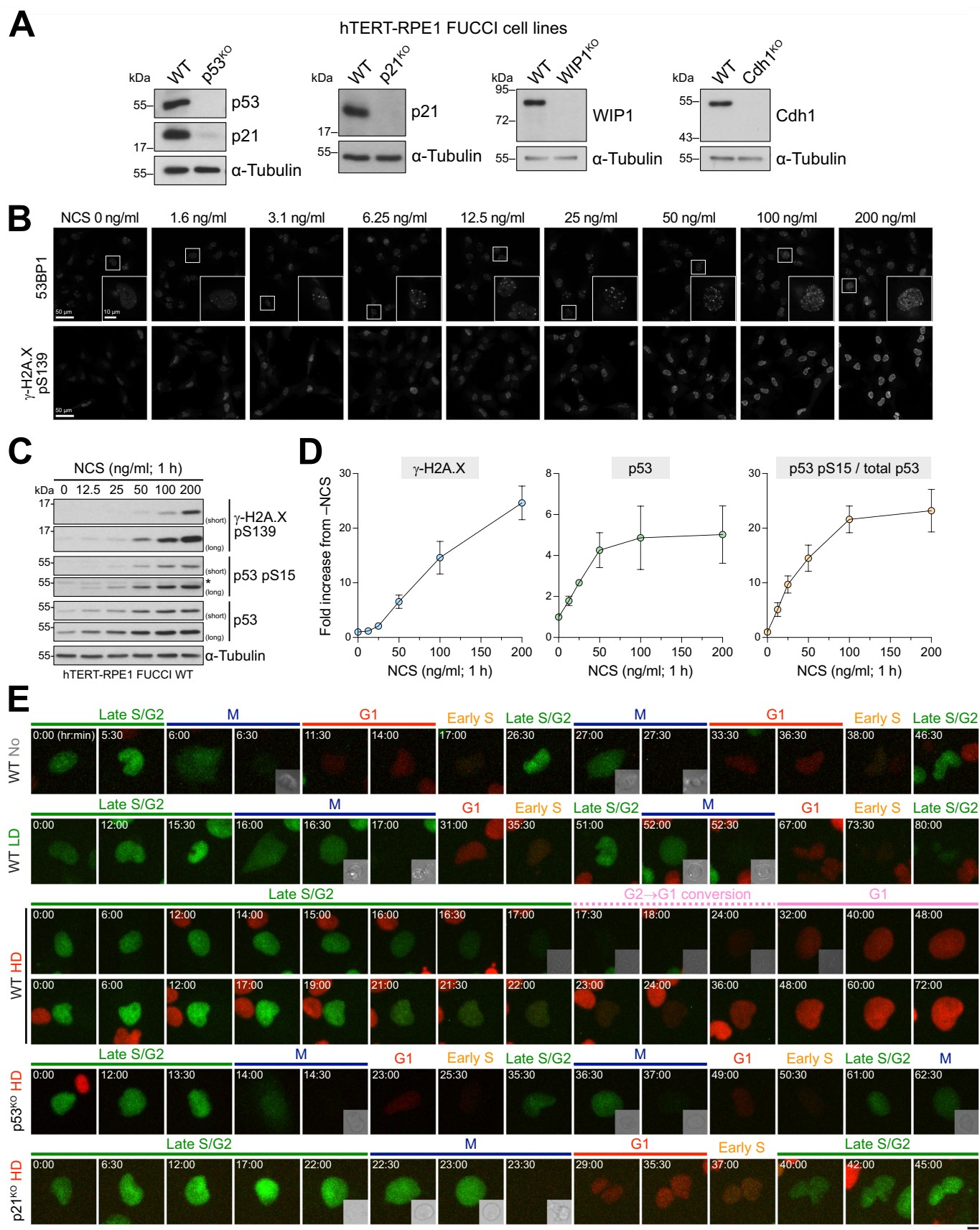

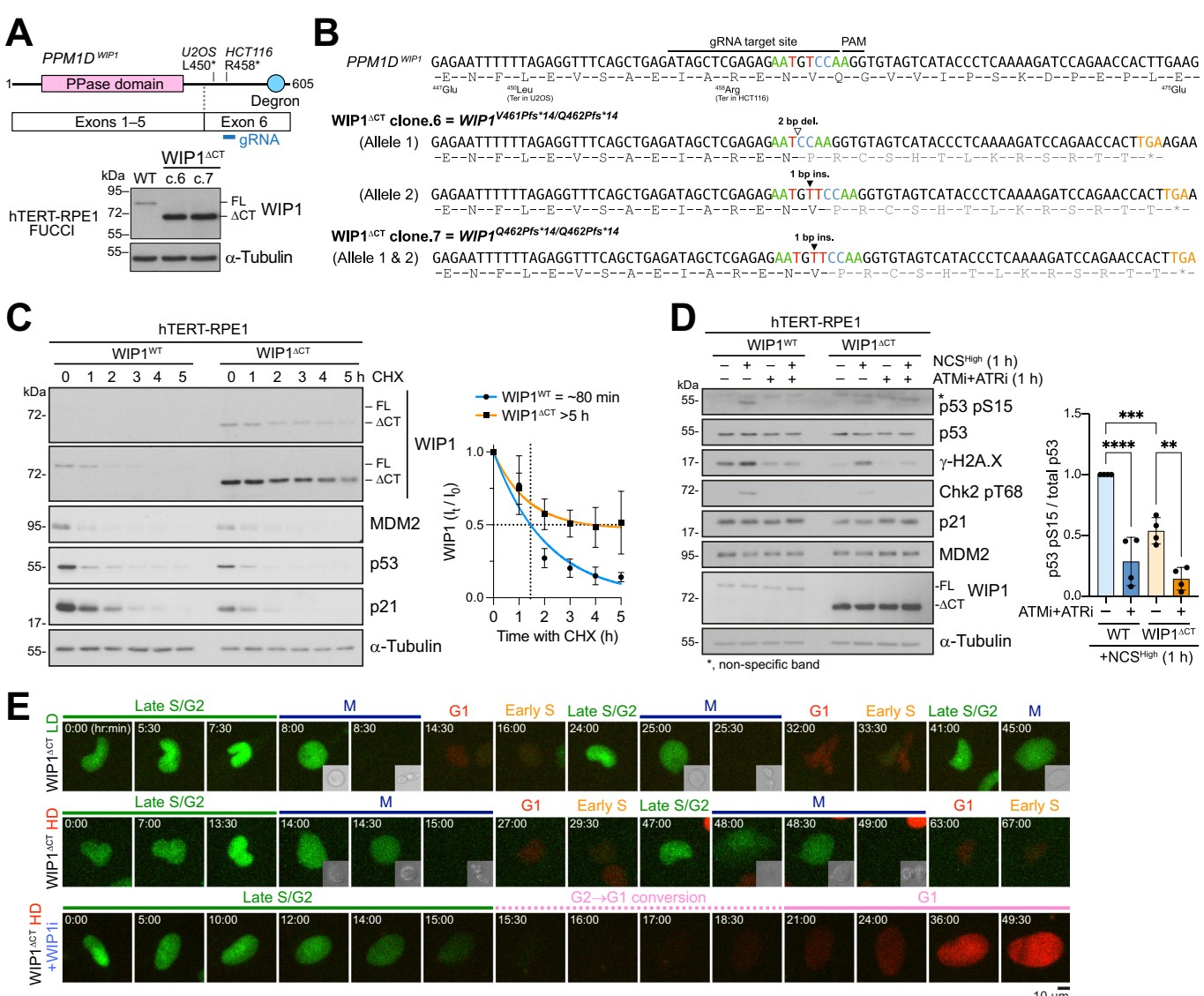

**Figure EV2. Generation and characterisation of hTERT-RPE1 WIP1ᐃCT cells.**

(**A**) Schematic detailing the CRISPR/Cas9 gene editing strategy used to obtain WIP1ᐃCT cells (top). Western blot validation of WIP1ᐃCT hTERT-RPE1 FUCCI cells (bottom; blots representative of 3 independent experiments). (**B**) DNA sequencing demonstrating successful WIP1ᐃCT generation in hTERT-RPE1 FUCCI cells ($n = 1$). (**C**) WIP1ᵂᵀ and WIP1ᐃCT hTERT-RPE1 FUCCI cells were treated with cycloheximide (CHX) for the indicated times and then Western blotted with the indicated antibodies (left). The relative half-lives of full-length WIP1 and WIP1ᐃCT are plotted (right; mean ± SEM; $n = 3$ independent experiments). (**D**) WIP1ᵂᵀ and WIP1ᐃCT hTERT-RPE1 FUCCI cells were treated with NCS and ATM/ATR inhibitors as shown for 1 h before lysis. Samples were subjected to immunoblotting with the indicated antibodies (left). The ratio of p53 pSer15/total p53 is plotted for the different conditions (right; mean ± SD; $n = 4$ independent experiments). Statistical significance was analysed using an ordinary one-way ANOVA with Tukey's multiple comparisons test (**$P < 0.01$; ***$P < 0.001$; ****$P < 0.0001$). (**E**) Representative images for the cells described in Fig. 2H ($n = 3$ independent experiments). Scale bar: 10 μm. All $P$ values are listed in Dataset EV1.

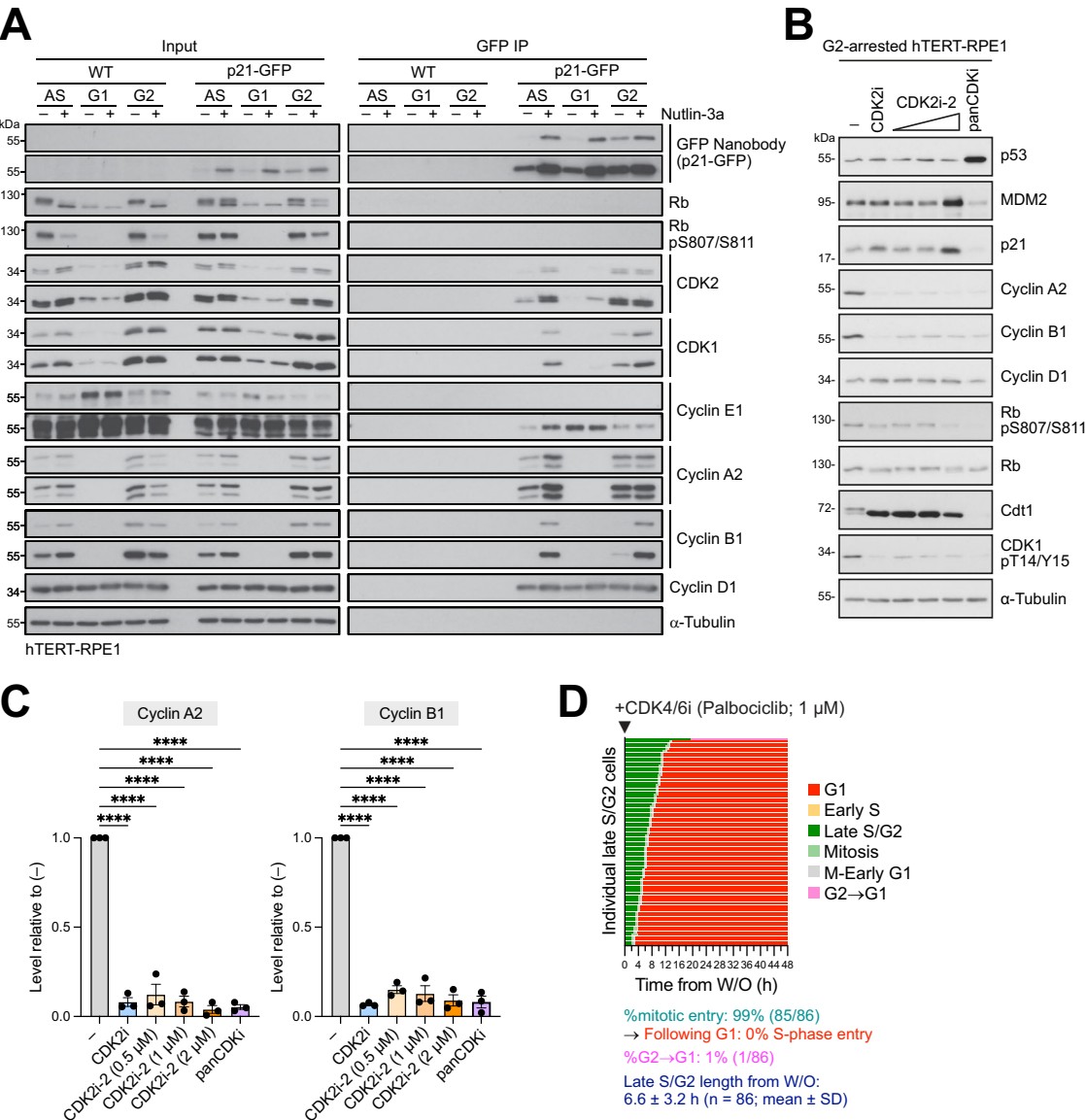

**Figure EV3. Differential targeting of CDK-cyclin complexes by p21 in G1 and G2.**

(**A**) Wild-type and p21-GFP hTERT-RPE1 cells were arrested in G1 with CDK4/6i or in G2 with CDK1i for 18 h. Asynchronous (AS) cells were included as a control. Cells were treated with or without Nutlin-3A for 6 h prior to lysis, and proteins were isolated from the lysates by GFP-TRAP beads (GFP IP). Input and IP extracts were subjected to immunoblotting with the indicated antibodies (*n* = 2 independent experiments). (**B**) Wild-type hTERT-RPE1 FUCCI cells were synchronised in G2 with CDK1 inhibitor for 18 h, treated with 100 nM of the CDK2 inhibitor PF-06873600 (CDK2i), 500 nM, 1 μM or 2 μM (left to right) of the alternative CDK2 inhibitor INX-315 (CDK2i-2), or 5 μM of the pan CDK inhibitor Flavopiridol (panCDKi) or DMSO (−) for 8 h, and Western blotted for cell cycle markers. (**C**) Cyclin A2 and B1 levels after 8 h treatment as in (**B**) are plotted for each condition (mean ± SEM; *n* = 3 independent experiments). Statistical significance was analysed using a one-way ANOVA with Dunnett's multiple comparisons test (****$P < 0.0001$). (**D**) Wild-type hTERT-RPE1 FUCCI cells were treated with the CDK4/6 inhibitor Palbociclib (CDK4/6i) for 1 h before late S/G2 cells were imaged continuously for 2 days. Cell cycle fate is plotted for individual cells. Pooled analyses are shown from 3 independent experiments. All *P* values are listed in Dataset EV1.

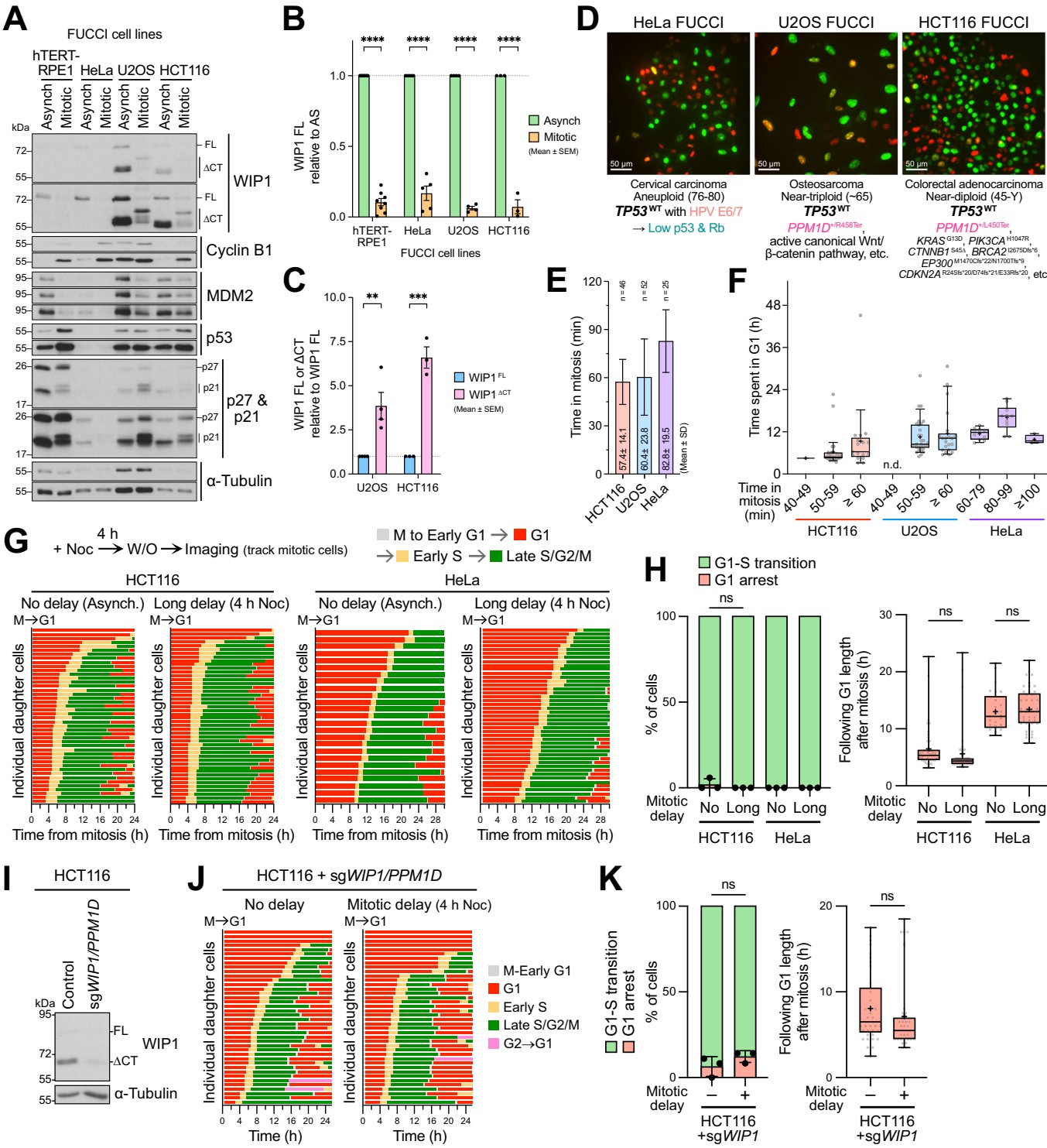

◄ **Figure EV4. Analysis of the mitotic timer in tumour cell lines with WIP1 mutations or lacking p53 function.**

(A) hTERT-RPE1, HeLa, U2OS and HCT116 FUCCI cells were arrested in mitosis for 18 h with nocodazole and western blotted with the indicated antibodies. Asynchronous (AS) cells were included as controls ($n = 3$–8 independent experiments). (B) The levels of full-length WIP1 in mitotic arrest conditions relative to AS from (A) are plotted (mean ± SEM; $n = 3$–8 independent experiments). Statistical significance was analysed using unpaired two-tailed $t$ test. (C) Relative levels of full-length WIP1 to the ΔCT fragment are plotted for U2OS and HCT116 cells described in (A) (mean ± SEM; $n = 3$–4 independent experiments). Statistical significance was analysed using unpaired two-tailed $t$ test. (D) Representative images of the HeLa, U2OS and HCT116 FUCCI cell lines, with known mutations highlighted underneath. Scale bars: 50 μm. (E) Mean time in mitosis in HCT116, U2OS and HeLa FUCCI cells in asynchronous culture is plotted (mean ± SD; pooled analyses shown from 3 independent experiments). (F) G1 length for HCT116, U2OS and HeLa FUCCI cells are plotted in a box and whiskers plot, categorised based on their time in mitosis (median, 25th and 75th percentiles and whiskers extending to minimum and maximum values; mean (+) for the different conditions; pooled analyses shown from 3 independent experiments). n.d.: no data points could be obtained. (G) HCT116 and HeLa FUCCI cells were treated with or without nocodazole to induce mitotic delay (4 h) as indicated. Nocodazole was removed by washing, and the mitotic cells were imaged continuously. Cell cycle fate is plotted for individual cells. Pooled analyses are shown from 3 independent experiments. (H) The percentage of cells undergoing the G1/S transition or G1 arrest following the treatments as in (G) is plotted (mean ± SD; $n = 3$ independent experiments) (left). Statistical significance was analysed using an unpaired two-tailed $t$ test. G1 length following mitosis in cells that entered mitosis following treatments as in (G) is plotted for individual cells in a box and whiskers plot (median, 25th and 75th percentiles and whiskers extending to minimum and maximum values, mean (+) for the different conditions; pooled analyses shown from 3 independent experiments) (right). Statistical significance was analysed using an unpaired two-tailed $t$ test. (I) Western blot showing efficient WIP1 knockout in HCT116 FUCCI cells targeted with WIP1 single guide RNA (sgWIP1/PPM1D) ($n = 1$ experiment). (J) HCT116 FUCCI cells described in (I) were treated with nocodazole to induce mitotic delay (4 h) or left untreated (No delay). Nocodazole was removed by washing, and the cells then imaged continuously for 24 h. Cell cycle fate is plotted for individual cells. Pooled analyses are shown from 3 independent experiments. (K) The percentage of cells undergoing the G1/S transition or G1 arrest after mitotic delay or no delay described in (J) is plotted (left) (mean ± SD; $n = 3$ independent experiments). Statistical significance was analysed using an unpaired two-tailed $t$ test. G1 length following mitosis in cells that entered mitosis following treatments as in (J) is plotted for individual cells in a box and whiskers plot (median, 25th and 75th percentiles and whiskers extending to minimum and maximum values, mean (+) for the different conditions; pooled analyses shown from 3 independent experiments) (right). Statistical significance was analysed using an unpaired two-tailed $t$ test. Significance for all experiments: **$P < 0.01$; ***$P < 0.001$; ****$P < 0.0001$; ns, not significant. All $P$ values are listed in Dataset EV1.

