## [Peer Review File · The EMBO Journal]

WIP1 mutations suppress DNA damage triggered bypass of the mitotic timer

Tomoaki Sobajima, Luke Fulcher, Caleb Batley, Susanna Alsop, Jonah Veakins, and Francis Barr

Corresponding author(s): Francis Barr (francis.barr@bioch.ox.ac.uk)

Review Timeline:

Submission Date:	28th Jan 25
Editorial Decision:	26th Feb 25
Revision Received:	20th May 25
Editorial Decision:	8th Jun 25
Revision Received:	9th Jun 25
Accepted:	12th Jun 25

Editor: Hartmut Vodermaier

Transaction Report:

Prof. Francis A. Barr
University of Oxford
Department of Biochemistry
South Parks Road
Oxford OX1 3QU
United Kingdom

26th Feb 2025

Re: EMBOJ-2025-120319
WIP1 mutations attenuate DNA damage triggered bypass of the mitotic timer

Dear Dr. Barr,

Thank you for submitting your study on the interplay between the G2 DNA damage checkpoint and the mitotic timer for our consideration. We have now received the reports from three expert referees, copied below for your information. As you will see, all referees appreciate the importance as well as the technical quality of the work, and would in principle support its publication. At the same time, they all do note that the manuscript would require major presentational modifications, in order to better highlight truly novel findings and differentiate them from more confirmatory data. In addition, referees 1 and 3 also raise a number of experimental concerns (ref 1 major points 1-3, ref 3 points 2, 3, 6, 7), whose satisfactory clarification would be important for strengthening the conclusiveness of the work.

Should you be able to adequately address these points, we would be happy to pursue a revised version further for EMBO Journal publication. Please be reminded that it is our policy to allow only a single round of major revision, making it important to carefully respond to all points at the time of resubmission. Also, please do not hesitate to contact me already during the early stages of the revision work with a tentative response letter, in case you would like discuss how to best tackle particular issues. Finally, should you require more time than our default three-months revision period, we would be happy to offer an extension, during which our 'scooping protection' (meaning that competing work appearing elsewhere in the meantime will not affect our considerations of your study) would of course remain valid.

Further information on preparing and uploading a revised manuscript can be found below and in our Guide to Authors. Thank you again for the opportunity to consider this work for The EMBO Journal, and I look forward to your revision.

With kind regards,

Hartmut Vodermaier

3) Revised manuscript text (including main tables, and figure legends for main and EV figures) has to be submitted as editable

text file (e.g., .docx format). We encourage highlighting of changes (e.g., via text color) for the referees' reference.

4) Each main and each Expanded View (EV) figure should be uploaded as individual production-quality files (preferably in .eps, .tif, .jpg formats). For suggestions on figure preparation/layout, please refer to our Figure Preparation Guidelines:

8) Please note that supplementary information at EMBO Press has been superseded by the 'Expanded View' for inclusion of additional figures, tables, movies or datasets; with up to five EV Figures being typeset and directly accessible in the HTML version of the article. For details and guidance, please refer to:

embopress.org/page/journal/14602075/authorguide#expandedview

9) To facilitate reproducibility and cross-laboratory adoption of methodologies, please structure the Materials & Methods section as outlined in our guide to authors, including a completed Reagents and Tools Table that can be downloaded from our author guidelines as well (<https://www.embopress.org/page/journal/14602075/authorguide#structuredmethods>).

10) Digital image enhancement is acceptable practice, as long as it accurately represents the original data and conforms to community standards. If a figure has been subjected to significant electronic manipulation, this must be clearly noted in the figure legend and/or the 'Materials and Methods' section. The editors reserve the right to request original versions of figures and the original images that were used to assemble the figure. Finally, we generally encourage uploading of numerical as well as gel/blot image source data; for details see: embopress.org/page/journal/14602075/authorguide#sourcedata

At EMBO Press, we ask authors to provide source data for the main manuscript figures. Our source data coordinator will contact you to discuss which figure panels we would need source data for and will also provide you with helpful tips on how to upload and organize the files.

Further information is available in our Guide For Authors:

In the interest of ensuring the conceptual advance provided by the work, we recommend submitting a revision within 3 months (27th May 2025). Please discuss the revision progress ahead of this time with the editor if you require more time to complete the revisions. Use the link below to submit your revision:

Link Not Available

Referee #1:

In this manuscript, Sobajima et al used live cell imaging to investigate the cell cycle trajectories upon induction of DNA double strand breaks. Using cells expressing the Fucci-based cell cycle indicator they describe how cells bypass mitosis after induction of DNA damage. Consistent with previous reports, they find that induction of DNA damage above certain threshold leads to a p53/p21/APC-mediated degradation of the mitotic cyclins in G2 cells followed by mitotic bypass and generation of G1 cells with 4N content (Krenning et al, Mol Cell 2014; Mullers et al, Cell Cycle 2014). They also show that besides APC-Cdh1, activation of APC-Cdc20 participates in the mitotic bypass (Fig. 4). Further, they observed that cells carrying truncating mutations in PPM1D phosphatase (also called WIP1) fail to activate p53 and progress to mitosis in the presence of DNA damage. This observation is in agreement with previous reports that showed G1 and G2 checkpoint override in cells carrying the mutant PPM1D as well as its ability to suppress p53 response (Kleiblova et al., JCB 2013; Burocchiova et al., Cell Death Dis. 2019; Stoyanov et al., Oncogene 2024). In the absence of DNA damage, the gain of function mutations of PPM1D had little effect on the cell cycle progression including the cells that experienced extended duration of mitosis. Authors conclude that PPM1D plays role mainly in suppressing cellular responses to exogenous stress while it is insensitive to activation of p53 pathway that is independent on ATM/ATR signaling. This conclusion is in line with the reported pathogenic behavior of PPM1D

mutations in therapy-induced hematological cancers including AML and MDS (Kahn et al., Blood 2018; Miller et al., Blood 2023; Burocchiova et al., Leukemia 2023). The authors have recently shown that mitotic degradation of MDM2 serves as a mitotic timer allowing activation of p53 pathway in daughter cells (Fulcher et al., Nat Cell Biol. 2025). The main finding reported here is that PPM1D does not participate in this intrinsic mitotic timer, which is interesting and further extends our knowledge of PPM1D functions. Overall, the study confirms previous findings that PPM1D activity allows override of the cell cycle checkpoints while having relatively little impact on normal cell cycle progression. These events are convincingly documented by following the cell cycle trajectories of individual control and CRISPR/Cas9-edited cells. The manuscript contains an extensive data set comprising of 7 main and 7 Supplementary figures. Presentation of the results as well as the statistical evaluation are convincing and support the main conclusions of the manuscript. Below I include some suggestions for improving the manuscript.

Major points

1. In Fig. 6 authors show that inhibition of CDK2 promotes mitotic bypass and they propose that this is caused by inability to sustain G2 rather than by failure to enter mitosis. However, PF-06873600 is a CDK2/4/6 inhibitor with similar IC50 values for all three CDKs and therefore should not be mentioned as CDK2 inhibitor. CDK2 inhibition is also suggested to trigger G1 arrest, but substantial fraction of G1 cells progressed to S phase despite the presence of CDK2i in Fig. 6F which is puzzling.
2. Based on data in Fig. 7, authors conclude that in nontransformed RPE cells the truncating PPM1D mutants do not inhibit the mitotic timer (in contrast to HCT116 and U2OS cells shown in Supplementary files) in the absence of DNA damage. On the other hand, Fig. 7E shows that the amount of cells that entered to S phase after extended mitosis was about doubled in PPM1D- Δ Ct cells compared to parental cells. This mild phenotype might reflect a partial inhibition of p53 observed in PPM1D- Δ Ct cells and not surprisingly the phenotype is probably much stronger in p53-KO cells. Different PPM1D truncating mutations may also show different levels of PPM1D stabilization and thus it is hard to strictly exclude that there is no phenotype in promoting proliferation after extended mitosis. Besides investigating just the cells with the truncating PPM1D mutation, it might be informative to determine the phenotype in PPM1D knock out cells or in parental cells treated with PPM1D inhibitor (GSK2830371). If PPM1D activity does not play any role in G1 arrest following after extended mitosis then no decrease in fraction of cells progressing to S phase would be expected in Fig. 7E.
3. Similarly as MDM2, PPM1D is also degraded in mitosis (Macurek et al., 2013) although by distinct molecular mechanism. It is possible that PPM1D does not sense the extended mitosis simply because it is degraded by APC-CDC20 during normal duration of mitosis. One possible explanation for the absence of phenotype of PPM1D- Δ Ct compared to the wild type PPM1D is that both are efficiently degraded by APC-CDC20 in mitosis (although exhibiting different stability during interphase). Comparing stability of both PPM1D variants in mitotic cells would help to explain this.

Minor points

PPM1D is the correct name of the gene, while WIP1 is sometimes used as a synonym for the protein product. It would be good to mention PPM1D in the text to avoid confusion.

Fig. 7D - this is not 4h mitotic delay as stated in the text but 18h synchronization with nocodazole as described in the figure legend

Introduction or Discussion sections should contain citations of relevant literature that showed the pathogenic behavior of PPM1D as result of silencing the cell cycle checkpoints and avoiding cell death.

Referee #2:

This study exploits a gain-of-function (GOF) cancer mutant of the p53 phosphatase WIP1 to investigate two distinct aspects of mitotic regulation. First, it examines the role of DNA damage response (DDR) kinases in promoting a mitotic bypass phenotype following G2 DNA damage. WIP1 GOF mutations attenuate the normal DNA damage-induced G2-to-G1 bypass, allowing cells to proceed through mitosis despite the presence of DNA damage. This failure to engage mitotic bypass appears to be associated with reduced CDK2 inhibition and impaired APC/C-mediated degradation of mitotic cyclins, thereby promoting continued cell cycle progression.

Second, the study investigates the role of the same DDR kinases in regulating the mitotic timer, a p53-dependent mechanism that governs G1 arrest in response to mitotic delays. The findings reveal that, in the absence of DNA damage, the mitotic timer remains functional in WIP1 GOF mutant cells, leading to a p53-dependent G1 arrest after prolonged mitosis. However, in the presence of DNA damage, WIP1 GOF mutations increase MDM2 levels, thereby suppressing the mitotic timer response and enabling continued cell cycle progression despite mitotic delays.

Together, these findings distinguish the regulatory pathways governing DNA damage-induced mitotic bypass from those controlling the mitotic timer, while also highlighting how oncogenic WIP1 mutations selectively disrupt DNA damage responses while leaving the mitotic timer intact under normal conditions.

While the first part of the study builds upon previous research published over a decade ago, it is solidly executed using an array of modern methodological approaches that were not available at the time. The original studies are properly referenced, and this manuscript adds a new layer of understanding by integrating the WIP1 mutation component into the existing framework. The final part of the study (Fig. 7) is more novel and impactful, as it provides new mechanistic insights into the distinct upstream determinants of p53 activation.

This reviewer believes that the manuscript is rigorous, well-supported, and makes a meaningful contribution to the field. Therefore, I recommend acceptance in its present form.

Referee #3:

Sobajima et al examine the interplay between two cell cycle checkpoints that restrain cell cycle progression in G2 or G1: the G2 DNA damage checkpoint and the mitotic timer. The checkpoints can both cause a p53-dependent cell cycle arrest, but the mechanisms differ. However, a core set of components are needed for both checkpoints and so a thorough investigation of if and how they are connected mechanistically is important and timely. This manuscript address this by focussing on an activating deletion in the WIP1 phosphatase. This is well-known to inhibit p53 and attenuate a DNA damage response, but it is unclear if and how this may impact on the mitotic timer.

Through a very carefully executed and controlled series of experiments the authors demonstrate that WIP1 mutation attenuates the mitotic timer, but only in the presence of prior DNA damage. They link this mechanistically to the upregulation of MDM2 prior to mitosis, which preserves MDM2 levels and inhibits the timer response after delayed mitosis. This follows on from important recent work by the Barr lab showing that MDM2 levels and its degradation during mitosis are important mechanistic aspects of the mitotic timer.

This was in my opinion the most novel and important result in the manuscript, but there are many other findings, including importantly that CDK2 inhibition causes cells to revert into G1 from G2 via APC-dependent degradation of cyclins. This has clinical significance because many CDK2 inhibitors are currently in clinical trials to overcome CDK4/6 inhibitor resistance.

The overall technical quality of the work is exceptional, and I have no criticism of any of the data shown. Most of the interpretations are logical and sound, although I question a few key points below. My main comments concern what is and isn't particularly novel about this study, whether the presentation could be improved by focussing on the key findings, and whether these could be improved by some additional experiments.

General major comments

When I read the manuscript, I felt that a large part of the data presented in the early figures was either known already or predictable based on current knowledge. I do not mean that comment to downplay the results, because it is entirely possible that I have missed the important advances. However, I thought we knew already why G2 delays cause mitotic bypass by p53-dependent p21 induction and APC-CDH1 mediated degradation of Cyclin A and B. Admittedly, there are some important new findings here, namely that APC-CDC20 may play a role and that CDK2 is the key target of p21. However, even this was hinted at previously by Johmumar et al (2014) and CDK4/6 may also be a crucial target given recent work by the Capell lab (as the authors point out in the discussion). In general, I felt that it was easy to lose these key points in the narrative with so much other data to consider. Perhaps I am just missing the key differences myself though, and maybe there is a clear distinction from the earlier studies by Krenning et al (2014), Johmura et al (2014), Mullers et al (2017), Zeng et al (2023); Zuniga et al., (2024) and others who have investigated the mechanism of mitotic bypass. The authors do cite some of these papers, but even in this case, I think it would be good for the authors to clarify better the key differences and to cite all the key preceding literature. If data is consistent with earlier work, then this should be stated and, in that case, perhaps figures could be reduced to supplementary data to allow the more important data to shine through more clearly.

In my opinion, the most exciting new findings in this study related to the link between the DNA damage checkpoint and the mitotic timer, and the suggestion that WIP1 deletion can abolish both these key cell cycle checkpoints. This could have important implications for cancer cells containing these mutations. However, this data is not presented until the very end and so it may be hard for the reader to extract this key information from all the other data presented. I also felt the key experiments could be validated more carefully, as highlighted in the specific comments below.

Specific major comments:

I have the following suggestions for new experiments and/or analysis. Some of these I view as crucial to validate the claims (2, 3, 6, 7), whilst others are interesting mechanistic questions that could add value to the manuscript (1, 4, 5). The final point (8) concerns re-arranging the manuscript to improve readability and to better highlight the novel findings.

1) It is surprising and intriguing that WIP1 mutation does attenuate the p53-dependent G1 DNA damage response but not the p53-dependent response to the mitotic timer delay. Is this also found at the level of p53 itself e.g. is there less p53-S15 following a damage arrest compared to timer-mediated G1 arrest or can this difference be explained mechanistically otherwise? Have the authors actually looked at the mitotic timer complex to see if levels are affected by WIP1 status or prior DNA damage?

2) The data in Figure 7G-J are intriguing and important since it links the mechanism of WIP1 mutation to MDM2 levels following DNA damage, and it suggests the WIP1 mutation could provide a double hit to cancer by overriding two different checkpoints. However, I feel it is important here to compare the levels of MDM2 in WIP1-WT cells during G2, even though they are not permissive to enter mitosis. At first, I presumed that the prediction would be that MDM2 levels are higher in dCT cells, thus

explaining the mitotic timer defects. However, I noticed that MDM2 levels are compared in WT and dCT cells in Fig-3A and they look identical after high DNA damage - can the authors reconcile the fact that MDM2 levels are similar between WT and dCT cells after high DNA damage, and yet the mitotic timer is abolished in dCT cells? Is it then the fact that dCT cells cannot arrest in G2 that is the key difference that has implications for the timer (i.e. WT cells would have a timer defect but they never reach mitosis)?

3) In relation to 2, if the lack of G2 arrest is the reason dCT cells can enter mitosis and have no timer, then what happens if the G2 checkpoint is bypassed in WIP1-WT cells after high DNA damage - e.g. after ATM/ATR inhibition. Is the mitotic timer similar abolished in this situation?

Addressing points 2 and 3 are important to validate the claim that MDM2 levels are crucial, and this may also explain that the key difference in WT/dCT cells is the inability of dCT to arrest in G2. This would have important clinical implication given that ATM/ATR inhibitors are currently in trials to treat cancer.

4) Do the cancer cells with gain-of-function mutations in WIP1 identified by Meitinger et al, also display a DNA-damage dependent inhibition of the mitotic timer, or is the timer basally inhibited in these cells, as suggested by Meitinger et al? If the latter, then do these cells produce enough endogenous DNA damage to elevate MDM2 in mitosis? In this case, does the WIP1 inhibitor used by Meitinger reduce MDM2 levels in these cells to override the timer?

5) In relation to 4, does a WIP1 inhibitor, as used by Meitinger, restore timer function in WIP1 mutant cells exposed to high DNA damage, and if so, is this only when given prior to the DNA damage?

6) Regarding this statement on line 209 "One difference was that WIP1 Δ CT cells took ~50% longer to complete mitosis after high levels of DNA damage, with a mean time in mitosis of 91 {plus minus} 35 min. Although these cells showed extended G1 length of 17.9 {plus minus} 6.5 h the majority were able to enter S phase, suggesting that in addition to an attenuation of the DNA damage response they bypass the mitotic timer response if DNA damage is present"

From Figure 2F I can see approx. 50% of cells that enter S-phase after transiting through mitosis and 50% that arrest in G1. Do the 50% that enter S-phase have a delayed mitosis, consistent with the conclusion that the timer response is bypassed? Also, what about the 50% that arrest in G1 - is their timer functional or do they arrest for some other reason? To answer these questions, it is crucial to plot mitotic arrest duration against G1 length/S-phase entry, to show that the G1 arrest specifically associated with a prolonged mitosis is abolished in dCT compared to WT cells.

7) In relation to the point above, can the authors rule out the alternative explanation that WIP1-dCT cells simply do not arrest in G1 after DNA damage, regardless of M duration? WIP1 mutation has been shown previously to impair IR-induced G1 arrest (Kleiblova et al., 2013).

8) The introduction to figure 3 makes the point that earlier reports have shown the destruction of cyclins in G2 after DNA damage, but they have not explored the relationship of the DDR with the mitotic timer. This indeed is an important novel aspect of this study, but the data looking at the relationship to the timer does not come until much later in figure 7. The intervening data is interesting and performed very well, but it seems largely an extension of what is already known and the key novel data on the timer does not come until the very last figure. I think the manuscript would benefit from condensing the proceeding data to focus on the key novel aspect and move the confirmation data into supplementary.

Minor points

9) The CDK2 data is clinically relevant and this could warrant discussion. I would also add that Johmura et al (2014) do hint at CDK2 being the crucial target of p21, so the finding could also be compared and contrasted to relevant earlier work.

10) Re this statement on line 80: "One caveat is that tumour cell lines deficient for the mitotic timer response carry many other mutations, and thus this correlation has not been confirmed as a causal change leading to attenuation of the timer response."

Meitinger et al (2024) show a partial recovery of the M timer in U2OS treated with WIP1 inhibitor

11) regarding line 127: "These findings suggest that the level of DNA damage needed to sustain a delay in G2 is higher than the level needed to trigger delays in G1"

This is also consistent with Barr AR et al (2017, Nat Comms) and it could be explained by many reasons, including that incomplete replication creates DSBs following mitosis and/or that APC-CDH1 activity rises to prevent SKP2-mediated p21 degradation post-mitosis. An altered timer threshold is another possibility, as the authors suggest, but this earlier work and possibilities should be briefly mentioned.

12) Regarding line 194: "However, like wild-type cells, delays in mitosis or treatment with a CDK4/6 inhibitor resulted in penetrant G1 arrest with only diploid WIP1 Δ CT cells (Figures 2D, WIP1 Δ CT CDK4/6i and M \rightarrow G1)."

This is an important result showing that WIP1 mutation does not abolish the mitotic timer. This could perhaps be emphasized here.

13) Figure 2E shows numbers of 53BP1 foci as well as % cells entering M in WIP1 wt-dCT cell after NCS. WIP1 dCT cells have a substantial decrease in CHK2pT68 and gH2AXpS139 after NCS treatment (Figure 2B), but there is no difference in the number of 53BP1 foci between WT & WIP1 dCT cells (Figure 2E). Can this difference be explained given gH2AX recruits 53BP1 to DSBs? It has been shown previously that ectopic expression of WT WIP1 reduces 53BP1 recruitment (Macurek et al, 2013).

14) Regarding line 289: "DNA damage extends G2 by delaying CDK1-cyclin B activation and inducing the broad spectrum CDK inhibitor p21. It was therefore important to understand which of these pathways was required to divert G2 cells to G1."

The authors should refer to previous work demonstrating the requirement of p21 for mitotic skipping, e.g., Zeng et al (2023); Zuniga et al., (2024), Krenning et al (2014), Johmura et al (2014).

Re: EMBOJ-2025-120319. Sobajima et al. WIP1 mutations attenuate DNA damage triggered bypass of the mitotic timer**Point-by-point response to reviewer comments****Referee #1:**

In this manuscript, Sobajima et al used live cell imaging to investigate the cell cycle trajectories upon induction of DNA double strand breaks. Using cells expressing the FUCCI-based cell cycle indicator they describe how cells bypass mitosis after induction of DNA damage. Consistent with previous reports, they find that induction of DNA damage above certain threshold leads to a p53/p21/APC-mediated degradation of the mitotic cyclins in G2 cells followed by mitotic bypass and generation of G1 cells with 4N content (Krenning et al, Mol Cell 2014; Mullers et al, Cell Cycle 2014). They also show that besides APC-Cdh1, activation of APC-Cdc20 participates in the mitotic bypass (Fig. 4). Further, they observed that cells carrying truncating mutations in PPM1D phosphatase (also called WIP1) fail to activate p53 and progress to mitosis in the presence of DNA damage. This observation is in agreement with previous reports that showed G1 and G2 checkpoint override in cells carrying the mutant PPM1D as well as its ability to suppress p53 response (Kleiblova et al., JCB 2013; Burocchiova et al., Cell Death Dis. 2019; Stoyanov et al., Oncogene 2024). In the absence of DNA damage, the gain of function mutations of PPM1D had little effect on the cell cycle progression including the cells that experienced extended duration of mitosis. Authors conclude that PPM1D plays role mainly in suppressing cellular responses to exogenous stress while it is insensitive to activation of p53 pathway that is independent on ATM/ATR signaling. This conclusion is in line with the reported pathogenic behavior of PPM1D mutations in therapy-induced hematological cancers including AML and MDS (Kahn et al., Blood 2018; Miller et al., Blood 2023; Burocchiova et al., Leukemia 2023). The authors have recently shown that mitotic degradation of MDM2 serves as a mitotic timer allowing activation of p53 pathway in daughter cells (Fulcher et al., Nat Cell Biol. 2025). The main finding reported here is that PPM1D does not participate in this intrinsic mitotic timer, which is interesting and further extends our knowledge of PPM1D functions. Overall, the study confirms previous findings that PPM1D activity allows override of the cell cycle checkpoints while having relatively little impact on normal cell cycle progression. These events are convincingly documented by following the cell cycle trajectories of individual control and CRISPR/Cas9-edited cells. The manuscript contains an extensive data set comprising of 7 main and 7 Supplementary figures. Presentation of the results as well as the statistical evaluation are convincing and support the main conclusions of the manuscript. Below I include some suggestions for improving the manuscript.

Major points

1. In Fig. 6 authors show that inhibition of CDK2 promotes mitotic bypass and they propose that this is caused by inability to sustain G2 rather than by failure to enter mitosis. However, PF-06873600 is a CDK2/4/6 inhibitor with similar IC50 values for all three CDKs and therefore should not be mentioned as CDK2 inhibitor. CDK2 inhibition is also suggested to trigger G1 arrest, but substantial fraction of G1 cells progressed to S phase despite the presence of CDK2i in Fig. 6F which is puzzling.

Response: The reviewer raises two points, first about CDK2 inhibitor specificity and the second about progression of G1 cells into S phase which we have addressed.

1. We have added new data using an additional chemically unrelated CDK2 inhibitor (new Figure EV3B-EV3C), and explain how existing and new controls show the effects observed are due to CDK2 inhibition rather than CDK4/6 inhibition. PF-06873600 is approximately 10-fold better at inhibiting CDK2 than CDK6 *in vitro* (PMID: 34110834, Compound #22). Furthermore, published reports indicate that in cells 100 nM PF-06873600 primarily targets

CDK2, whereas only higher concentrations inhibit the activity of CDK4/6 complexes as well (PMID: 37267950; PMID: 34520734). Consequently, we used 100 nM PF-06873600 to enable selective CDK2 inhibition without significantly affecting CDK4/6 activity. Control conditions directly targeting CDK4/6 further support the idea the effects seen with PF-06873600 are due to CDK2 inhibition. Palbociclib, a highly specific CDK4/6 inhibitor, triggers G1 cell cycle arrest with 2C DNA content, i.e. a diploid G1 arrest without mitotic skipping (Figure 1H – CDK4/6i). Additionally, Palbociclib-treated G2 cells still enter and complete mitosis before arresting in G1 (new Figure EV3D). In contrast, 100 nM PF-06873600 leads to G2 collapse to G1 without mitosis, and crucially the cells still progress into S phase (Figure 6E-6F). If PF-06873600 were effectively targeting CDK4/6 activity under these experimental conditions, one would expect the cells to arrest in G1 after mitotic skipping since CDK4/6 are needed to exit G1 and enter S phase. To further support this conclusion, we have used a reportedly more selective CDK2 inhibitor (INX-315, CDK2i-2) with an IC₅₀ value of 0.6 and 2.5 nM for CDK2/cyclin E1 and CDK2/cyclin A2, respectively (PMID: 38047585). Conversely, the IC₅₀ value for CDK4/cyclin D1 and CDK6/cyclin D3 are 126 and 349 nM, respectively. In good agreement with the PF-06873600 data, we observed G2 collapse to G1 with INX-315 (CDK2i-2 in new Figure EV3B-EV3C).

2. The referee writes that a “*substantial fraction of G1 cells progressed to S phase despite the presence of CDK2i*”. This is the expected result if only CDK2 is inhibited. To fully prevent entry into S phase and trap cells in G1 would require additional CDK4/6 inhibition. In the context of the paper, we use CDK2 inhibitors to trigger the G2 collapse process and mitotic skipping without inducing the natural inhibitor p21 which targets multiple CDK complexes including CDK2/4/6. These experiments were also relevant to demonstrate that DNA damage is not required for the G2 to G1 skipping process *per se*, but simply inhibiting CDK2 activity (a function performed by p21 during G2 collapse to G1) is sufficient. Under physiological circumstances G2 collapsed cells would have high levels of p21 that would inhibit CDK4/6 in G1 to lock in the cell cycle arrest. Absence of p21 induction with chemical CDK2 inhibitors allows skipped cells to enter the next S phase, whereas DNA damage induced skipped cells would remain arrested in G1 due to p21-dependent inhibition of CDK4/6. Importantly, when cells undergo G2 collapse we observe destruction of cyclins A and B, whereas the CDK4/6-activating cyclin D remains stable (see Figures 3-6). This means that CDK1 and 2 activities are more strongly attenuated during G2 collapse. Thus, while we are using CDK2 inhibitors as an alternative way of demonstrating CDK2 inhibition is a prerequisite for G2 collapse, the biochemical data showing cyclin A and B destruction, but not cyclin D, supports this model where CDK2 activity is the key driver of the G2 state, and the p53-p21 pathway can overcome this under DNA damage conditions to trigger G2 collapse.

2. Based on data in Fig. 7, authors conclude that in non-transformed RPE cells the truncating PPM1D mutants do not inhibit the mitotic timer (in contrast to HCT116 and U2OS cells shown in Supplementary files) in the absence of DNA damage. On the other hand, Fig. 7E shows that the amount of cells that entered to S phase after extended mitosis was about doubled in PPM1D-ΔCt cells compared to parental cells. This mild phenotype might reflect a partial inhibition of p53 observed in PPM1D-ΔCt cells and not surprisingly the phenotype is probably much stronger in p53-KO cells. Different PPM1D truncating mutations may also show different levels of PPM1D stabilization and thus it is hard to strictly exclude that there is no phenotype in promoting proliferation after extended mitosis. Besides investigating just the cells with the truncating PPM1D mutation, it might be informative to determine the phenotype in PPM1D knock out cells or in parental cells treated with PPM1D inhibitor (GSK2830371). If PPM1D activity does not play any role in G1 arrest following after extended mitosis then no decrease in fraction of cells progressing to S phase would be expected in Fig. 7E.

Response: To date there has been no definitive demonstration that WIP1/PPM1D truncating mutations are causal for loss of timer function in U2OS and HCT116 cells. One issue with studying cancer cell lines is the large number of mutations, genomic rearrangements and

ploidy changes which render it difficult to establish causality, despite the reported correlation. For that reason, we have carried out our study in a diploid cell background (hTERT-RPE1). This approach allowed us to specifically test the hypothesis that WIP1 mutations are causal for the loss of timer response. Our important findings are that WIP1 Δ CT cells still respond to mitotic delays and remain competent for G1 arrest following prolonged mitosis (see Figure 7B-7D), as long as there is no prior DNA damage in G2.

To fully address the comment of the referee we decided to directly test the involvement of truncating WIP1 mutations in the mitotic timer pathway in U2OS and HCT116 cells using WIP1 knockout. If the WIP1 Δ CT mutations are causal for loss of the mitotic timer response, then knockout should rescue that activity. We have therefore added additional analysis of different WIP1 knockout (WIP1 KO) cell lines. As expected, WIP1 KO in hTERT-RPE1 cells had no effect on the mitotic timer pathway (see new panels in Figure 7B-7D and Figure 8C). This reinforces the conclusion that the mitotic timer is a DNA damage-independent mechanism and that WIP1 does not play a direct role in the pathway in the absence of DNA damage. Previous work has suggested that chemical inhibition of WIP1 in tumour cell lines expressing WIP1 Δ CT alleles can reactivate the mitotic timer pathway. To fully test this idea, we knocked out PPM1D/WIP1 in U2OS and HCT116 cells which is the cleaner and more definitive approach. Neither WIP1 KO in U2OS or HCT116 cells nor WIP1 inhibition in U2OS cells restored mitotic timer function. This compelling new data strongly suggests that WIP1 mutation is not causal for loss of mitotic timer function, and that other genetic changes render these cancer cell lines unable to respond to mitotic delays (see new Figures 7E-7K, and Figure EV4I-EVK). As an alternative approach, we treated WIP1 Δ CT hTERT-RPE1 FUCCI G2 cells with DNA damage in the presence of the WIP1/PPM1D inhibitor GSK2830371. While in the absence of the WIP1 inhibitor WIP1 Δ CT cells enter mitosis in the presence of DNA damage (see Figure 2H-I), inhibition of WIP1 largely prevented mitotic entry and cells skipped to G1 without cell division (see new panel in Figure 2H-I).

In summary, inhibition of WIP1 prevents WIP1 Δ CT cells from tolerating the DNA damage insult and entering mitosis. However, genetic removal of WIP1 from hTERT-RPE1 cells did not affect the mitotic timer threshold and WIP1 KO cells still respond to mitotic delays like wild-type cells. In U2OS and HCT116 cells harbouring gain-of-function WIP1 Δ CT alleles, WIP1 KO did not restore mitotic timer functionality in the absence of DNA damage. Therefore, WIP1 does not play a direct role in the mitotic timer pathway and only influences the outcome of timer activation if cells are pre-exposed to DNA damage before entering mitosis.

3. Similarly as MDM2, PPM1D is also degraded in mitosis (Macurek et al., 2013) although by distinct molecular mechanism. It is possible that PPM1D does not sense the extended mitosis simply because it is degraded by APC-CDC20 during normal duration of mitosis. One possible explanation for the absence of phenotype of PPM1D- Δ Ct compared to the wild type PPM1D is that both are efficiently degraded by APC-CDC20 in mitosis (although exhibiting different stability during interphase). Comparing stability of both PPM1D variants in mitotic cells would help to explain this.

Response: As the reviewer notes, WIP1/PPM1D is reported to be an APC/C-CDC20 substrate. If this is correct then WIP1 would be rapidly and efficiently degraded in every mitosis, regardless of length. This notion, in combination with the data that neither WIP1 KO nor WIP1 Δ CT gain-of-function cells exhibit defects in the mitotic timer pathway (see Figure 7), rule out a direct role for WIP1 in sensing extended mitosis.

Under the conditions we have used, both full-length and Δ CT WIP1 forms are still evident during mitotic delays in hTERT-RPE1, U2OS and HCT116 cells, albeit less than interphase cells (Figure EV4A). However, we noticed that both full length and Δ CT WIP1 forms exhibit an electrophoretic mobility shift in M phase samples, which is weakly detected by the WIP1

antibody, suggesting potential post-translational modification. To test this further, we arrested wild-type and WIP1 Δ CT hTERT-RPE1 cells in mitosis and forced mitotic exit by treating cells with the pan-CDK inhibitor Flavopiridol. Under these conditions we observed rapid reappearance of the WIP1 band by Western blot upon Flavopiridol treatment of mitotic cells, suggesting WIP1 may be post-translationally modified in M phase (see Rebuttal Figure 1 below). In contrast, rapid and slower targets of the APC/C e.g. cyclin A, cyclin B and Aurora A were destroyed during mitotic exit, and did not reappear within the experimental time frame (see Rebuttal Figure 1). Because WIP1 KO cells do not affect the mitotic timer response (see new Figure 7, and EV4), these data strongly argue that mitotic destruction or modification of WIP1 are not playing a direct role in time sensing or in the response to delays in mitosis. Furthermore, we do not observe loss of WIP1 during mitotic skipping from G2 to G1 in either wild-type or Cdh1KO cells (see Figures 3-4). This behaviour is in contrast to other APC/C substrates such as cyclins A and B, and geminin (new Figure 4F-4G) that are rapidly destroyed during G2 collapse. Destruction of Cyclins A, B and Geminin during G2 collapse to G1 in response to DNA damage can be blocked through inhibiting the APC/C with the chemical inhibitor proTAME, whereas WIP1 levels are not affected by proTAME treatment (new Figure 4F-4G, and Rebuttal Figure 1).

Rebuttal Figure 1. Left: Wild-type (WT), WIP1 Δ CT and WIP1KO hTERT-RPE1 cells were arrested in mitosis for 18 h using 25 ng/ml nocodazole. Following treatment, mitotic cells were isolated by shake-off, and either lysed immediately (M) or incubated with the CDK inhibitor Flavopiridol (5 μ M) for 30 or 60 minutes to trigger mitotic exit to G1. Asynchronous (AS) cells were used as controls. Whole cell extracts were subjected to Western blotting with the indicated antibodies.

Right: Quantification of WIP1 protein levels during DNA damage induced G2 collapse in the absence and presence of the APC/C inhibitor proTAME (related to Figure 4F).

Minor points

PPM1D is the correct name of the gene, while WIP1 is sometimes used as a synonym for the protein product. It would be good to mention PPM1D in the text to avoid confusion.

Response: We have now added the gene name PPM1D to the abstract and introduction.

Fig. 7D - this is not 4h mitotic delay as stated in the text but 18h synchronization with nocodazole as described in the figure legend

Response: We thank the reviewer for bringing this to our attention. This has now been corrected.

Introduction or Discussion sections should contain citations of relevant literature that showed the pathogenic behavior of PPM1D as result of silencing the cell cycle checkpoints and avoiding cell death.

Response: Additional discussion and citations of this literature have been added to the discussion.

Referee #2:

This study exploits a gain-of-function (GOF) cancer mutant of the p53 phosphatase WIP1 to investigate two distinct aspects of mitotic regulation. First, it examines the role of DNA damage response (DDR) kinases in promoting a mitotic bypass phenotype following G2 DNA damage. WIP1 GOF mutations attenuate the normal DNA damage-induced G2-to-G1 bypass, allowing cells to proceed through mitosis despite the presence of DNA damage. This failure to engage mitotic bypass appears to be associated with reduced CDK2 inhibition and impaired APC/C-mediated degradation of mitotic cyclins, thereby promoting continued cell cycle progression.

Second, the study investigates the role of the same DDR kinases in regulating the mitotic timer, a p53-dependent mechanism that governs G1 arrest in response to mitotic delays. The findings reveal that, in the absence of DNA damage, the mitotic timer remains functional in WIP1 GOF mutant cells, leading to a p53-dependent G1 arrest after prolonged mitosis. However, in the presence of DNA damage, WIP1 GOF mutations increase MDM2 levels, thereby suppressing the mitotic timer response and enabling continued cell cycle progression despite mitotic delays.

Together, these findings distinguish the regulatory pathways governing DNA damage-induced mitotic bypass from those controlling the mitotic timer, while also highlighting how oncogenic WIP1 mutations selectively disrupt DNA damage responses while leaving the mitotic timer intact under normal conditions.

While the first part of the study builds upon previous research published over a decade ago, it is solidly executed using an array of modern methodological approaches that were not available at the time. The original studies are properly referenced, and this manuscript adds a new layer of understanding by integrating the WIP1 mutation component into the existing framework.

The final part of the study (Fig. 7) is more novel and impactful, as it provides new mechanistic insights into the distinct upstream determinants of p53 activation.

This reviewer believes that the manuscript is rigorous, well-supported, and makes a meaningful contribution to the field. Therefore, I recommend acceptance in its present form.

Response: We thank the reviewer for their critical appraisal of our manuscript, and for highlighting the advantages of revisiting and building on previous observations, using more modern technological approaches.

Referee #3:

Sobajima et al examine the interplay between two cell cycle checkpoints that restrain cell cycle progression in G2 or G1: the G2 DNA damage checkpoint and the mitotic timer. The checkpoints can both cause a p53-dependent cell cycle arrest, but the mechanisms differ. However, a core set of components are needed for both checkpoints and so a thorough investigation of if and how they are connected mechanistically is important and timely. This manuscript address this by focussing on an activating deletion in the WIP1 phosphatase. This is well-known to inhibit p53 and attenuate a DNA damage response, but it is unclear if and how this may impact on the mitotic timer.

Through a very carefully executed and controlled series of experiments the authors demonstrate that WIP1 mutation attenuates the mitotic timer, but only in the presence of prior DNA damage. They link this mechanistically to the upregulation of MDM2 prior to mitosis, which preserves MDM2 levels and inhibits the timer response after delayed mitosis. This follows on from important recent work by the Barr lab showing that MDM2 levels and its degradation during mitosis are important mechanistic aspects of the mitotic timer.

This was in my opinion the most novel and important result in the manuscript, but there are many other findings, including importantly that CDK2 inhibition causes cells to revert into G1 from G2 via APC-dependent degradation of cyclins. This has clinical significance because many CDK2 inhibitors are currently in clinical trials to overcome CDK4/6 inhibitor resistance.

The overall technical quality of the work is exceptional, and I have no criticism of any of the data shown. Most of the interpretations are logical and sound, although I question a few key points below. My main comments concern what is and isn't particularly novel about this study, whether the presentation could be improved by focussing on the key findings, and whether these could be improved by some additional experiments.

General major comments

When I read the manuscript, I felt that a large part of the data presented in the early figures was either known already or predictable based on current knowledge. I do not mean that comment to downplay the results, because it is entirely possible that I have missed the important advances. However, I thought we knew already why G2 delays cause mitotic bypass by p53-dependent p21 induction and APC-CDH1 mediated degradation of Cyclin A and B. Admittedly, there are some important new findings here, namely that APC-CDC20 may play a role and that CDK2 is the key target of p21. However, even this was hinted at previously by Johmurar et al (2014) and CDK4/6 may also be a crucial target given recent work by the Capell lab (as the authors point out in the discussion). In general, I felt that it was easy to lose these key points in the narrative with so much other data to consider. Perhaps I am just missing the key differences myself though, and maybe there is a clear distinction from the earlier studies by Krenning et al (2014), Johmura et al (2014), Mullers et al (2017), Zeng et al (2023); Zuniga et al., (2024) and others who have investigated the mechanism of mitotic bypass. The authors do cite some of these papers, but even in this case, I think it would be good for the authors to clarify better the key differences and to cite all the key preceding literature. If data is consistent with earlier work, then this should be stated and, in that case, perhaps figures could be reduced to supplementary data to allow the more important data to shine through more clearly.

In my opinion, the most exciting new findings in this study related to the link between the DNA damage checkpoint and the mitotic timer, and the suggestion that WIP1 deletion can abolish both these key cell cycle checkpoints. This could have important implications for cancer cells containing these mutations. However, this data is not presented until the very end and so it may be hard for the reader to extract this key information from all the other

data presented. I also felt the key experiments could be validated more carefully, as highlighted in the specific comments below.

Response: Our work builds on previous important studies, which we have discussed and cited. We do not see any of the data in our current manuscript as confirmatory to previous research, but as essential to fully understand how G2 DNA damage can impact the mitotic timer pathway, and how this behaviour differs under our different conditions including in the WIP1 Δ CT and KO cell lines. Most of the comparisons we show require this data and cannot simply be cited due to the need to compare carefully matched experimental conditions. As the reviewer notes, in addition to our exciting new findings, during this process we have uncovered some missing details on APC/C-CDC20 and p21-dependent inhibition of CDK2. This highlights a major advantage of generating comprehensive experimental datasets and argues in favour of including all data rather than leaving readers to wonder how the conditions relate to previous work. To better emphasise the more novel aspects of the work on how DNA damage signalling and the mitotic timer pathway interact, we have rewritten to text and added crucial key experiments in the fully revised Figure 7 and new Figure 8. These changes are addressed in more detail below.

Specific major comments:

I have the following suggestions for new experiments and/or analysis. Some of these I view as crucial to validate the claims (2, 3, 6, 7), whilst others are interesting mechanistic questions that could add value to the manuscript (1, 4, 5). The final point (8) concerns re-arranging the manuscript to improve readability and to better highlight the novel findings.

1) It is surprising and intriguing that WIP1 mutation does attenuate the p53-dependent G1 DNA damage response but not the p53-dependent response to the mitotic timer delay. Is this also found at the level of p53 itself e.g. is there less p53-S15 following a damage arrest compared to timer-mediated G1 arrest or can this difference be explained mechanistically otherwise? Have the authors actually looked at the mitotic timer complex to see if levels are affected by WIP1 status or prior DNA damage?

Response: We and others have tested timer arrested cells for the presence of p53 phosphorylation markers, indicative of DNA damage, and have been unable to detect these or other more generic markers of DNA damage (γ H2AX, p53BP1 foci) (PMID: 15286707; 20832319, 39789219). Indeed, these observations and the existing and new data presented here that neither WIP1 KO nor WIP1 Δ CT mutations have any appreciable effect on the mitotic timer pathway strongly suggest that it is a DNA damage independent response (see also response to Reviewer 1 above). While DNA damage signalling relies on p53 phosphorylation to trigger the p53-dependent response, the mitotic timer is activated by progressive loss of MDM2 as time in mitosis increases. These different routes to p53 activation demonstrate the DNA damage dependent and independent role of p53 in cell cycle fate.

2) The data in Figure 7G-J are intriguing and important since it links the mechanism of WIP1 mutation to MDM2 levels following DNA damage, and it suggests the WIP1 mutation could provide a double hit to cancer by overriding two different checkpoints. However, I feel it is important here to compare the levels of MDM2 in WIP1-WT cells during G2, even though they are not permissive to enter mitosis. At first, I presumed that the prediction would be that MDM2 levels are higher in Δ CT cells, thus explaining the mitotic timer defects. However, I noticed that MDM2 levels are compared in WT and Δ CT cells in Fig-3A and they look identical after high DNA damage - can the authors reconcile the fact that MDM2 levels are similar between WT and Δ CT cells after high DNA damage, and yet the mitotic timer is

abolished in dCT cells? Is it then the fact that dCT cells cannot arrest in G2 that is the key difference that has implications for the timer (i.e. WT cells would have a timer defect but they never reach mitosis)?

Response: The reviewer is correct with their latter interpretation. Wild-type cells exposed to G2 DNA damage do not enter mitosis, and the higher level of MDM2 simply signifies the level of p53 activity in those cells. The CDK-inhibitor p21 is also induced and triggers G2 collapse to G1 in the absence of cell division. WIP1 Δ CT cells are strongly attenuated for p53-dependent responses downstream of DNA damage signalling. Thus, in WIP1 Δ CT cells, p21 increases but does not reach the threshold required for mitotic collapse (see Figure 3). Similarly, there is a rise in the level of MDM2. Hence, damaged WIP1 Δ CT cells enter mitosis with more MDM2 than normal and thus can tolerate longer mitotic delays before activating and enforcing a timer arrest.

3) In relation to 2, if the lack of G2 arrest is the reason dCT cells can enter mitosis and have no timer, then what happens if the G2 checkpoint is bypassed in WIP1-WT cells after high DNA damage - e.g. after ATM/ATR inhibition. Is the mitotic timer similar abolished in this situation?

Addressing points 2 and 3 are important to validate the claim that MDM2 levels are crucial, and this may also explain that the key difference in WT/dCT cells is the inability of dCT to arrest in G2. This would have important clinical implication given that ATM/ATR inhibitors are currently in trials to treat cancer.

Response: Shortening G2 length by collapsing the G2 checkpoint with Wee1 inhibitors does not change the fate of DNA damaged wild-type cells, which still collapse G2 and arrest in G1 as a tetraploid (Figure 5A-5B). Thus, shortening of G2 length does not appear to explain the ability of WIP1 Δ CT cells to enter mitosis in the presence of DNA damage. The reviewer suggests an ATM/ATR inhibition strategy aimed at forcing wild-type cells to circumvent the G2 damage signalling and enter mitosis with higher levels of MDM2 to try and override the mitotic timer pathway. Inhibiting ATM/ATR signalling does indeed allow G2 DNA damaged wild-type cells to enter mitosis (Figure 1E), but the suppression of ATM/ATR signalling also has a direct effect on p53 signalling and hence MDM2. DNA damage signalling through ATM/ATR leads to G2 checkpoint activation, p53/MDM2 phosphorylation, p53 stabilisation, and elevated target gene expression including MDM2 and p21 (Figure 3). Inhibition of ATM/ATR during G2 DNA damage thus prevents the p53-dependent rise in MDM2 (**Rebuttal Figure 2 – see next page**), so this strategy cannot be used as suggested. A further caveat to the approach is that wash out of the ATM/ATR inhibitors once cells have entered mitosis and divided would inevitably lead to arrest in G1 due to the G1 DNA damage response. To overcome these issues, we use the WIP1 Δ CT mutation as a tool to uncouple p53-dependent DNA damage arrest from p53-dependent mitotic timer arrest (new Figure 8).

Rebuttal Figure 2. Wild-type hTERT-RPE1 cells were synchronised in G2 using the CDK1 inhibitor RO-3306 (6 μ M; 18 h). G2 cells were then lysed as a control (-) or treated with high dose neocarzinostatin (NCS) (HD DNA damage) for 1, 2, 4 or 8 h in the presence or absence of ATM/ATR inhibitors (2 μ M each). Cells were lysed at each time point. Cell extracts were subjected to Western blotting with the indicated antibodies. Relative Cyclin A and B levels are plotted on the right (n=3; error bars = mean \pm SEM).

4) Do the cancer cells with gain-of-function mutations in WIP1 identified by Meitinger et al, also display a DNA-damage dependent inhibition of the mitotic timer, or is the timer basally inhibited in these cells, as suggested by Meitinger et al? If the latter, then do these cells produce enough endogenous DNA damage to elevate MDM2 in mitosis? In this case, does the WIP1 inhibitor used by Meitinger reduce MDM2 levels in these cells to override the timer?

Response: We have knocked out both the full-length and truncated Δ CT mutant WIP1 forms in U2OS and HCT116 cells harbouring cancer-associated gain-of-functions mutations in WIP1. These WIP1 KO cells are still deficient for the mitotic timer response (Figure 7E-7K and Figure EV4G-EV4K). These data, in combination with data on hTERT-RPE1 cells showing that neither loss of WIP1 nor gain-of-function WIP1 Δ CT mutations impact the mitotic timer threshold/functionality in the absence of DNA damage (Figure 7B-7D), strongly suggest that WIP1 is not the causal factor explaining loss of the mitotic timer pathway in U2OS and HCT116 cells. Treating DNA damaged WIP1 Δ CT hTERT-RPE1 cells with the WIP1 inhibitor causes them to collapse from G2 to G1 without cell division (Figure 2H-2I). Lowering WIP1 activity in WIP1 Δ CT cells therefore prevents these cells from entering mitosis in the presence of DNA damage, and stops them from over-riding the mitotic timer response, as expected. Under the conditions we have used neither WIP1 inhibition nor genetic knockout of WIP1 in U2OS cells restores mitotic timer functionality (new Figure 7G-7K), suggesting that these cancer cell lines have additional mutations beyond WIP1 truncations that allow them to circumvent the mitotic timer response.

5) In relation to 4, does a WIP1 inhibitor, as used by Meitinger, restore timer function in WIP1 mutant cells exposed to high DNA damage, and if so, is this only when given prior to the DNA damage?

Response: This was addressed in point 4. Treating DNA damaged WIP1 Δ CT hTERT-RPE1 G2 cells with the WIP1 inhibitor causes them to collapse from G2 to G1 without cell division (Figure 2H-2I). Lowering WIP1 activity in WIP1 Δ CT cells therefore prevents these cells from entering mitosis in the presence of DNA damage, and stops them from over-riding the mitotic timer response, as expected.

6) Regarding this statement on line 209 "One difference was that WIP1 Δ CT cells took ~50% longer to complete mitosis after high levels of DNA damage, with a mean time in mitosis of 91 {plus minus} 35 min. Although these cells showed extended G1 length of 17.9 {plus minus} 6.5 h the majority were able to enter S phase, suggesting that in addition to an attenuation of the DNA damage response they bypass the mitotic timer response if DNA damage is present"

From Figure 2F I can see approx. 50% of cells that enter S-phase after transiting through mitosis and 50% that arrest in G1. Do the 50% that enter S-phase have a delayed mitosis, consistent with the conclusion that the timer response is bypassed? Also, what about the 50% that arrest in G1 - is their timer functional or do they arrest for some other reason? To answer these questions, it is crucial to plot mitotic arrest duration against G1 length/S-phase entry, to show that the G1 arrest specifically associated with a prolonged mitosis is abolished in dCT compared to WT cells.

Response: We have performed the analysis the reviewer suggested (see Rebuttal Figure 3) and find a correlation between time in mitosis and G1 length ($r = 0.5086$, $P = <0.0001$). This supports the view that WIP1 Δ CT cells exposed to DNA damage that experience a delayed mitosis are more resistant to G1 arrest, unless the time in mitosis is increased beyond a time window much greater than that required to cause a G1 arrest in undamaged WT cells (new Figure 8B-8C). To directly test this model, we performed single cell imaging of WIP1 Δ CT cells exposed to a high dose pulse of DNA damage in G2 and found that they spent longer in mitosis and had an altered threshold for G1 arrest compared to undamaged WIP1 Δ CT cells which arrest after 60-minute mitotic delays, similarly to wild-type cells (new Figure 8B and 8D).

Rebuttal Figure 3. Correlation between time in mitosis (h) and G1 length (h) for WIP1 Δ CT hTERT-RPE1 cells treated with high dose NCS (NCS^{HD}) for 1 h prior to tracking (related to Figure 2H). Spearman's correlation was calculated using GraphPad Prism and the line of best fit is shown in red.

7) In relation to the point above, can the authors rule out the alternative explanation that WIP1-dCT cells simply do not arrest in G1 after DNA damage, regardless of M duration?

WIP1 mutation has been shown previously to impair IR-induced G1 arrest (Kleiblova et al., 2013).

Response: First, it is important to note that WIP1 Δ CT cells can undergo p53-dependent cell cycle arrest after mitotic delays in the absence of DNA damage. We have carried out an extensive analysis of WIP1 wild-type, KO and Δ CT cells after normal and delayed mitosis in the presence and absence of DNA damage in the revised Figure 8. This shows that depending on the precise conditions of mitotic length and DNA damage WIP1 Δ CT cells can arrest in G1, albeit with altered thresholds for both pathways. We have also confirmed that WIP1 Δ CT cells are able to arrest in G1 in response to DNA damage, as expected. The threshold for G1 arrest is attenuated compared to wild-type cells (see Figure 2D-2E). This latter data agrees with Kleiblova et al., 2013 although noting that study did not explore mitotic timing.

8) The introduction to figure 3 makes the point that earlier reports have shown the destruction of cyclins in G2 after DNA damage, but they have not explored the relationship of the DDR with the mitotic timer. This indeed is an important novel aspect of this study, but the data looking at the relationship to the timer does not come until much later in figure 7. The intervening data is interesting and performed very well, but it seems largely an extension of what is already known and the key novel data on the timer does not come until the very last figure. I think the manuscript would benefit from condensing the proceeding data to focus on the key novel aspect and move the confirmation data into supplementary.

Response: While the intervening data can be seen as complementary and supportive of previous findings in the field, we felt it extremely important to show these data for the simple reason that everything is performed under standardised conditions and is therefore directly comparable between experiments. If we didn't show the response to G2 DNA damage in our different cell lines including wild-type cells, it would be challenging to compare how the response is attenuated in WIP1 Δ CT cells, and our other experimental conditions. As explained above, it is also necessary to monitor the levels of MDM2 under all the experimental conditions, which was not tested in the previous studies. Indeed, this is central to understanding how DNA damaged WIP1 Δ CT cells can overcome mitotic timer-induced G1 arrest.

Minor points

9) The CDK2 data is clinically relevant and this could warrant discussion. I would also add that Johmura et al (2014) do hint at CDK2 being the crucial target of p21, so the finding could also be compared and contrasted to relevant earlier work.

Response: We have extended the discussion to highlight these points.

10) Re this statement on line 80: "One caveat is that tumour cell lines deficient for the mitotic timer response carry many other mutations, and thus this correlation has not been confirmed as a causal change leading to attenuation of the timer response."

Meitinger et al (2024) show a partial recovery of the M timer in U2OS treated with WIP1 inhibitor

Response: We find that WIP1 gain-of-function mutations in U2OS, HCT116 and hTERT-RPE1 cells are not causal for loss of timer function, and that WIP1 KO in U2OS or HCT116 cells does not restore mitotic timer function (Figures 7 and 8, Figure EV4).

11) regarding line127: "These findings suggest that the level of DNA damage needed to sustain a delay in G2 is higher than the level needed to trigger delays in G1"

This is also consistent with Barr AR et al (2017, Nat Comms) and it could be explained by many reasons, including that incomplete replication creates DSBs following mitosis and/or that APC-CDH1 activity rises to prevent SKP2-mediated p21 degradation post-mitosis. An altered timer threshold is another possibility, as the authors suggest, but this earlier work and possibilities should be briefly mentioned.

Response: We have revised the text to include this study, and discuss the possibilities for this difference in line with this suggestion.

12) Regarding line 194: "However, like wild-type cells, delays in mitosis or treatment with a CDK4/6 inhibitor resulted in penetrant G1 arrest with only diploid WIP1 Δ CT cells (Figures 2D, WIP1 Δ CT CDK4/6i and M \rightarrow G1)."

This is an important result showing that WIP1 mutation does not abolish the mitotic timer. This could perhaps be emphasized here.

Response: We have revised the text to emphasise this important and supportive result.

13) Figure 2E shows numbers of 53BP1 foci as well as % cells entering M in WIP1 wt-dCT cell after NCS. WIP1 dCT cells have a substantial decrease in CHK2pT68 and γ H2AXpS139 after NCS treatment (Figure 2B), but there is no difference in the number of 53BP1 foci between WT & WIP1 dCT cells (Figure 2E). Can this difference be explained given γ H2AX recruits 53BP1 to DSBs? It has been shown previously that ectopic expression of WT WIP1 reduces 53BP1 recruitment (Macurek et al, 2013).

Response: Previous work has found that 53BP1 is recruited to DNA damage sites through multiple mechanisms including chromatin methylation and ubiquitination (PMID: 25695757; 22373579). We find that, although γ -H2A.X and p53BP1 do co-localise at sites of DNA damage, the recruitment of γ -H2A.X but not 53BP1 is affected by inhibition of ATM/ATR during the DNA damage response (Rebuttal Figure 4).

Rebuttal Figure 4: Wild-type hTERT-RPE1 cells treated with 100 ng/ml neocarzinostatin (NCS) to induce high dose DNA damage (HD DNA damage) for 1 h in the presence or absence of ATM/ATR inhibitors (2 μ M each). Following treatment, cells were fixed in 4% (w/v) paraformaldehyde and stained with the indicated antibodies and the DNA stain DAPI. White boxes depict enlarged nuclei on the right. Scale bars, 10 μ m.

14) Regarding line 289: "DNA damage extends G2 by delaying CDK1-cyclin B activation and inducing the broad spectrum CDK inhibitor p21. It was therefore important to understand

which of these pathways was required to divert G2 cells to G1."

The authors should refer to previous work demonstrating the requirement of p21 for mitotic skipping, e.g., Zeng et al (2023); Zuniga et al., (2024), Krenning et al (2014), Johmura et al (2014).

Response: we have revised the text to include these suggested studies, except for Zuniga *et al* as they look at oncogene-induced senescence, a phenomenon which we view as separate to G2 collapse following DNA damage since they report it is p53 independent.

Prof. Francis A. Barr
University of Oxford
Department of Biochemistry
South Parks Road
Oxford OX1 3QU
United Kingdom

8th Jun 2025

Re: EMBOJ-2025-120319R
WIP1 mutations suppress DNA damage triggered bypass of the mitotic timer

Dear Dr. Barr,

Thank you for submitting your revised manuscript to The EMBO Journal. It has now been re-reviewed by original referees 1 and 2, who were both fully satisfied with the revisions. We shall therefore be happy to accept the study for publication, as soon as a few remaining editorial issues have been addressed:

- Please upload all main Figures and all Expanded View figures as individual files with sufficient resolution/quality for production.
- Please adjust the order of the manuscript sections: Title page with complete author information, Abstract, Keywords, Introduction, Results, Discussion, Methods, Data Availability, Acknowledgements, Disclosure and Competing Interests Statement, References, Main Figure Legends, Tables, Expanded Figure Legends.
- On the abstract page of the manuscript, please include 4-5 general keyword terms to enhance searchability.
- Please rename the Conflict of Interest section into "Disclosure and Competing Interests Statement", in accordance with our updated Guide to Authors (<https://www.embopress.org/competing-interests>)
- As we are switching from a free-text author contribution statement towards a more formal statement based on Contributor Role Taxonomy (CRediT) terms, please remove the present Author Contribution section and instead specify each author's contribution(s) directly in the Author Information page of our submission system during upload of the final manuscript. See <https://casrai.org/credit/> for more information.
- Please move all funding information into the Acknowledgements section.
- Please carefully go through the reference list and make sure that each reference is complete with citation year, volume, and page/locator numbers (currently missing for several of them).
- Please move the Reagents and Tools table from the main article file, and upload it as a separate text file. Also, please make sure to adhere to the template table downloadable from our author guidelines: <https://www.embopress.org/page/journal/14693178/authorguide#structuredmethods>
- Regarding Appendix Figures S1/2: These should either be compiled, together with their respective legends below each of them, in a single "Appendix" PDF, headed by a brief title page with a table of contents; and referenced as "Appendix Figures S1/2" throughout the text and within the Appendix. Alternatively, you might simply convert them into additional Expanded View figures, making sure to update EV figure numbering, legend and in-text references accordingly.
- Finally, please provide suggestions for a short 'blurb' text prefacing and summing up the conceptual aspect of the study in two sentences (max. 250 characters), followed by 3-5 one-sentence 'bullet points' with brief factual statements of key results of the paper; they will form the basis of an editor-written 'Synopsis' accompanying the online version of the article. Please also upload a synopsis image, which can be used as a "visual title" for the synopsis section of your paper. The image should be in PNG or JPG format, and please make sure that it remains in the modest dimensions of (exactly) 550 pixels wide and 300-600 pixels high.

I am returning the manuscript to you for a final round of minor revision, solely to allow you to make these modifications and upload the revised files. Once we will have received them, we should be ready to swiftly proceed with formal acceptance and production of the manuscript.

Yours sincerely,

Hartmut Vodermaier

*** PLEASE NOTE: All revised manuscripts are subject to initial checks for completeness and adherence to our formatting guidelines. Revisions may be returned to the authors and delayed in their editorial re-evaluation if they fail to comply to the following requirements (see also our Guide to Authors for further information):

- 1) Every manuscript requires a Data Availability section (even if only stating that no deposited datasets are included). Primary datasets or computer code produced in the current study have to be deposited in appropriate public repositories prior to resubmission, and reviewer access details provided in case that public access is not yet allowed. Further information: embopress.org/page/journal/14602075/authorguide#dataavailability
- 2) Each figure legend must specify
 - size of the scale bars that are mandatory for all micrograph panels
 - the statistical test used to generate error bars and P-values
 - the type error bars (e.g., S.E.M., S.D.)
 - the number (n) and nature (biological or technical replicate) of independent experiments underlying each data point
 - Figures may not include error bars for experiments with $n < 3$; scatter plots showing individual data points should be used instead.
- 3) Revised manuscript text (including main tables, and figure legends for main and EV figures) has to be submitted as editable text file (e.g., .docx format). We encourage highlighting of changes (e.g., via text color) for the referees' reference.
- 4) Each main and each Expanded View (EV) figure should be uploaded as individual production-quality files (preferably in .eps, .tif, .jpg formats). For suggestions on figure preparation/layout, please refer to our Figure Preparation Guidelines: <http://bit.ly/EMBOPressFigurePreparationGuideline>
- 5) Point-by-point response letters should include the original referee comments in full together with your detailed responses to them (and to specific editor requests if applicable), and also be uploaded as editable (e.g., .docx) text files.
- 6) Please complete our Author Checklist, and make sure that information entered into the checklist is also reflected in the manuscript; the checklist will be available to readers as part of the Review Process File. A download link is found at the top of our Guide to Authors: embopress.org/page/journal/14602075/authorguide
- 7) All authors listed as (co-)corresponding need to deposit, in their respective author profiles in our submission system, a unique ORCID identifier linked to their name. Please see our Guide to Authors for detailed instructions.
- 8) Please note that supplementary information at EMBO Press has been superseded by the 'Expanded View' for inclusion of additional figures, tables, movies or datasets; with up to five EV Figures being typeset and directly accessible in the HTML version of the article. For details and guidance, please refer to: embopress.org/page/journal/14602075/authorguide#expandedview
- 9) To facilitate reproducibility and cross-laboratory adoption of methodologies, please structure the Materials & Methods section as outlined in our guide to authors, including a completed Reagents and Tools Table that can be downloaded from our author guidelines as well (<https://www.embopress.org/page/journal/14602075/authorguide#structuredmethods>).
- 10) Digital image enhancement is acceptable practice, as long as it accurately represents the original data and conforms to community standards. If a figure has been subjected to significant electronic manipulation, this must be clearly noted in the figure legend and/or the 'Materials and Methods' section. The editors reserve the right to request original versions of figures and the original images that were used to assemble the figure. Finally, we generally encourage uploading of numerical as well as gel/blot image source data; for details see: embopress.org/page/journal/14602075/authorguide#sourcedata

In the interest of ensuring the conceptual advance provided by the work, we recommend submitting a revision within 3 months (6th Sep 2025). Please discuss the revision progress ahead of this time with the editor if you require more time to complete the revisions. Use the link below to submit your revision:

Link Not Available

Referee #1:

In the revised manuscript, the authors edited the text and included new data to clarify the issues raised after the first submission. In particular, they include new controls in Fig. EV3 that support their original conclusion that p21 triggers mitotic bypass by inhibition of CDK2 followed by activation of APC/C and destruction of cyclin A/B. Further, they generated WIP1 knock-out cells that supported their conclusion that WIP1 does not participate in the mitotic timer in non-stressed conditions but becomes important after induction of DNA damage (Fig. 7E-K). These conclusions are in good agreement with recent findings focusing on the oncogenic potential of pathogenic WIP1 variants and this literature is now correctly cited. Overall, the authors successfully addressed all my points and I enthusiastically recommend the manuscript for publication.

Referee #2:

Although I did not raise specific concerns about novelty in my initial review, I find that the comments raised by the other reviewers on this point have been addressed convincingly. The authors provided a clear rationale for revisiting previously described experiments, emphasizing methodological differences and the need for consistent experimental conditions. They also expanded the manuscript with new data and reorganized the text to highlight the truly novel aspects, particularly the uncoupling of the DNA damage checkpoint and the mitotic timer. I therefore consider that the concerns regarding novelty have been adequately addressed.